# Distributionally Robust Skeleton Learning of Discrete Bayesian Networks

**Yeshu Li**[*]
Alibaba Group
liyeshu.lys@alibaba-inc.com

**Brian D. Ziebart**
Department of Computer Science
University of Illinois at Chicago
bziebart@uic.edu

## Abstract

We consider the problem of learning the exact skeleton of general discrete Bayesian networks from potentially corrupted data. Building on distributionally robust optimization and a regression approach, we propose to optimize the most adverse risk over a family of distributions within bounded Wasserstein distance or KL divergence to the empirical distribution. The worst-case risk accounts for the effect of outliers. The proposed approach applies for general categorical random variables without assuming faithfulness, an ordinal relationship or a specific form of conditional distribution. We present efficient algorithms and show the proposed methods are closely related to the standard regularized regression approach. Under mild assumptions, we derive non-asymptotic guarantees for successful structure learning with logarithmic sample complexities for bounded-degree graphs. Numerical study on synthetic and real datasets validates the effectiveness of our method.

## 1  Introduction

A Bayesian network is a prominent class of probabilistic graphical models that encodes the conditional dependencies among variables with a directed acyclic graph (DAG). It provides a mathematical framework for formally understanding the interaction among variables of interest, together with computationally attractive factorization for modeling multivariate distributions. If we impose causal relationships on the edges between variables, the model becomes a causal Bayesian network that encodes the more informative causation. Without such interpretation, a Bayesian network serves as a dependency graph for factorization of a multivariate distribution. We focus on discrete Bayesian networks with purely categorical random variables that are not ordinal, but will discuss related work on both discrete and continuous Bayesian networks for completeness.

The DAG structure of a Bayesian network is typically unknown in practice [Natori et al., 2017, Kitson et al., 2023]. Structure learning is therefore an important task that infers the structure from data. The *score-based* approach defines a scoring function that measures the goodness-of-fit of each structure and aims to find an optimal DAG that maximizes the score. Unfortunately, the resulting combinatorial optimization problem is known to be NP-hard [Chickering et al., 2004] without distributional assumptions. Representative approaches include those based on heuristic search [Chickering, 2002], dynamic programming [Silander and Myllymäki, 2006], integer linear programming [Jaakkola et al., 2010] or continuous optimization [Zheng et al., 2018], which either yields an approximate solution or an exact solution in worst-case exponential time. The *constraint-based* approach [Spirtes and Glymour, 1991, Spirtes et al., 1999, Colombo et al., 2014] performs conditional independence tests to determine the existence and directionality of edges. The time complexity is, however, exponential with the maximum in-degree. Furthermore, the independence test results may be unreliable or inconsistent with the true distribution because of finite samples or

---

[*]Work done when Yeshu was a PhD student at UIC.

even corrupted samples. In general, without interventional data or assumptions on the underlying distribution, we can only identify a Markov equivalence class (MEC) the true DAG belongs to from observational data where DAGs in the MEC are Markov equivalent, that is, encoding the same set of conditional independencies.

A super-structure is an undirected graph that contains the skeleton as a subgraph which removes directionality from the true DAG. It has been shown that a given super-structure possibly reduces the search space or the number of independence tests to be performed. For example, exact structure learning of Bayesian networks may be (fixed-parameter) tractable [Downey and Fellows, 1995] if the super-structure satisfies certain graph-theoretic properties such as bounded tree-width [Korhonen and Parviainen, 2013, Loh and Bühlmann, 2014], bounded maximum degree [Ordyniak and Szeider, 2013] and the feedback edge number [Ganian and Korchemna, 2021]. An incomplete super-structure with missing edges also helps improve the learned DAG with a post-processing hill-climbing method [Tsamardinos et al., 2006, Perrier et al., 2008]. Furthermore, a combination of a skeleton and a variable ordering determines a unique DAG structure. Learning the exact skeleton rather than a rough super-structure is desirable in Bayesian network structure learning.

Spirtes and Glymour [1991], Tsamardinos et al. [2006] make use of independence tests to estimate the skeleton. Loh and Bühlmann [2014] learn a super-structure called moralized graph via graphical lasso [Friedman et al., 2008]. Shojaie and Michailidis [2010] learn the skeleton assuming an ordering of variables. Bank and Honorio [2020] leverage linear regression for skeleton recovery in polynomial time. These methods either rely on independence test results, which are unstable, or a regularized empirical risk minimization problem, where regularization is usually heuristically chosen to combat overfitting. In practice, the observational data is commonly contaminated by sensor failure, transmission error or adversarial perturbation [Lorch et al., 2022, Sankararaman et al., 2022, Kitson et al., 2023]. Sometimes only a small amount of data is available for learning. As a result, the existing algorithms are vulnerable to such distributional uncertainty and may produce false or missing edges in the estimated skeleton.

In this paper, we propose a distributionally robust optimization (DRO) method [Rahimian and Mehrotra, 2019] that solves a node-wise multivariate regression problem [Bank and Honorio, 2020] for skeleton learning of general discrete Bayesian networks to overcome the above limitations. We do not assume any specific form of conditional distributions. We take into account the settings with a small sample size and potential perturbations, which makes the true data generating distribution highly uncertain. Our method explicitly models the uncertainty by constructing an ambiguity set of distributions characterized by certain a priori properties of the true distribution. The optimal parameter is learned by minimizing the worst-case expected loss over all the distributions within the ambiguity set so that it performs uniformly well on all the considered distributions. The ambiguity set is usually defined in such a way that it includes all the distributions close to the empirical distribution in terms of some divergence. With an appropriately chosen divergence measure, the set contains the true distribution with high probability. Hence the worst-case risk can be interpreted as an upper confidence bound of the true risk. The fact that a discrete Bayesian network encompasses an exponential number of states may pose a challenge to solve the DRO problem. We develop efficient algorithms for problems with ambiguity sets defined by Wasserstein distances and Kullback–Leibler (KL) divergences. We show that a group regularized regression method is a special case of our approach. We study statistical guarantees of the proposed estimators such as sample complexities. Experimental results on synthetic and real-world datasets contaminated by various perturbations validate the superior performance of the proposed methods.

## 1.1 Related Work

Bayesian networks have been widely adopted in a number of applications such as gene regulatory networks [Werhli et al., 2006], medical decision making [Kyrimi et al., 2020] and spam filtering [Manjusha and Kumar, 2010].

In addition to the score-based structure learning methods and constraint-based methods discussed in the introduction section, there are a third class of hybrid algorithms leveraging constraint-based methods to restrict the search space of a score-based method [Tsamardinos et al., 2006, Gasse et al., 2014, Nandy et al., 2018]. There is also a flurry of work on score-based methods based on neural networks and continuous optimization [Zheng et al., 2018, Wei et al., 2020, Ng et al., 2020, Yu et al., 2021, Ng et al., 2022, Gao et al., 2022], motivated by differentiable characterization of acyclicity

without rigorous theoretical guarantees. We refer the interested readers to survey papers [Drton and Maathuis, 2017, Heinze-Deml et al., 2018, Constantinou et al., 2021] for a more thorough introduction of DAG structure learning and causal discovery methods.

Recently, there is an emerging line of work proposing polynomial-time algorithms for DAG learning [Park and Raskutti, 2017, Ghoshal and Honorio, 2017, 2018, Chen et al., 2019, Bank and Honorio, 2020, Gao et al., 2020, Rajendran et al., 2021], among which Bank and Honorio [2020] particularly focuses on general discrete Bayesian networks without resorting to independence tests.

Learning a super-structure can be done by independence tests, graphical lasso or regression, as discussed in introduction. Given a super-structure, how to determine the orientation has been studied by Perrier et al. [2008], Ordyniak and Szeider [2013], Korhonen and Parviainen [2013], Loh and Bühlmann [2014], Ng et al. [2021], Ganian and Korchemna [2021].

DRO is a powerful framework emerging from operations research [Delage and Ye, 2010, Blanchet and Murthy, 2019, Shafieezadeh-Abadeh et al., 2019, Duchi and Namkoong, 2019] and has seen effective applications in many graph learning problems such as inverse covariance estimation [Nguyen et al., 2022], graphical lasso learning [Cisneros-Velarde et al., 2020], graph Laplacian learning [Wang et al., 2021], Markov random field (MRF) parameter learning [Fathony et al., 2018], MRF structure learning [Li et al., 2022] and causal inference [Bertsimas et al., 2022].

## 2 Preliminaries

We introduce necessary background and a baseline method for skeleton learning of Bayesian networks.

### 2.1 Notations

We refer to $[n]$ as the index set $\{1, 2, \ldots, n\}$. For a vector $\boldsymbol{x} \in \mathbb{R}^n$, we use $x_i$ for its $i$-th element and $\boldsymbol{x}_{\mathcal{S}}$ for the subset of elements indexed by $\mathcal{S} \subseteq [n]$ with $\bar{i} \triangleq [n] \backslash \{i\}$. For a matrix $\boldsymbol{A} \in \mathbb{R}^{n \times m}$, we use $A_{ij}$, $\boldsymbol{A}_{i\cdot}$ and $\boldsymbol{A}_{\cdot j}$ to denote its $(i, j)$-th entry, $i$-th row and $j$-th column respectively. $\boldsymbol{A}_{\mathcal{S}\mathcal{T}}$ represents the submatrix of $\boldsymbol{A}$ with rows restricted to $\mathcal{S}$ and columns restricted to $\mathcal{T} \subseteq [m]$. We define a row-partitioned block matrix as $\boldsymbol{A} \triangleq [\boldsymbol{A}_1 \boldsymbol{A}_2 \cdots \boldsymbol{A}_k]^\intercal \in \mathbb{R}^{\sum_i n_i \times m}$ where $\boldsymbol{A}_i \in \mathbb{R}^{n_i \times m}$. The $\ell_p$-norm of a vector $\boldsymbol{x}$ is defined as $\|\boldsymbol{x}\|_p := (\sum_i |x_i|^p)^{1/p}$ with $|\cdot|$ being the absolute value function. The $\ell_{p,q}$ norm of a matrix $\boldsymbol{A}$ is defined as $\|\boldsymbol{A}\|_{p,q} := (\sum_j \|\boldsymbol{A}_{\cdot j}\|_p^q)^{1/q}$. When $p = q = 2$, it becomes the Frobenius norm $\|\cdot\|_F$. The operator norm is written as $\|\boldsymbol{A}\|_{p,q} := \sup_{\|\boldsymbol{v}\|_p = 1} \|\boldsymbol{A}\boldsymbol{v}\|_q$. The block matrix norm is defined as $\|\boldsymbol{A}\|_{B,p,q} := (\sum_{i=1}^k \|\boldsymbol{A}_i\|_p^q)^{1/q}$. The inner product of two matrices is designated by $\langle \boldsymbol{A}, \boldsymbol{B} \rangle \triangleq \mathrm{Tr}[\boldsymbol{A}^\intercal \boldsymbol{B}]$ where $\boldsymbol{A}^\intercal$ is the transpose of $\boldsymbol{A}$. Denote by $\otimes$ the tensor product operation. With a slight abuse of notation, $|\mathcal{S}|$ stands for the cardinality of a set $\mathcal{S}$. We denote by $\boldsymbol{1}$ ($\boldsymbol{0}$) a vector or matrix of all ones (zeros). Given a distribution $\mathbb{P}$ on $\Xi$, we denote by $\mathbb{E}_{\mathbb{P}}$ the expectation under $\mathbb{P}$. The least $c$-Lipschitz constant of a function $f : \Xi \to \mathbb{R}$ with a metric $c : \Xi \times \Xi \to \mathbb{R}$ is written as $\mathrm{lip}_c(f) := \inf \Lambda_c(f)$ where $\Lambda_c(f) := \{\lambda > 0 : \forall \xi_1, \xi_2 \in \Xi \quad |f(\xi_1) - f(\xi_2)| \leqslant \lambda c(\xi_1, \xi_2)\}$.

### 2.2 Bayesian Network Skeleton Learning

Let $\mathbb{P}$ be a discrete joint probability distribution on $n$ categorical random variables $\mathcal{V} := \{X_1, X_2, \ldots, X_n\}$. Let $\mathcal{G} := (\mathcal{V}, \mathcal{E}_{\text{true}})$ be a DAG with edge set $\mathcal{E}_{\text{true}}$. We use $X_i$ to represent the $i$-th random variable or node interchangeably. We call $(\mathcal{G}, \mathbb{P})$ a Bayesian network if it satisfies the Markov condition, i.e., each variable $X_r$ is independent of any subset of its non-descendants conditioned on its parents $\mathbf{Pa}_r$. We denote the children of $X_r$ by $\mathbf{Ch}_r$, its neighbors by $\mathbf{Ne}_r := \mathbf{Pa}_r \cup \mathbf{Ch}_r$ and the complement by $\mathbf{Co}_r := [n] - \mathbf{Ne}_r - \{r\}$. The joint probability distribution can thus be factorized in terms of local conditional distributions:

$$\mathbb{P}(\boldsymbol{X}) = \mathbb{P}(X_1, X_2, \ldots, X_n) \triangleq \prod_{i=1}^n \mathbb{P}(X_i | \mathbf{Pa}_i).$$

Let $\mathcal{G}_{\text{skel}} := (\mathcal{V}, \mathcal{E}_{\text{skel}})$ be the undirected graph that removes directionality from $\mathcal{G}$. Given $m$ samples $\{\boldsymbol{x}^{(i)}\}_{i=1}^m$ drawn i.i.d. from $\mathbb{P}$, the goal of skeleton learning is to estimate $\mathcal{G}_{\text{skel}}$ from the samples.

We do not assume faithfulness [Spirtes et al., 2000] or any specific parametric form for the conditional distributions. The distribution is faithful to a graph if all (conditional) independencies that hold true

in the distribution are entailed by the graph, which is commonly violated in practice [Uhler et al., 2013, Mabrouk et al., 2014]. The unavailability of a true model entails a substitute model. Bank and Honorio [2020] propose such a model based on encoding schemes and surrogate parameters.

Assume that each variable $X_r$ takes values from a finite set $\mathcal{C}_r$ with cardinality $|\mathcal{C}_r| > 1$. For an indexing set $\mathcal{S} \subseteq [n]$, define $\rho_{\mathcal{S}} := \sum_{i \in \mathcal{S}} |\mathcal{C}_i| - 1$ and $\rho_{\mathcal{S}}^+ := \sum_{i \in \mathcal{S}} |\mathcal{C}_i|$. The maximum cardinality minus one is defined as $\rho_{\max} := \max_{i \in [n]} |\mathcal{C}_i| - 1$. Let $\mathcal{S}_r := \bigcup_{i \in \mathbf{Ne}_r} \{\rho_{[i-1]} + 1, \ldots, \rho_{[i]}\}$ be indices for $\mathbf{Ne}_r$ in $\rho_{[n]}$ and its complement by $\mathcal{S}_r^c := [\rho_{[n]}] - \mathcal{S}_r - \{\rho_{[r-1]} + 1, \ldots, \rho_{[r]}\}$. Let $\mathcal{E} : \mathcal{C}_r \to \mathcal{B}^{\rho_r}$ be an encoding mapping with a bounded and countable set $\mathcal{B} \subset \mathbb{R}$. We adopt encoding schemes with $\mathcal{B} = \{-1, 0, 1\}$ such as dummy encoding and unweighted effects encoding[2] which satisfy a linear independence condition. With a little abuse of notation, we reuse $\mathcal{E}$ for encoding any $X_r$ and denote by $\mathcal{E}(\boldsymbol{X}_{\mathcal{S}}) \in \mathcal{B}^{\rho_{\mathcal{S}}}$ the concatenation of the encoded vectors $\{\mathcal{E}(X_i)\}_{i \in \mathcal{S}}$. Consider a linear structural equation model for each $X_r$: $\mathcal{E}(X_r) = \boldsymbol{W}^{*\mathsf{T}}\mathcal{E}(\boldsymbol{X}_{\bar{r}}) + \boldsymbol{e}$, where $\boldsymbol{W}^* \triangleq [\boldsymbol{W}_1^* \cdots \boldsymbol{W}_{r-1}^* \boldsymbol{W}_{r+1}^* \cdots \boldsymbol{W}_n^*]^{\mathsf{T}} \in \mathbb{R}^{\rho_{\bar{r}} \times \rho_r}$ with $\boldsymbol{W}_i^* \in \mathbb{R}^{\rho_i \times \rho_r}$ is a surrogate parameter matrix and $\boldsymbol{e} \in \mathbb{R}^{\rho_r}$ is a vector of errors not necessarily independent of other quantities. A natural choice of a fixed $\boldsymbol{W}^*$ is the solution to the following problem given knowledge of the true Bayesian network:

$$\boldsymbol{W}^* \in \arg\inf_{\boldsymbol{W}} \frac{1}{2}\mathbb{E}_{\mathbb{P}}\|\mathcal{E}(X_r) - \boldsymbol{W}^{\mathsf{T}}\mathcal{E}(\boldsymbol{X}_{\bar{r}})\|_2^2 \quad \text{s.t.} \quad \boldsymbol{W}_i = \boldsymbol{0} \quad \forall i \in \mathbf{Co}_r. \tag{1}$$

Therefore $\boldsymbol{W}^* = (\boldsymbol{W}_{\mathcal{S}_r}^*; \boldsymbol{0})$ with $\boldsymbol{W}_{\mathcal{S}_r}^* = \mathbb{E}_{\mathbb{P}}[\mathcal{E}(\boldsymbol{X}_{\bar{r}})_{\mathcal{S}_r}\mathcal{E}(\boldsymbol{X}_{\bar{r}})_{\mathcal{S}_r}^{\mathsf{T}}]^{-1}\mathbb{E}_{\mathbb{P}}[\mathcal{E}(\boldsymbol{X}_{\bar{r}})_{\mathcal{S}_r}\mathcal{E}(X_r)^{\mathsf{T}}]$ is the optimal solution by the first-order optimality condition assuming that $\mathbb{E}_{\mathbb{P}}[\mathcal{E}(\boldsymbol{X}_{\bar{r}})_{\mathcal{S}_r}\mathcal{E}(\boldsymbol{X}_{\bar{r}})_{\mathcal{S}_r}^{\mathsf{T}}]$ is invertible. The expression of $\boldsymbol{W}_{\mathcal{S}_r}^*$ captures the intuitions that neighbor nodes should be highly related to the current node $r$ while the interaction among neighbor nodes should be weak for them to be distinguishable. We further assume that the errors are bounded:

**Assumption 1** (Bounded error). For the error vector, $\|\boldsymbol{e}\|_\infty \leq \sigma$ and $\|\mathbb{E}_{\mathbb{P}}[|\boldsymbol{e}|]\|_\infty \leq \mu$.

Note that the true distribution does not have to follow a linear structural equation model. Equation (1) only serves as a surrogate model to find technical conditions for successful skeleton learning, which will be discussed in a moment.

The surrogate model under the true distribution indicates that $\|\boldsymbol{W}_i^*\|_{2,2} > 0 \implies X_i \in \mathbf{Ne}_r$. This suggests a regularized empirical risk minimization (ERM) problem to estimate $\boldsymbol{W}^*$:

$$\tilde{\boldsymbol{W}} \in \arg\inf_{\boldsymbol{W}} \tilde{L}(\boldsymbol{W}) := \frac{1}{2}\mathbb{E}_{\tilde{\mathbb{P}}_m}\|\mathcal{E}(X_r) - \boldsymbol{W}^{\mathsf{T}}\mathcal{E}(\boldsymbol{X}_{\bar{r}})\|_2^2 + \tilde{\lambda}\|\boldsymbol{W}\|_{B,2,1}, \tag{2}$$

where $\tilde{\lambda} > 0$ is a regularization coefficient, the block $\ell_{2,1}$ norm is adopted to induce sparsity and $\tilde{\mathbb{P}}_m := \frac{1}{m}\sum_{i=1}^m \delta_{\boldsymbol{x}^{(i)}}$ stands for the empirical distribution with $\delta_{\boldsymbol{x}^{(i)}}$ being the Dirac point measure at $\boldsymbol{x}^{(i)}$. This approach is expected to succeed as long as only neighbor nodes have a non-trivial impact on the current node, namely, $\|\boldsymbol{W}_i^*\|_{2,2} > 0 \iff X_i \in \mathbf{Ne}_r$.

Define the risk of some $\boldsymbol{W}$ under a distribution $\tilde{\mathbb{P}}$ as

$$R^{\tilde{\mathbb{P}}}(\boldsymbol{W}) := \mathbb{E}_{\tilde{\mathbb{P}}}\ell_{\boldsymbol{W}}(\boldsymbol{X}) := \mathbb{E}_{\tilde{\mathbb{P}}}\frac{1}{2}\|\mathcal{E}(X_r) - \boldsymbol{W}^{\mathsf{T}}\mathcal{E}(\boldsymbol{X}_{\bar{r}})\|_2^2,$$

where $\ell_{\boldsymbol{W}}(\cdot)$ is the squared loss function. The Hessian of the empirical risk $R^{\tilde{\mathbb{P}}_m}(\boldsymbol{W})$ is a block diagonal matrix $\nabla^2 R^{\tilde{\mathbb{P}}_m}(\boldsymbol{W}) \triangleq \tilde{\boldsymbol{H}} \otimes \boldsymbol{I}_{\rho_r} \in \mathbb{R}^{\rho_r\rho_{\bar{r}} \times \rho_r\rho_{\bar{r}}}$, where $\tilde{\boldsymbol{H}} := \mathbb{E}_{\tilde{\mathbb{P}}_m}[\mathcal{E}(\boldsymbol{X}_{\bar{r}})\mathcal{E}(\boldsymbol{X}_{\bar{r}})^{\mathsf{T}}] \in \mathbb{R}^{\rho_{\bar{r}} \times \rho_{\bar{r}}}$ and $\boldsymbol{I}_{\rho_r} \in \mathbb{R}^{\rho_r \times \rho_r}$ is the identity matrix of dimension $\rho_r$. Similarly under the true distribution, $\boldsymbol{H} := \mathbb{E}_{\mathbb{P}}[\mathcal{E}(\boldsymbol{X}_{\bar{r}})\mathcal{E}(\boldsymbol{X}_{\bar{r}})^{\mathsf{T}}]$. As a result, $\boldsymbol{H}$ is independent of the surrogate parameters $\boldsymbol{W}^*$ thus conditions on the Hessian translate to conditions on a matrix of cross-moments of encodings, which only depend on the encoding function $\mathcal{E}$ and $\mathbb{P}$.

In order for this baseline method to work, we make the following assumptions.

**Assumption 2** (Minimum weight). For each node $r$, the minimum norm of the true weight matrix $\boldsymbol{W}^*$ for neighbor nodes is lower bounded: $\min_{i \in \mathbf{Ne}_r}\|\boldsymbol{W}_i\|_F \geq \beta > 0$.

**Assumption 3** (Positive definiteness of the Hessian). For each node $r$, $\boldsymbol{H}_{\mathcal{S}_r, \mathcal{S}_r} > 0$, or equivalently, $\Lambda_{\min}(\boldsymbol{H}_{\mathcal{S}_r, \mathcal{S}_r}) \geq \Lambda > 0$ where $\Lambda_{\min}(\cdot)$ denotes the minimum eigenvalue.

---

[2]If there are four variables, dummy encoding may adopt $\{(1,0,0), (0,1,0), (0,0,1), (0,0,0)\}$ whereas unweighted effects encoding may adopt $\{(1,0,0), (0,1,0), (0,0,1), (-1,-1,-1)\}$ as encoding vectors.

**Assumption 4** (Mutual incoherence). For each node $r$, $\|\boldsymbol{H}_{\mathcal{S}_r^c \mathcal{S}_r} \boldsymbol{H}_{\mathcal{S}_r \mathcal{S}_r}^{-1}\|_{B,1,\infty} \leqslant 1 - \alpha$ for some $0 < \alpha \leqslant 1$.

Assumption 2 guarantees that the influence of neighbor nodes is significant in terms of a non-zero value bounded away from zero, otherwise they will be indistinguishable from those with zero weight. Assumption 3 ensures that Equation (2) yields a unique solution. Assumption 4 is a widely adopted assumption that controls the impact of non-neighbor nodes on $r$ [Wainwright, 2009, Ravikumar et al., 2010, Daneshmand et al., 2014]. One interpretation is that the rows of $\boldsymbol{H}_{\mathcal{S}_r^c \mathcal{S}_r}$ should be nearly orthogonal to the rows of $\boldsymbol{H}_{\mathcal{S}_r \mathcal{S}_r}$. Bank and Honorio [2020] show that these assumptions hold for common encoding schemes and finite-sample settings with high probability under mild conditions. They also show that incoherence is more commonly satisfied for the neighbors than the Markov blanket, which justifies the significance of skeleton learning.

Finally, we take the union of all the learned neighbor nodes for each $r \in [n]$ by solving Equation (2) to get the estimated skeleton $\tilde{\mathcal{G}} := (\mathcal{V}, \tilde{\mathcal{E}}_{\text{skel}})$.

## 3 Method

As noted in Bank and Honorio [2020], due to model misspecification, even in the infinite sample setting, there is possible discrepancy between the ERM minimizer $\tilde{\boldsymbol{W}}$ and the true solution $\boldsymbol{W}^*$, resulting in false or missing edges. In the high-dimensional setting ($m < n$) or the adversarial setting, this issue becomes more serious due to limited knowledge about the data-generating mechanism $\mathbb{P}$.

In this section, we attempt to leverage a DRO framework to incorporate distributional uncertainty into the estimation process. We present efficient algorithms and study the theoretical guarantees of our methods. All technical proofs are deferred to the supplementary materials.

### 3.1 Basic Formulation

Let $\mathcal{X}$ be a measurable space of all states of the Bayesian network $(\mathcal{G}, \mathbb{P})$, i.e., $\boldsymbol{X} \in \mathcal{X}$. Let $\mathcal{P}(\mathcal{X})$ be the space of all Borel probability measures on $\mathcal{X}$. Denote by $\mathcal{X}^{\mathcal{E}} := \{\mathcal{E}(\boldsymbol{X}) : \forall \boldsymbol{X} \in \mathcal{X}\}$ the space of all the allowed encodings.

Instead of minimizing the empirical risk and relying on regularization, we seek a distributionally robust estimator that optimizes the worst-case risk over an ambiguity set of distributions:

$$\hat{\boldsymbol{W}} \in \arg\inf_{\boldsymbol{W}} \sup_{\mathbb{Q} \in \mathcal{A}} \frac{1}{2} \mathbb{E}_{\mathbb{Q}} \|\mathcal{E}(X_r) - \boldsymbol{W}^{\mathsf{T}} \mathcal{E}(\boldsymbol{X}_{\bar{r}})\|_2^2, \tag{3}$$

where $\mathcal{A} \subseteq \mathcal{P}(\mathcal{X})$ is an ambiguity set typically defined by a nominal probability measure $\tilde{\mathbb{P}}$ equipped with a discrepancy measure $\text{div}(\cdot, \cdot)$ for two distributions $\mathcal{A}_\varepsilon^{\text{div}}(\tilde{\mathbb{P}}) := \{\mathbb{Q} \in \mathcal{P}(\mathcal{X}) : \text{div}(\mathbb{Q}, \tilde{\mathbb{P}}) \leqslant \varepsilon\}$, where $\varepsilon$ is known as the ambiguity radius or size. This way of uncertainty quantification can be interpreted as an adversary that captures out-of-sample effect by making perturbations on samples within some budget $\varepsilon$. Some common statistical distances satisfy $\text{div}(\mathbb{Q}, \mathbb{P}) = 0 \iff \mathbb{Q} = \mathbb{P}$. In this case, if $\varepsilon$ is set to zero, Equation (3) reduces to Equation (2) without regularization. We will show that the DRO estimator $\hat{\boldsymbol{W}}$ can be found efficiently and encompasses attractive statistical properties with a judicious choice of $\mathcal{A}$.

### 3.2 Wasserstein DRO

Wasserstein distances or Kantorovich–Rubinstein metric in optimal transport theory can be interpreted as the cost of the optimal transport plan to move the mass from $\mathbb{P}$ to $\mathbb{Q}$ with unit transport cost $c : \mathcal{X} \times \mathcal{X} \rightarrow \mathbb{R}_+$. Denote by $\mathcal{P}_p(\mathcal{X})$ the space of all $\mathbb{P} \in \mathcal{P}(\mathcal{X})$ with finite $p$-th moments for $p \geqslant 1$. Let $\mathcal{M}(\mathcal{X}^2)$ be the set of probability measures on the product space $\mathcal{X} \times \mathcal{X}$. The $p$-Wasserstein distance between two distributions $\mathbb{P}, \mathbb{Q} \in \mathcal{P}_p(\mathcal{X})$ is defined as $W_p(\mathbb{P}, \mathbb{Q}) :=$

$\inf_{\Pi \in \mathcal{M}(\mathcal{X}^2)} \left\{ \left[ \int_{\mathcal{X}^2} c^p(\boldsymbol{x}, \boldsymbol{x}') \Pi(\mathrm{d}\boldsymbol{x}, \mathrm{d}\boldsymbol{x}') \right]^{\frac{1}{p}} : \Pi(\mathrm{d}\boldsymbol{x}, \mathcal{X}) = \mathbb{P}(\mathrm{d}\boldsymbol{x}), \Pi(\mathcal{X}, \mathrm{d}\boldsymbol{x}') = \mathbb{Q}(\mathrm{d}\boldsymbol{x}') \right\}.$

We adopt the Wasserstein distance of order $p = 1$ as the discrepancy measure, the empirical distribution as the nominal distribution, and cost function $c(\boldsymbol{x}, \boldsymbol{x}') = \|\mathcal{E}(\boldsymbol{x}) - \mathcal{E}(\boldsymbol{x}')\|$ for some norm

$\|\cdot\|$. The primal DRO formulation becomes

$$\hat{\boldsymbol{W}} \in \arg\inf_{\boldsymbol{W}} \sup_{\mathbb{Q} \in \mathcal{A}_{\varepsilon}^{W_p}(\tilde{\mathbb{P}}_m)} \frac{1}{2}\mathbb{E}_{\mathbb{Q}}\|\mathcal{E}(X_r) - \boldsymbol{W}^\intercal \mathcal{E}(\boldsymbol{X}_{\bar{r}})\|_2^2. \tag{4}$$

According to Blanchet and Murthy [2019], the dual problem of Equation (4) can be written as

$$\inf_{\boldsymbol{W},\gamma \geqslant 0} \gamma\varepsilon + \frac{1}{m}\sum_{i=1}^m \sup_{\boldsymbol{x} \in \mathcal{X}} \frac{1}{2}\|\mathcal{E}(x_r) - \boldsymbol{W}^\intercal \mathcal{E}(\boldsymbol{x}_{\bar{r}})\|_2^2 - \gamma\|\mathcal{E}(\boldsymbol{x}) - \mathcal{E}(\boldsymbol{x}^{(i)})\|. \tag{5}$$

Strong duality holds according to Theorem 1 in Gao and Kleywegt [2022]. The inner supremum problems can be solved independently for each $\boldsymbol{x}^{(i)}$. Henceforth, we focus on solving it for some $i \in [m]$:

$$\sup_{\boldsymbol{x} \in \mathcal{X}} \frac{1}{2}\|\mathcal{E}(x_r) - \boldsymbol{W}^\intercal \mathcal{E}(\boldsymbol{x}_{\bar{r}})\|_2^2 - \gamma\|\mathcal{E}(\boldsymbol{x}) - \mathcal{E}(\boldsymbol{x}^{(i)})\|. \tag{6}$$

Equation (6) is a supremum of $|\mathcal{X}|$ convex functions of $\boldsymbol{W}$, thus convex. Since $\mathcal{X}^{\mathcal{E}}$ is a discrete set consisting of a factorial number of points ($\Pi_{i \in [n]}\rho_i$), unlike the regression problem with continuous random variables in Chen and Paschalidis [2018], we may not simplify Equation (6) into a regularization form by leveraging convex conjugate functions because $\mathcal{X}^{\mathcal{E}}$ is non-convex and not equal to $\mathbb{R}^{\rho_{[n]}}$. Moreover, since changing the value of $x_j$ for some $j \in \bar{r}$ is equivalent to changing $\boldsymbol{W}^\intercal \mathcal{E}(\boldsymbol{x}_{\bar{r}})$ by a vector, unlike Li et al. [2022] where only a set of discrete labels rather than encodings are dealt with, there may not be a greedy algorithm based on sufficient statistics to find the optimal solution to Equation (6). In fact, let the norm be the $\ell_1$ norm, we can rewrite Equation (6) by fixing the values of $\|\mathcal{E}(\boldsymbol{x}) - \mathcal{E}(\boldsymbol{x}^{(i)})\|_1$:

$$\sup_{\boldsymbol{x} \in \mathcal{X}, 0 \leqslant k \leqslant \rho_{[n]}^+, \|\mathcal{E}(\boldsymbol{x}) - \mathcal{E}(\boldsymbol{x}^{(i)})\|_1 = k} \frac{1}{2}\|\mathcal{E}(\boldsymbol{x}_r) - \boldsymbol{W}^\intercal \mathcal{E}(\boldsymbol{x}_{\bar{r}})\|_2^2 - \gamma k. \tag{7}$$

If we fix $k$, Equation (7) is a generalization of the 0-1 quadratic programming problem, which can be transformed into a maximizing quadratic programming (MAXQP) problem. As a result, Equation (6) is an NP-hard problem with proof presented in Proposition 11 in appendix. Charikar and Wirth [2004] develop an algorithm to find an $\Omega(1/\log n)$ solution based on semi-definite programming (SDP) and sampling for the MAXQP problem. Instead of adopting a similar SDP algorithm with quadratic constraints, we propose a random and greedy algorithm to approximate the optimal solution, which is illustrated in Algorithm 1 in appendix, whose per-iteration time complexity is $\Theta(n^2 m\rho_{\max})$. It follows a simple idea that for a random node order $\boldsymbol{\pi}$, we select a partial optimal solution sequentially from $\pi_1$ to $\pi_n$. We enumerate the possible states of the first node to reduce uncertainty. In practice, we find that this algorithm always finds the exact solution that is NP-hard to find for random data with $n \leqslant 12$ and $\rho_{\max} \leqslant 5$ in most cases.

Since $\mathcal{X}^{\mathcal{E}}$ is non-convex and not equal to $\mathbb{R}^{\rho_{[n]}}$, using convex conjugate functions will not yield exact equivalence between Equation (5) and a regularized ERM problem. However, we can draw such a connection by imposing constraints on the dual variables as shown by the following proposition:

**Proposition 5** (Regularization Equivalence). *Let* $\ddot{\boldsymbol{W}} := [\boldsymbol{W}; -\boldsymbol{I}_{\rho_r}]^\intercal \in \mathbb{R}^{\rho_{[n]} \times \rho_r}$ *with* $\boldsymbol{W}_r = -\boldsymbol{I}_{\rho_r}$. *If* $\gamma \geqslant \rho_{[n]}\|\ddot{\boldsymbol{W}}\|_F^2$, *the Wasserstein DRO problem in Equation* (5) *is equivalent to*

$$\inf_{\boldsymbol{W}} \mathbb{E}_{\tilde{\mathbb{P}}_m} \frac{1}{2}\|\mathcal{E}(X_r) - \boldsymbol{W}^\intercal \mathcal{E}(\boldsymbol{X}_{\bar{r}})\|_2^2 + \varepsilon\rho_{[n]}\|\ddot{\boldsymbol{W}}\|_F^2,$$

*which subsumes a linear regression approach regularized by the Frobenius norm as a special case.*

This suggests that minimizing a regularized empirical risk may not be enough to achieve distributional robustness. Note that exact equivalence between DRO and regularized ERM in Chen and Paschalidis [2018] requires $\mathcal{X}^{\mathcal{E}} = \mathbb{R}^d$.

Now we perform non-asymptotic analysis on the proposed DRO estimator $\hat{\boldsymbol{W}}$. First, we would like to show that the solution to the Wasserstein DRO estimator in Equation (4) is unique so that we refer to an estimator unambiguously. Note that Equation (4) is a convex optimization problem but not necessarily strictly convex, and actually never convex in the high-dimensional setting. However, given

a sufficient number of samples, the problem becomes strictly convex and yields a unique solution with high probability. Second, we show that the correct skeleton $\mathcal{E}_{\text{skel}}$ can be recovered with high probability. This is achieved by showing that, for each node $X_r$, the estimator has zero weights for non-neighbor nodes $\mathbf{Co}_r$ and has non-zero weights for its neighbors $\mathbf{Ne}_r$ with high confidence. Before presenting the main results, we note that they are based on several important lemmas.

**Lemma 6.** *Suppose $\Xi$ is separable Banach space and fix $\mathbb{P}_0 \in \mathcal{P}(\Xi')$ for some $\Xi' \subseteq \Xi$. Suppose $c : \Xi \to \mathbb{R}_{\geqslant 0}$ is closed convex, $k$-positively homogeneous. Suppose $f : \Xi \to \mathcal{Y}$ is a mapping in the Lebesgue space of functions with finite first-order moment under $\mathbb{P}_0$ and upper semi-continuous with finite Lipschitz constant $lip_c(f)$. Then for all $\varepsilon \geqslant 0$, the following inequality holds with probability 1:*
$\sup_{\mathbb{Q} \in \mathcal{A}_\varepsilon^{W_p}(\mathbb{P}_0), \mathbb{Q} \in \mathcal{P}(\Xi')} \int f(\xi')\mathbb{Q}(\mathrm{d}\xi') \leqslant \varepsilon lip_c(f) + \int f(\xi')\mathbb{P}_0(\mathrm{d}\xi')$.

Lemma 6 follows directly from Cranko et al. [2021] and allows us to obtain an upper bound between the worst-case risk and empirical risk. It is crucial for the following finite-sample guarantees.

**Lemma 7.** *If Assumption 3 holds, for any $\mathbb{Q} \in \mathcal{A}_\varepsilon^{W_p}(\tilde{\mathbb{P}}_m)$, with high probability, $\boldsymbol{H}_{\mathcal{S}_r \mathcal{S}_r}^{\mathbb{Q}}$ is positive definite.*

**Lemma 8.** *If Assumption 3 and Assumption 4 hold, for any $\mathbb{Q} \in \mathcal{A}_\varepsilon^{W_p}(\tilde{\mathbb{P}}_m)$ and $\alpha \in (0, 1]$, with high probability,*

$$\|\boldsymbol{H}_{\mathcal{S}_r^c \mathcal{S}_r}^{\mathbb{Q}}(\boldsymbol{H}_{\mathcal{S}_r \mathcal{S}_r}^{\mathbb{Q}})^{-1}\|_{B,1,\infty} \leqslant 1 - \frac{\alpha}{2}.$$

The above two lemmas illustrate that Assumption 3 and Assumption 4 hold in the finite-sample setting. Let the estimated skeleton, neighbor nodes and the complement be $\hat{\mathcal{G}} := (\mathcal{V}, \hat{\mathcal{E}}_{\text{skel}})$, $\hat{\mathbf{Ne}}_r$ and $\hat{\mathbf{Co}}_r$ respectively. We derive the following guarantees for the proposed Wasserstein DRO estimator.

**Theorem 9.** *Given a Bayesian network $(\mathcal{G}, \mathbb{P})$ of $n$ categorical random variables and its skeleton $\mathcal{G}_{skel} := (\mathcal{V}, \mathcal{E}_{skel})$. Assume that the condition $\|\boldsymbol{W}^*\|_{B,2,1} \leqslant \bar{B}$ holds for some $\bar{B} > 0$ associated with an optimal Lagrange multiplier $\lambda_B^* > 0$ for $\boldsymbol{W}^*$ defined in Equation (1). Suppose that $\hat{\boldsymbol{W}}$ is a DRO risk minimizer of Equation (4) with a Wasserstein distance of order 1 and an ambiguity radius $\varepsilon = \varepsilon_0/m$ where $m$ is the number of samples drawn i.i.d. from $\mathbb{P}$. Under Assumptions 1, 2, 3, 4, if the number of samples satisfies*

$$m = \mathcal{O}\Big(\frac{C(\varepsilon_0 + \log(n/\delta) + \log \rho_{[n]})\sigma^2 \rho_{max}^4 \rho_{[n]}^3}{\min(\mu^2, 1)}\Big),$$

*where $C$ only depends on $\alpha$, $\Lambda$, and if the Lagrange multiplier satisfies*

$$\frac{32\mu\rho_{max}}{\alpha} < \lambda_B^* < \frac{\beta}{(\alpha/(4 - 2\alpha) + 2)\rho_{max}\sqrt{\rho_{[n]}}}\sqrt{\frac{\Lambda}{4}},$$

*then for any $\delta \in (0, 1]$, $r \in [n]$, with probability at least $1 - \delta$, the following properties hold:*

(a) *The optimal estimator $\hat{\boldsymbol{W}}$ is unique.*

(b) *All the non-neighbor nodes are excluded: $\boldsymbol{Co}_r \subseteq \hat{\boldsymbol{Co}}_r$.*

(c) *All the neighbor nodes are identified: $\boldsymbol{Ne}_r \subseteq \hat{\boldsymbol{Ne}}_r$.*

(d) *The true skeleton is successfully reconstructed: $\mathcal{G}_{skel} = \hat{\mathcal{G}}_{skel}$.*

*Proof sketch.* The main idea in the proof follows that in the lasso estimator [Wainwright, 2009]. Based on a primal-dual witness construction method and Lemma 8, it can be shown that if we control $\lambda_B^*$, a solution constrained to have zero weight for all the non-neighbor nodes is indeed optimal. Furthermore, Lemma 7 implies that there is a unique solution given information about the true neighbors. The uniqueness of the aforementioned optimal solution without knowing the true skeleton is then verified via convexity and a conjugate formulation of the block $\ell_{2,1}$ norm. Hereby we have shown that the optimal solution to Equation (4) is unique and excluding all the non-neighbor nodes. Next, we derive conditions on $\lambda_B^*$ for the estimation bias $\|\hat{\boldsymbol{W}} - \boldsymbol{W}^*\|_{B,2,\infty} < \beta/2$ to hold, which allows us to recover all the neighbor nodes. In such manner, applying the union bound over all the nodes $r \in [n]$ leads to successful exact skeleton discovery with high probability. $\square$

The results in Theorem 9 encompass some intuitive interpretations. Compared to Theorem 1 in Bank and Honorio [2020], we make more explicit the relationship among $m$, $\lambda_B^*$ and $\delta$. On one hand, the lower bound of $\lambda_B^*$ ensures that a sparse solution excluding non-neighbor nodes is obtained. A large error magnitude expectation $\mu$ therefore elicits stronger regularization. On the other hand, the upper bound $\lambda_B^*$ is imposed to guarantee that all the neighbor nodes are identified with less restriction on $\boldsymbol{W}$. There is naturally a trade-off when choosing $\bar{B}$ in order to learn the exact skeleton. The sample complexity depends on cardinalities $\rho_{[n]}$, confidence level $\delta$, the number of nodes $n$, the ambiguity level $\varepsilon_0$ and assumptions on errors. The dependence on $\sigma$ indicates that higher uncertainty caused by larger error norms demands more samples whereas the dependence on $\mu^{-2}$ results from the lower bound condition on $\lambda_B^*$ with respect to $\mu$. The ambiguity level is set to $\varepsilon_0/m$ based on the observation that obtaining more samples reduces ambiguity of the true distribution. In practice, we find that $\varepsilon_0$ is usually small thus negligible. Note that the sample complexity is polynomial in $n$. Furthermore, if we assume that the true graph has a bounded degree of $d$, we find that $m = \mathcal{O}(\frac{C(\varepsilon_0 + \log(n/\delta) + \log n + \log \rho_{\max})\sigma^2 \rho_{\max}^7 d^3}{\min(\mu^2, 1)})$ is logarithmic with respect to $n$, consistent with the results in Wainwright [2009].

We introduce constants $\bar{B}$ and $\lambda_B^*$ in order to find a condition for the statements in Theorem 9 to hold. If there exists a $\boldsymbol{W}$ incurring a finite loss, we can always find a solution $\hat{\boldsymbol{W}}$ and let $\bar{B} \triangleq \max_{\hat{\boldsymbol{W}}} \|\hat{\boldsymbol{W}}\|_{B,2,1}$ be the maximum norm of all solutions. Imposing $\|\boldsymbol{W}\|_{B,2,1} \leqslant \bar{B}$ is equivalent to the original problem. By Lagrange duality and similar argument for the lasso estimator, there exists a $\lambda_B^*$ that finds all the solutions with $\|\hat{\boldsymbol{W}}\|_{B,2,1} = \bar{B}$. Therefore we have a mapping between $\varepsilon$ and $\lambda_B^*$.

### 3.3 Kullback-Leibler DRO

In addition to optimal transport, $\phi$-divergence is also widely used to construct an ambiguity set for DRO problems. We consider the following definition of a special $\phi$-divergence called the KL divergence: $D(\mathbb{Q} \| \mathbb{P}) \coloneqq \int_{\mathcal{X}} \ln \frac{\mathbb{Q}(\mathrm{d}\boldsymbol{x})}{\mathbb{P}(\mathrm{d}\boldsymbol{x})} \mathbb{Q}(\mathrm{d}\boldsymbol{x})$, where $\mathbb{Q} \in \mathcal{P}(\mathcal{X})$ is absolutely continuous with respect to $\mathbb{P} \in \mathcal{P}(\mathcal{X})$ and $\frac{\mathbb{Q}(\mathrm{d}\boldsymbol{x})}{\mathbb{P}(\mathrm{d}\boldsymbol{x})}$ denotes the Radon-Nikodym derivative. A noteworthy property of ambiguity sets based on the KL divergence is absolute continuity of all the candidate distributions with respect to the empirical distribution. It implies that if the true distribution is an absolutely continuous probability distribution, the ambiguity set will never include it. In fact, any other point outside the support of the nominal distribution remains to have zero probability. Unlike the Wasserstein metric, the KL divergence does not measure some closeness between two points, nor does it have some measure concentration results. However, we argue that adopting the KL divergence may bring advantages over the Wasserstein distance since the Bayesian network distribution we study is a discrete distribution over purely categorical random variables. Moreover, as illustrated below, adopting the KL divergence leads to better computational efficiency.

Let $\mathcal{A} \triangleq \mathcal{A}_\varepsilon^D(\tilde{\mathbb{P}}_m)$ be the ambiguity set, the dual formulation of Equation (3) follows directly from Theorem 4 in Hu and Hong [2013]:

$$\inf_{\boldsymbol{W}, \gamma > 0} \gamma \ln \big[ \frac{1}{m} \sum_{i \in [m]} e^{\frac{1}{2} \|\mathcal{E}(x_r^{(i)}) - \boldsymbol{W}^\intercal \mathcal{E}(\boldsymbol{x}_{\bar{r}}^{(i)})\|_2^2 / \gamma} \big] + \gamma \varepsilon,$$

which directly minimizes a convex objective. In contrast to the approximate Wasserstein estimator, this KL DRO estimator finds the exact solution to the primal problem by strong duality.

The worst-case risk over a KL divergence ball can be bounded by variance [Lam, 2019], similar to Lipschitz regularization in Lemma 6. Based on this observation, we derive the following results:

**Theorem 10.** *Suppose that $\hat{\boldsymbol{W}}$ is a DRO risk minimizer of Equation* (4) *with the KL divergence and an ambiguity radius $\varepsilon = \varepsilon_0/m$. Given the same definitions of $(\mathcal{G}, \mathbb{P})$, $\mathcal{G}_{skel}$, $\bar{B}$, $\lambda_B^*$, $m$ in Theorem 9. Under Assumptions 1, 2, 3, 4, if the number of samples satisfies*

$$m = \mathcal{O}\big( \frac{C(\varepsilon_0 + \log(n/\delta) + \log \rho_{[n]})\sigma^2 \rho_{max}^4 \rho_{[n]}^3}{\min(\mu^2, 1)} \big).$$

*where $C$ depends on $\alpha$, $\Lambda$ while independent of $n$, and if the Lagrange multiplier satisfies the same condition as in Theorem 9, then for any $\delta \in (0, 1]$, $r \in [n]$, with probability at least $1 - \delta$, the properties (a)-(d) in Theorem 9 hold.*

The sample complexities in Theorem 9 and Theorem 10 differ in the constant $C$ due to the difference between the two probability metrics. Note that $C$ is independent of $n$ in both methods. The dependency on $1/(\lambda_B^*)^2$ is absorbed in the denominator because we require that $\lambda_B^* - 16\mu\rho_{\max}/\alpha > 0$. The sample complexities provide a perspective of our confidence on upper bounding the true risk in terms of the ambiguity radius. $\varepsilon_0$ serves as our initial guess on distributional uncertainty and increases the sample complexity only slightly because it is usually dominated by other terms in practice: $\varepsilon \ll \log(n/\delta)$. Even though the samples are drawn from an adversarial distribution with a proportion of noises, the proposed methods may still succeed as long as the true distribution can be made close to an upper confidence bound.

## 4 Experiments

We conduct experiments[3] on benchmark datasets [Scutari, 2010] and real-world datasets [Malone et al., 2015] perturbed by the following contamination models:

- **Noisefree model.** This is the baseline model without any noises.
- **Huber's contamination model.** In this model, each sample has a fixed probability of $\zeta$ to be replaced by a sample drawn from an arbitrary distribution.
- **Independent failure model.** Each entry of a sample is independently corrupted with probability $\zeta$.

We conduct all experiments on a laptop with an Intel Core i7 2.7 GHz processor. We adopt the proposed approaches based on Wasserstein DRO and KL DRO, the group norm regularization method [Bank and Honorio, 2020], the MMPC algorithm [Tsamardinos et al., 2006] and the GRaSP algorithm [Lam et al., 2022] for skeleton learning. Based on the learned skeletons, we infer a DAG with the hill-climbing (HC) algorithm [Tsamardinos et al., 2006]. For the Wasserstein-based method, we leverage Adam [Kingma and Ba, 2014] to optimize the overall objective with $\beta_1 = 0.9$, $\beta_2 = 0.990$, a learning rate of $1.0$, a batch size of $500$, a maximum of $200$ iterations for optimization and $10$ iterations for approximating the worst-case distribution. For the KL-based and standard regularization methods, we use the L-BFGS-B [Byrd et al., 1995] optimization method with default parameters. We set the cardinality of the maximum conditional set to $3$ in MMPC. The Bayesian information criterion (BIC) [Neath and Cavanaugh, 2012] score is adopted in the HC algorithm. A random mixture of $20$ random Bayesian networks serves as the adversarial distribution for both contamination models. All hyper-parameters are chosen based on the best performance on random Bayesian networks with the same size as the input one. Each experimental result is taken as an average over $10$ independent runs. When dealing with real-world datasets, we randomly split the data into two halves for training and testing.

We use the F1-score, or the Dice coefficient (regarding the label of each edge indicating its presence as a binary random variable and considering all possible edges), to evaluate performance on benchmark datasets and BIC for real-world datasets. The results are reported in Table 1 and more results can be found in Table 2 in appendix. We observe that in most cases the proposed DRO methods are comparable to MMPC and MMHC, which are generally the best-performing methods in Bank and Honorio [2020]. We illustrate in Figure 1 the results on `earthquake` by varying the number of samples, corruption level and ambiguity radius or regularization coefficient. Figure 1 (a) suggests that all the methods perfectly recover the true skeleton given more than $2,000$ samples. The results in Figure 1 (b-c) indicate that, in highly uncertain settings, Wasserstein DRO as well as KL DRO is superior to other approaches. Meanwhile, Table 1 and Figure 1 (a) suggest that the DRO methods and the regularized ERM approach are comparable to MMPC and GRaSP when clean data is given. The sensitivity analysis (Figure 1 (d)) suggests a trade-off between robustness and target performance (F1-score in our case). All the approaches have similar execution time except that Wasserstein DRO is several times slower due to the combinatorial sub-problem of computing the worst-case distribution.

## 5 Discussion and Conclusion

In this paper, we put forward a distributionally robust optimization method to recover the skeleton of a general discrete Bayesian network. We discussed two specific probability metrics, developed

---

[3]Our code is publicly available at `https://github.com/DanielLeee/drslbn`.

Table 1: Comparisons of F1 scores for benchmark datasets and BIC for real-world datasets (backache, voting). BIC is not applicable to skeletons. The best and runner-up results are marked in bold. Significant differences are marked by † (paired t-test, $p < 0.05$).

| Dataset | n | m | Noise | ζ | Wass | KL | Reg | MMPC | GRASP | Wass+HC | KL+HC | Reg+HC | MMPC+HC | GRASP+HC | HC |
|---|---|---|---|---|---|---|---|---|---|---|---|---|---|---|---|
| asia | 8 | 1000 | Noisefree | 0 | 0.7800† | 0.7285† | 0.7897† | **0.9067** | **0.8167** | 0.5123 | 0.6367 | 0.5743 | **0.6667** | **0.6583** | 0.6550 |
| asia | 8 | 1000 | Huber | 0.2 | **0.7333†** | 0.7124† | **0.7297†** | 0.5468 | 0.6570 | **0.3943** | **0.3724** | 0.3487 | 0.2907 | 0.3664 | 0.2183 |
| asia | 8 | 1000 | Independent | 0.2 | **0.6933** | 0.6797 | 0.6868 | 0.6359 | 0.3632† | **0.2676** | **0.2632** | 0.2581 | 0.2469 | 0.1794 | 0.2443 |
| cancer | 5 | 1000 | Noisefree | 0 | **1.0000†** | **1.0000†** | **1.0000†** | 0.6133 | 0.6133 | 0.2800 | 0.2800 | 0.2800 | 0.2800 | 0.2800 | 0.2800 |
| cancer | 5 | 1000 | Huber | 0.5 | **0.9156†** | 0.8933† | **0.9092†** | 0.6133 | 0.5357 | **0.4333** | 0.3833 | **0.4143** | 0.2589 | 0.2714 | 0.2589 |
| cancer | 5 | 1000 | Independent | 0.2 | **0.9048†** | **0.9029†** | 0.8992† | 0.0000 | 0.0000 | 0.0000 | 0.0000 | 0.0000 | 0.0000 | 0.0000 | 0.0000 |
| earthquake | 5 | 1000 | Noisefree | 0 | 0.8447† | 0.9333† | 0.9778 | **1.0000** | 0.9778 | 0.2500 | 0.2500 | 0.2500 | 0.2500 | 0.2500 | 0.2278† |
| earthquake | 5 | 1000 | Huber | 0.2 | **0.7509†** | **0.7509†** | **0.7509†** | 0.5978 | 0.6583† | 0.4618 | 0.4618 | 0.4618 | 0.3860 | 0.4547 | 0.3860 |
| earthquake | 5 | 1000 | Independent | 0.2 | **0.6786†** | **0.6350†** | **0.6350†** | 0.0000 | 0.0000 | 0.0000 | 0.0000 | 0.0000 | 0.0000 | 0.0000 | 0.0000 |
| sachs | 11 | 1000 | Noisefree | 0 | 0.8357† | **0.8402†** | 0.8374† | **0.9697** | 0.7678† | 0.4310† | 0.4535† | 0.4641† | **0.5935** | 0.4112† | **0.5873** |
| sachs | 11 | 1000 | Huber | 0.2 | 0.7765 | **0.8064** | 0.7893 | 0.7498 | 0.5663† | **0.5194** | 0.4815 | 0.4520 | 0.4736 | 0.2380 | **0.5028** |
| sachs | 11 | 1000 | Independent | 0.5 | **0.5268†** | **0.5208†** | 0.5172† | 0.0000 | 0.0000 | 0.0000 | 0.0000 | 0.0000 | 0.0000 | 0.0000 | 0.0000 |
| survey | 6 | 1000 | Noisefree | 0 | **0.6596** | **0.6545** | 0.6506 | 0.6533 | 0.1714† | **0.1789** | **0.1789** | **0.1789** | **0.1789** | 0.0571 | **0.1789** |
| survey | 6 | 1000 | Huber | 0.2 | **0.7303†** | 0.6778† | **0.7095†** | 0.5396 | 0.3810 | **0.1444** | **0.1444** | **0.1444** | **0.1444** | 0.1516 | **0.1444** |
| survey | 6 | 1000 | Independent | 0.2 | **0.6311†** | **0.6705†** | 0.6220† | 0.2032 | 0.0000† | **0.1071** | **0.1071** | **0.1071** | **0.1071** | 0.0000 | **0.1071** |
| alarm | 37 | 1000 | Noisefree | 0 | 0.4750† | 0.7863† | **0.8042†** | **0.8530** | 0.6824† | 0.3483† | 0.4949† | 0.4470† | **0.5635** | **0.4976** | 0.4494† |
| alarm | 37 | 1000 | Huber | 0.2 | 0.1432† | 0.1619† | **0.6571†** | 0.5486 | 0.1945† | 0.2192 | 0.1680† | **0.3148** | **0.2774** | 0.2092† | 0.2582 |
| alarm | 37 | 1000 | Independent | 0.2 | 0.1419† | 0.1448† | **0.5458†** | **0.4309** | 0.2830† | 0.0000 | 0.0000 | 0.0000 | 0.0000 | 0.0000 | 0.0000 |
| voting | 17 | 216 | Noisefree | 0 | N/A | N/A | N/A | N/A | N/A | −2451.8631 | −2453.2737 | −2453.4091 | −2475.5799 | −2482.3835 | −2456.1489 |
| voting | 17 | 216 | Huber | 0.2 | N/A | N/A | N/A | N/A | N/A | −4418.9731 | −4418.9731 | −4487.4544 | −4450.3941 | −4445.0175 | −4418.9731 |
| voting | 17 | 216 | Independent | 0.2 | N/A | N/A | N/A | N/A | N/A | −4453.8298 | −4453.8298 | −4522.5521 | −4465.1076 | −4473.8612 | −4453.8298 |
| backache | 32 | 90 | Noisefree | 0 | N/A | N/A | N/A | N/A | N/A | −1729.8364 | −1726.8465 | **−1710.7248** | −1719.5002 | **−1713.7583** | −1729.7991 |
| backache | 32 | 90 | Huber | 0.2 | N/A | N/A | N/A | N/A | N/A | −3186.5001 | −3186.5001 | −3186.5001 | −3186.5001 | −3186.5001 | −3186.5001 |
| backache | 32 | 90 | Independent | 0.2 | N/A | N/A | N/A | N/A | N/A | −2800.9386 | −2800.9386 | −2800.9386 | −2800.9386 | −2800.9386 | −2800.9386 |

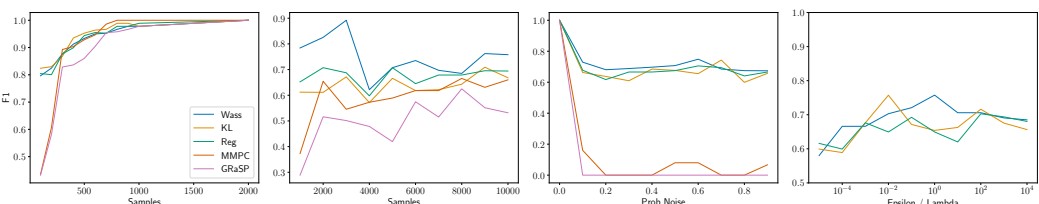

Figure 1: Skeleton estimation results in F1-score on the `earthquake` dataset for Wasserstein DRO, KL DRO, regularized linear regression, MMPC and GRaSP. From left to right: (a) noisefree, varying number of samples; (b) Huber with $\zeta = 0.5$ noisy data, varying number of samples; (c) independent failure, varying $\zeta$, the probability of noise; (d) independent failure with $\zeta = 0.2$, varying $\varepsilon$ the ambiguity radius, or $\tilde{\lambda}$ the regularization coefficient.

tractable algorithms to compute the estimators. We established the connection between the proposed method and regularization. We derived non-asymptotic bounds polynomial in the number of nodes for successful identification of the true skeleton. The sample complexities become logarithmic for bounded-degree graphs. Empirical results showcased the effectiveness of our methods.

The strength of making no distributional assumptions in our methods inherits from the regularized regression baseline, which is shown to be a special case of DRO. Besides the original benefits in Bank and Honorio [2020], our methods are explicitly robust and able to adjust $\varepsilon$ to incorporate our uncertainty about the data generating mechanism.

Since we do not make any specific assumptions on the conditional probability distributions or on the corruption models, our methods may not be superior to the approaches proposed to tackle certain family of noises or parametric distributions. In addition, making assumptions such as an ordinal relationship, continuous values or a Poisson structural equation model (SEM) may lead to more efficient algorithms and tighter bounds. In some cases, adopting an identity encoding mapping is sufficient to learn a continuous Bayesian network [Aragam et al., 2015]. Furthermore, it would be interesting to incorporate prior distributional information into design of the ambiguity set for better performance. Another important topic is whether the underlying structure is identifiable in a robust manner [Sankararaman et al., 2022]. Formulating the complete DAG learning problem as one optimization problem may lead to a non-convex problem. Nonetheless, leveraging an adversarial training approach in continuous optimization for Gaussian Bayesian networks and causal discovery is a promising future direction to pursue.

## Acknowledgments and Disclosure of Funding

This material is based upon work supported by the National Science Foundation under Grant No. 1652530.

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

# A Algorithms

The pseudo-code of the greedy algorithm for solving Equation (6) in Wasserstein DRO is illustrated in Algorithm 1.

---

**Algorithm 1** Greedy Algorithm for the Wasserstein Worst-case Risk

---

**Input:** $\boldsymbol{W}, \gamma, \boldsymbol{x}^{(i)}$
**Output:** a solution $\hat{\boldsymbol{x}}$ to Equation (6)
Initialize $\hat{\boldsymbol{x}} = \boldsymbol{x}^{(i)}$
**for all** $(j, x_j^t) \in [n] \times \mathcal{C}_j$ **do**
    Get a random permutation $\boldsymbol{\pi}$ over $[n]$ with $\pi_1 = j$
    **for** $k := 2$ **to** $n$ **do**
        $x_{\pi_j}^t \leftarrow \arg\sup_{x_{\pi_k}^t} \ell_{\boldsymbol{W}}(\boldsymbol{x}_{\boldsymbol{\pi}_{[k]}}^t) - \gamma\|\mathcal{E}(\boldsymbol{x}_{\boldsymbol{\pi}_{[k]}}^t) - \mathcal{E}(\boldsymbol{x}_{\boldsymbol{\pi}_{[k]}}^{(i)})\|$
    **end for**
    **if** $\boldsymbol{x}^t$ yields a greater objective than $\hat{\boldsymbol{x}}$ **then**
        $\hat{\boldsymbol{x}} \leftarrow \boldsymbol{x}^t$
    **end if**
**end for**

---

# B Optimization Details

Define

$$\ell_{\boldsymbol{W}}(\boldsymbol{X}) := \frac{1}{2}\|\mathcal{E}(X_r) - \boldsymbol{W}^\mathsf{T}\mathcal{E}(\boldsymbol{X}_{\bar{r}})\|_2^2.$$

The Lagrangian dual problem of the Wasserstein DRO problem is

$$\inf_{\boldsymbol{W},\gamma\geqslant 0} f(\boldsymbol{W}, \gamma) := \gamma\varepsilon + \frac{1}{m}\sum_{i=1}^m \sup_{\boldsymbol{x}\in\mathcal{X}} \ell_{\boldsymbol{W}}(\boldsymbol{x}) - \gamma\|\mathcal{E}(\boldsymbol{x}) - \mathcal{E}(\boldsymbol{x}^{(i)})\|_1.$$

One of its sub-gradients can be computed as

$$\frac{1}{m}\sum_{i=1}^m \mathcal{E}(\hat{\boldsymbol{x}}_{\bar{r}}^{(i)})\mathcal{E}(\hat{\boldsymbol{x}}_{\bar{r}}^{(i)})^\mathsf{T}\boldsymbol{W} - \mathcal{E}(\hat{\boldsymbol{x}}_{\bar{r}}^{(i)})\mathcal{E}(\hat{\boldsymbol{x}}_r^{(i)})^\mathsf{T} \in \frac{\partial}{\partial\boldsymbol{W}}f$$

$$\varepsilon - \frac{1}{m}\sum_{i=1}^m \|\mathcal{E}(\hat{\boldsymbol{x}}^{(i)}) - \mathcal{E}(\boldsymbol{x}^{(i)})\|_1 \in \frac{\partial}{\partial\gamma}f.$$

For the DRO problem based on the KL divergence:

$$\inf_{\boldsymbol{W},\gamma>0} f(\boldsymbol{W}, \gamma) := \gamma\ln\big[\frac{1}{m}\sum_{i\in[m]} e^{\ell_{\boldsymbol{W}}(\boldsymbol{x}^{(i)})/\gamma}\big] + \gamma\varepsilon,$$

a sub-gradient of which can be computed as

$$\frac{\sum_{i\in[m]} e^{\ell_{\boldsymbol{W}}(\boldsymbol{x}^{(i)})/\gamma}(\mathcal{E}(\boldsymbol{x}_{\bar{r}}^{(i)})\mathcal{E}(\boldsymbol{x}_{\bar{r}}^{(i)})^\mathsf{T}\boldsymbol{W} - \mathcal{E}(\boldsymbol{x}_{\bar{r}}^{(i)})\mathcal{E}(\boldsymbol{x}_r^{(i)})^\mathsf{T})}{\sum_{i\in[m]} e^{\ell_{\boldsymbol{W}}(\boldsymbol{x}^{(i)})/\gamma}} \in \frac{\partial}{\partial\boldsymbol{W}}f$$

$$\ln\big(\frac{1}{m}\sum_{i\in[m]} e^{\ell_{\boldsymbol{W}}(\boldsymbol{x}^{(i)})/\gamma}\big) - \frac{\sum_{i\in[m]} e^{\ell_{\boldsymbol{W}}(\boldsymbol{x}^{(i)})/\gamma}\cdot \ell_{\boldsymbol{W}}(\boldsymbol{x}^{(i)})}{\gamma\sum_{i\in[m]} e^{\ell_{\boldsymbol{W}}(\boldsymbol{x}^{(i)})/\gamma}} + \epsilon \in \frac{\partial}{\partial\gamma}f.$$

# C Technical Proofs

**Proposition 11** (NP-hardness of Wasserstein DRO Supremum). *The problem in Equation* (6) *is NP-hard.*

*Proof.* Recall the MAXQP problem:

$$\sum_{i,j=1}^{n} a_{ij} x_i x_j, \quad \text{s.t. } x_i \in \{-1, 1\} \, \forall i.$$

In Equation (6), let $\gamma = 0$, $\mathcal{E}(x_r) = 0$, $\boldsymbol{x}_{\bar{r}}$ correspond to $n$ binary variables taking values in {-1, 1} and $\mathcal{E}(\boldsymbol{x}_{\bar{r}}) = \boldsymbol{x}_{\bar{r}}$. Let $\boldsymbol{W} \in \mathbb{R}^{n \times n^2}$. For all $i, j \in [n]$, let $k = (i-1)n + j$. The $k$-the column of $\boldsymbol{W}$ satisfies $W_{ik} = 1$, $W_{jk} = a_{ij}/2$ and 0 for the other elements. We have obtained a polynomial-time reduction from an NP-hard problem to Equation (6). $\qquad \square$

**Proposition 5** (Regularization Equivalence). *Let $\ddot{\boldsymbol{W}} := [\boldsymbol{W}; -\boldsymbol{I}_{\rho_r}]^{\mathsf{T}} \in \mathbb{R}^{\rho_{[n]} \times \rho_r}$ with $\boldsymbol{W}_r = -\boldsymbol{I}_{\rho_r}$. If $\gamma \geqslant \rho_{[n]} \|\ddot{\boldsymbol{W}}\|_F^2$, the Wasserstein distributionally robust regression problem in Equation (5) is equivalent to*

$$\inf_{\boldsymbol{W}} \mathbb{E}_{\tilde{\mathbb{P}}_m} \frac{1}{2} \|\mathcal{E}(X_r) - \boldsymbol{W}^{\mathsf{T}}\mathcal{E}(\boldsymbol{X}_{\bar{r}})\|_2^2 + \varepsilon \rho_{[n]} \|\ddot{\boldsymbol{W}}\|_F^2,$$

*which subsumes a linear regression approach regularized by the Frobenius norm as a special case.*

*Proof.* Recapitulating on Equation (6):

$$\sup_{\boldsymbol{x} \in \mathcal{X}} \frac{1}{2} \|\mathcal{E}(x_r) - \boldsymbol{W}^{\mathsf{T}}\mathcal{E}(\boldsymbol{x}_{\bar{r}})\|_2^2 - \gamma \|\mathcal{E}(\boldsymbol{x}) - \mathcal{E}(\boldsymbol{x}^{(i)})\|_1.$$

Observe that

$$\begin{aligned}
\|\mathcal{E}(x_r) - \boldsymbol{W}^{\mathsf{T}}\mathcal{E}(\boldsymbol{x}_{\bar{r}})\|_2^2 &\triangleq \|\ddot{\boldsymbol{W}}^{\mathsf{T}}\mathcal{E}(\boldsymbol{x}_{[n]})\|_2^2 \\
&\leqslant \|\|\ddot{\boldsymbol{W}}^{\mathsf{T}}\|\|_{\infty,2}^2 \\
&\leqslant \|\ddot{\boldsymbol{W}}\|_{1,2}^2 \\
&\leqslant \rho_{[n]} \|\ddot{\boldsymbol{W}}\|_F^2 \\
&\leqslant \gamma.
\end{aligned}$$

Therefore, for any $\boldsymbol{x} \neq \boldsymbol{x}^{(i)}$,

$$\frac{1}{2}\|\mathcal{E}(x_r) - \boldsymbol{W}^{\mathsf{T}}\mathcal{E}(\boldsymbol{x}_{\bar{r}})\|_2^2 - \gamma \|\mathcal{E}(\boldsymbol{x}) - \mathcal{E}(\boldsymbol{x}^{(i)})\|_1 - (\frac{1}{2}\|\mathcal{E}(x_r^{(i)}) - \boldsymbol{W}^{\mathsf{T}}\mathcal{E}(\boldsymbol{x}_{\bar{r}}^{(i)})\|_2^2 - \gamma \|\mathcal{E}(\boldsymbol{x}^{(i)}) - \mathcal{E}(\boldsymbol{x}^{(i)})\|_1)$$

$$\leqslant \frac{1}{2}(\|\mathcal{E}(x_r) - \boldsymbol{W}^{\mathsf{T}}\mathcal{E}(\boldsymbol{x}_{\bar{r}})\|_2^2 - \|\mathcal{E}(x_r^{(i)}) - \boldsymbol{W}^{\mathsf{T}}\mathcal{E}(\boldsymbol{x}_{\bar{r}}^{(i)})\|_2^2) - \gamma \|\mathcal{E}(\boldsymbol{x}) - \mathcal{E}(\boldsymbol{x}^{(i)})\|_1$$

$$\leqslant \frac{1}{2}(2\gamma) - \gamma \|\mathcal{E}(\boldsymbol{x}) - \mathcal{E}(\boldsymbol{x}^{(i)})\|_1$$

$$\leqslant \gamma - \gamma$$

$$= 0,$$

which implies that the supremum can always be achieved at $\boldsymbol{x} = \boldsymbol{x}^{(i)}$. Minimizing over $\gamma$ leads to

$$\inf_{\boldsymbol{W}} \mathbb{E}_{\tilde{\mathbb{P}}_m} \frac{1}{2} \|\mathcal{E}(X_r) - \boldsymbol{W}^{\mathsf{T}}\mathcal{E}(\boldsymbol{X}_{\bar{r}})\|_2^2 + \varepsilon \rho_{[n]} \|\ddot{\boldsymbol{W}}\|_F^2.$$

$$\square$$

**Lemma 6.** *Suppose $\Xi$ is separable Banach space and fix $\mathbb{P}_0 \in \mathcal{P}(\Xi')$ for some $\Xi' \subseteq \Xi$. Suppose $c : \Xi \to \mathbb{R}_{\geqslant 0}$ is closed convex, $k$-positively homogeneous. Suppose $f : \Xi \to \mathcal{Y}$ is a mapping in the Lebesgue space of functions with finite first-order moment under $\mathbb{P}_0$ and upper semi-continuous with finite Lipschitz constant $lip_c(f)$. Then for all $\varepsilon \geqslant 0$, the following inequality holds with probability 1:*

$$\sup_{\mathbb{Q} \in \mathcal{A}_\varepsilon^{W_p}(\mathbb{P}_0), \mathbb{Q} \in \mathcal{P}(\Xi')} \int f(\xi') \mathbb{Q}(\mathrm{d}\xi') \leqslant \varepsilon lip_c(f) + \int f(\xi') \mathbb{P}_0(\mathrm{d}\xi').$$

*Proof.* The result follows directly from Theorem 1 in Cranko et al. [2021]:

$$\sup_{\mathbb{Q}\in\mathcal{A}_\varepsilon^{W_p}(\mathbb{P}_0),\mathbb{Q}\in\mathcal{P}(\Xi)}\int f(\xi)\mathbb{Q}(\mathrm{d}\xi)\leqslant\varepsilon\mathrm{lip}_c(f)+\int f(\xi')\mathbb{P}_0(\mathrm{d}\xi').$$

Since $\Xi'\subseteq\Xi$, observe

$$\sup_{\mathbb{Q}\in\mathcal{A}_\varepsilon^{W_p}(\mathbb{P}_0),\mathbb{Q}\in\mathcal{P}(\Xi')}\int f(\xi')\mathbb{Q}(\mathrm{d}\xi')\leqslant\sup_{\mathbb{Q}\in\mathcal{A}_\varepsilon^{W_p}(\mathbb{P}_0),\mathbb{Q}\in\mathcal{P}(\Xi)}\int f(\xi)\mathbb{Q}(\mathrm{d}\xi).$$

$\square$

**Lemma 7.** *If Assumption 3 holds, for any* $\mathbb{Q}\in\mathcal{A}_\varepsilon^{W_p}(\tilde{\mathbb{P}}_m)$, *with probability at least* $1-2|\mathcal{S}_r|^2\exp\left(-\frac{mt^2}{2|\mathcal{S}_r|^2}\right)$, *we have*

$$\Lambda_{min}(\boldsymbol{H}_{\mathcal{S}_r\mathcal{S}_r}^{\mathbb{Q}})\geqslant\Lambda_{min}(\boldsymbol{H}_{\mathcal{S}_r\mathcal{S}_r})-4\varepsilon|\mathcal{S}_r|^{\frac{1}{2}}-t.$$

*Proof.* The minimum eigenvalue of the true covariance matrix $\boldsymbol{H}_{\mathcal{S}_r\mathcal{S}_r}$ satisfies:

$$\begin{aligned}\Lambda_{\min}(\boldsymbol{H}_{\mathcal{S}_r\mathcal{S}_r})&\triangleq\min_{\|\boldsymbol{v}\|_2=1}\boldsymbol{v}^\intercal\boldsymbol{H}_{\mathcal{S}_r\mathcal{S}_r}\boldsymbol{v}\\&=\min_{\|\boldsymbol{v}\|_2=1}\boldsymbol{v}^\intercal\boldsymbol{H}_{\mathcal{S}_r\mathcal{S}_r}^{\mathbb{Q}}\boldsymbol{v}+\boldsymbol{v}^\intercal(\tilde{\boldsymbol{H}}_{\mathcal{S}_r\mathcal{S}_r}-\boldsymbol{H}_{\mathcal{S}_r\mathcal{S}_r}^{\mathbb{Q}})\boldsymbol{v}+\boldsymbol{v}^\intercal(\boldsymbol{H}_{\mathcal{S}_r\mathcal{S}_r}-\tilde{\boldsymbol{H}}_{\mathcal{S}_r\mathcal{S}_r})\boldsymbol{v}\\&\leqslant\Lambda_{\min}(\boldsymbol{H}_{\mathcal{S}_r\mathcal{S}_r}^{\mathbb{Q}})+\boldsymbol{u}^\intercal(\tilde{\boldsymbol{H}}_{\mathcal{S}_r\mathcal{S}_r}-\boldsymbol{H}_{\mathcal{S}_r\mathcal{S}_r}^{\mathbb{Q}})\boldsymbol{u}+\boldsymbol{u}^\intercal(\boldsymbol{H}_{\mathcal{S}_r\mathcal{S}_r}-\tilde{\boldsymbol{H}}_{\mathcal{S}_r\mathcal{S}_r})\boldsymbol{u},\end{aligned}$$

where $\|\boldsymbol{u}\|_2=1$ is an eigenvector of $\boldsymbol{H}_{\mathcal{S}_r\mathcal{S}_r}^{\mathbb{Q}}$ with minimum eigenvalue.

Therefore, $\Lambda_{\min}(\boldsymbol{H}_{\mathcal{S}_r\mathcal{S}_r}^{\mathbb{Q}})$ can be lower bounded as follows:

$$\begin{aligned}\Lambda_{\min}(\boldsymbol{H}_{\mathcal{S}_r\mathcal{S}_r}^{\mathbb{Q}})&\geqslant\Lambda_{\min}(\boldsymbol{H}_{\mathcal{S}_r\mathcal{S}_r})-\boldsymbol{u}^\intercal(\tilde{\boldsymbol{H}}_{\mathcal{S}_r\mathcal{S}_r}-\boldsymbol{H}_{\mathcal{S}_r\mathcal{S}_r}^{\mathbb{Q}})\boldsymbol{u}-\boldsymbol{u}^\intercal(\boldsymbol{H}_{\mathcal{S}_r\mathcal{S}_r}-\tilde{\boldsymbol{H}}_{\mathcal{S}_r\mathcal{S}_r})\boldsymbol{u}\\&\geqslant\Lambda_{\min}(\boldsymbol{H}_{\mathcal{S}_r\mathcal{S}_r})-|\boldsymbol{u}^\intercal(\tilde{\boldsymbol{H}}_{\mathcal{S}_r\mathcal{S}_r}-\boldsymbol{H}_{\mathcal{S}_r\mathcal{S}_r}^{\mathbb{Q}})\boldsymbol{u}|-\|(\boldsymbol{H}_{\mathcal{S}_r\mathcal{S}_r}-\tilde{\boldsymbol{H}}_{\mathcal{S}_r\mathcal{S}_r})\|_F,\end{aligned}$$

due to the fact that

$$\boldsymbol{u}^\intercal\boldsymbol{H}\boldsymbol{u}\leqslant\Lambda_{\max}(\boldsymbol{H})\leqslant\sqrt{\sum_i(\Lambda_i(\boldsymbol{H}))^2}\leqslant\|\boldsymbol{H}\|_{2,2}.$$

We can obtain an upper bound on $|\boldsymbol{u}^\intercal(\tilde{\boldsymbol{H}}_{\mathcal{S}_r\mathcal{S}_r}-\boldsymbol{H}_{\mathcal{S}_r\mathcal{S}_r}^{\mathbb{Q}})\boldsymbol{u}|$ based on Lemma 6:

$$|\boldsymbol{u}^\intercal(\tilde{\boldsymbol{H}}_{\mathcal{S}_r\mathcal{S}_r}-\boldsymbol{H}_{\mathcal{S}_r\mathcal{S}_r}^{\mathbb{Q}})\boldsymbol{u}|\leqslant4|\mathcal{S}_r|^{\frac{1}{2}}\varepsilon,$$

because for function $g(\mathcal{E}(\boldsymbol{x})):=\boldsymbol{u}^\intercal\boldsymbol{H}_{\mathcal{S}_r\mathcal{S}_r}\boldsymbol{u}$, it can be shown that for any $\|\mathcal{E}(\boldsymbol{x})-\mathcal{E}(\boldsymbol{x}')\|_1=k$ and some $|\mathcal{S}|=k$,

$$|g(\mathcal{E}(\boldsymbol{x}))-g(\mathcal{E}(\boldsymbol{x}'))|\leqslant\sum_{k\in\mathcal{S}}\sum_{i\in\mathcal{S}_r}|H_{ik}-H'_{ik}|u_iu_k+|H_{ki}-H'_{ki}|u_ku_i\leqslant4k|\mathcal{S}_r|^{\frac{1}{2}}.$$

Recall that we assume that the encoding schemes take values in $\mathcal{B}=\{-1,0,1\}$. Therefore $\mathrm{lip}_c(g)=4|\mathcal{S}_r|^{\frac{1}{2}}$.

We derive an upper bound of $\|(\boldsymbol{H}_{\mathcal{S}_r\mathcal{S}_r}-\tilde{\boldsymbol{H}}_{\mathcal{S}_r\mathcal{S}_r})\|_F$ as follows. Consider a random variable and its expectation

$$Z_{ij}:=(\tilde{\boldsymbol{H}}_{\mathcal{S}_r\mathcal{S}_r})_{ij}=\frac{1}{m}\sum_{l=1}^m\mathcal{E}(\boldsymbol{x}_{\tilde{r}}^{(l)})_i\mathcal{E}(\boldsymbol{x}_{\tilde{r}}^{(l)})_j\in[-1/m,1/m]$$

$$\mathbb{E}_\mathbb{P}Z_{ij}=(\boldsymbol{H}_{\mathcal{S}_r\mathcal{S}_r})_{ij}.$$

By Hoeffding's inequality, we observe

$$\mathrm{Prob}(|(\tilde{\boldsymbol{H}}_{\mathcal{S}_r\mathcal{S}_r})_{ij}-(\boldsymbol{H}_{\mathcal{S}_r\mathcal{S}_r})_{ij}|\geqslant t)\leqslant2\exp\left(-\frac{mt^2}{2}\right),$$

for $t > 0$. Setting $t = \frac{t}{|\mathcal{S}_r|}$ for all $i, j \in \mathcal{S}_r$ and applying the union bound,

$$\text{Prob}(\|(\tilde{\boldsymbol{H}}_{\mathcal{S}_r \mathcal{S}_r}) - (\boldsymbol{H}_{\mathcal{S}_r \mathcal{S}_r})\|_F \geqslant t) \leqslant 2|\mathcal{S}_r|^2 \exp\left(-\frac{mt^2}{2|\mathcal{S}_r|^2}\right). \tag{8}$$

To conclude, with probability at least $1 - 2|\mathcal{S}_r|^2 \exp\left(-\frac{mt^2}{2|\mathcal{S}_r|^2}\right)$, we have

$$\Lambda_{\min}(\boldsymbol{H}^{\mathbb{Q}}_{\mathcal{S}_r \mathcal{S}_r}) \geqslant \Lambda_{\min}(\boldsymbol{H}_{\mathcal{S}_r \mathcal{S}_r}) - 4\varepsilon|\mathcal{S}_r|^{\frac{1}{2}} - t.$$

$\square$

**Lemma 8.** *If Assumption 3 and Assumption 4 hold, for any $\mathbb{Q} \in \mathcal{A}^{W_p}_\varepsilon(\tilde{\mathbb{P}}_m)$ and $\alpha \in (0, 1]$, with probability at least $1 - \mathcal{O}(\exp\left(-\frac{Cm}{\rho^2_{max}|\mathcal{S}_r|^3} + \log|\mathcal{S}^c_r| + \log|\mathcal{S}_r|\right))$ and $\varepsilon \leqslant \frac{C}{\rho_{max}|\mathcal{S}_r|^{3/2}}$,*

$$\|\boldsymbol{H}^{\mathbb{Q}}_{\mathcal{S}^c_r \mathcal{S}_r}(\boldsymbol{H}^{\mathbb{Q}}_{\mathcal{S}_r \mathcal{S}_r})^{-1}\|_{B,1,\infty} \leqslant 1 - \frac{\alpha}{2},$$

*where $C$ only depends on $\alpha$, $\Lambda_{min}(\boldsymbol{H}_{\mathcal{S}_r \mathcal{S}_r})$.*

*Proof.* We would like to obtain an upper bound for $\|\boldsymbol{H}^{\mathbb{Q}}_{\mathcal{S}^c_r \mathcal{S}_r}(\boldsymbol{H}^{\mathbb{Q}}_{\mathcal{S}_r \mathcal{S}_r})^{-1}\|_{B,1,\infty}$. We may write

$$\begin{aligned}
\boldsymbol{H}^{\mathbb{Q}}_{\mathcal{S}^c_r \mathcal{S}_r}(\boldsymbol{H}^{\mathbb{Q}}_{\mathcal{S}_r \mathcal{S}_r})^{-1} =& \boldsymbol{H}_{\mathcal{S}^c_r \mathcal{S}_r}[(\boldsymbol{H}^{\mathbb{Q}}_{\mathcal{S}_r \mathcal{S}_r})^{-1} - (\boldsymbol{H}_{\mathcal{S}_r \mathcal{S}_r})^{-1}] \\
&+ [\boldsymbol{H}^{\mathbb{Q}}_{\mathcal{S}^c_r \mathcal{S}_r} - \boldsymbol{H}_{\mathcal{S}^c_r \mathcal{S}_r}](\boldsymbol{H}_{\mathcal{S}_r \mathcal{S}_r})^{-1} \\
&+ [\boldsymbol{H}^{\mathbb{Q}}_{\mathcal{S}^c_r \mathcal{S}_r} - \boldsymbol{H}_{\mathcal{S}^c_r \mathcal{S}_r}][(\boldsymbol{H}^{\mathbb{Q}}_{\mathcal{S}_r \mathcal{S}_r})^{-1} - (\boldsymbol{H}_{\mathcal{S}_r \mathcal{S}_r})^{-1}] \\
&+ \boldsymbol{H}_{\mathcal{S}^c_r \mathcal{S}_r}(\boldsymbol{H}_{\mathcal{S}_r \mathcal{S}_r})^{-1} \\
\implies& \\
\|\boldsymbol{H}^{\mathbb{Q}}_{\mathcal{S}^c_r \mathcal{S}_r}(\boldsymbol{H}^{\mathbb{Q}}_{\mathcal{S}_r \mathcal{S}_r})^{-1}\|_{B,1,\infty} \leqslant& \|\boldsymbol{H}_{\mathcal{S}^c_r \mathcal{S}_r}[(\boldsymbol{H}^{\mathbb{Q}}_{\mathcal{S}_r \mathcal{S}_r})^{-1} - (\boldsymbol{H}_{\mathcal{S}_r \mathcal{S}_r})^{-1}]\|_{B,1,\infty} \\
&+ \|[\boldsymbol{H}^{\mathbb{Q}}_{\mathcal{S}^c_r \mathcal{S}_r} - \boldsymbol{H}_{\mathcal{S}^c_r \mathcal{S}_r}](\boldsymbol{H}_{\mathcal{S}_r \mathcal{S}_r})^{-1}\|_{B,1,\infty} \\
&+ \|[\boldsymbol{H}^{\mathbb{Q}}_{\mathcal{S}^c_r \mathcal{S}_r} - \boldsymbol{H}_{\mathcal{S}^c_r \mathcal{S}_r}][(\boldsymbol{H}^{\mathbb{Q}}_{\mathcal{S}_r \mathcal{S}_r})^{-1} - (\boldsymbol{H}_{\mathcal{S}_r \mathcal{S}_r})^{-1}]\|_{B,1,\infty} \\
&+ \|\boldsymbol{H}_{\mathcal{S}^c_r \mathcal{S}_r}(\boldsymbol{H}_{\mathcal{S}_r \mathcal{S}_r})^{-1}\|_{B,1,\infty}.
\end{aligned}$$

By Hoeffding's inequality,

$$\text{Prob}(|(\tilde{\boldsymbol{H}}_{\mathcal{S}^c_r \mathcal{S}_r})_{ij} - (\boldsymbol{H}_{\mathcal{S}^c_r \mathcal{S}_r})_{ij}| \geqslant t) \leqslant 2\exp\left(-\frac{mt^2}{2}\right),$$

for $t > 0$. Taking $t = \frac{t}{\rho_i|\mathcal{S}_r|}$ and applying the union bound over $i \in \textbf{Co}_r$, we observe that

$$\begin{aligned}
\text{Prob}(\|\tilde{\boldsymbol{H}}_{\mathcal{S}^c_r \mathcal{S}_r} - \boldsymbol{H}_{\mathcal{S}^c_r \mathcal{S}_r}\|_{B,1,\infty} \geqslant t) &\leqslant \sum_{i \in \textbf{Co}_r} 2\rho_i|\mathcal{S}_r| \exp\left(-\frac{mt^2}{2\rho_i^2|\mathcal{S}_r|^2}\right) \\
&\leqslant 2|\mathcal{S}^c_r||\mathcal{S}_r| \exp\left(-\frac{mt^2}{2\rho_{\max}^2|\mathcal{S}_r|^2}\right).
\end{aligned}$$

Similarly, taking $t = \frac{t}{|\mathcal{S}_r|}$,

$$\begin{aligned}
\text{Prob}(\|\!\|\tilde{\boldsymbol{H}}_{\mathcal{S}_r \mathcal{S}_r} - \boldsymbol{H}_{\mathcal{S}_r \mathcal{S}_r}\|\!\|_{\infty,\infty} \geqslant t) &\leqslant \sum_{i \in \mathcal{S}_r} \sum_{j \in \mathcal{S}_r} 2\exp\left(-\frac{mt^2}{2|\mathcal{S}_r|^2}\right) \\
&= 2|\mathcal{S}_r|^2 \exp\left(-\frac{mt^2}{2|\mathcal{S}_r|^2}\right).
\end{aligned}$$

In order to bound $\|\boldsymbol{H}^{\mathbb{Q}}_{\mathcal{S}^c_r \mathcal{S}_r} - \boldsymbol{H}_{\mathcal{S}^c_r \mathcal{S}_r}\|_{B,1,\infty}$, for $\mathbb{Q} \neq \tilde{\mathbb{P}}$, consider

$$\begin{aligned}
\|\boldsymbol{H}^{\mathbb{Q}}_{\mathcal{S}^c_r \mathcal{S}_r} - \tilde{\boldsymbol{H}}_{\mathcal{S}^c_r \mathcal{S}_r}\|_{B,1,\infty} \leqslant& \|\boldsymbol{H}^{\mathbb{Q}}_{\mathcal{S}^c_r \mathcal{S}_r}\|_{B,1,\infty} + \|\tilde{\boldsymbol{H}}_{\mathcal{S}^c_r \mathcal{S}_r}\|_{B,1,\infty} \\
\leqslant& \mathbb{E}_{\mathbb{Q}}\|\mathcal{E}(\boldsymbol{X}_{\bar{r}})_{\mathcal{S}^c_r}\mathcal{E}(\boldsymbol{X}_{\bar{r}})^\intercal_{\mathcal{S}_r}\|_{B,1,\infty} + \mathbb{E}_{\tilde{\mathbb{P}}_m}\|\mathcal{E}(\boldsymbol{X}_{\bar{r}})_{\mathcal{S}^c_r}\mathcal{E}(\boldsymbol{X}_{\bar{r}})^\intercal_{\mathcal{S}_r}\|_{B,1,\infty} \\
=& \sup_{\tilde{\mathbb{P}}'_m, \mathbb{Q}' \in \mathcal{A}^{W_p}_\varepsilon(\tilde{\mathbb{P}}'_m)} |\mathbb{E}_{\mathbb{Q}'}\xi_1\|\mathcal{E}(\boldsymbol{X}_{\bar{r}})_{\mathcal{S}^c_r}\mathcal{E}(\boldsymbol{X}_{\bar{r}})^\intercal_{\mathcal{S}_r}\|_{B,1,\infty} - \mathbb{E}_{\tilde{\mathbb{P}}'_m}\xi_2\|\mathcal{E}(\boldsymbol{X}_{\bar{r}})_{\mathcal{S}^c_r}\mathcal{E}(\boldsymbol{X}_{\bar{r}})^\intercal_{\mathcal{S}_r}\|_{B,1,\infty}|,
\end{aligned}$$

where $\mathbb{Q}'$ and $\tilde{\mathbb{P}}'_m$ are probability measures on $\mathcal{X} \times \Xi$ with $\Xi = \{-1, +1\}$ and identical marginals as $\mathbb{Q}$ and $\tilde{\mathbb{P}}_m$ respectively. We assume that $\mathbb{Q} \neq \tilde{\mathbb{P}}$ because otherwise $\|\boldsymbol{H}^{\mathbb{Q}}_{\mathcal{S}^c_r \mathcal{S}_r} - \tilde{\boldsymbol{H}}_{\mathcal{S}^c_r \mathcal{S}_r}\|_{B,1,\infty} = 0$ holds trivially. In this way, the equality is always achieved by some $\mathbb{Q}', \tilde{\mathbb{P}}'_m$, i.e., setting $\mathbb{Q}'(\mathcal{X}, \xi = 1) = 1$ and $\tilde{\mathbb{P}}'_m(\mathcal{X}, \xi = -1) = 1$.

Define the transport cost function in the ambiguity set $\mathcal{A}^{W_p}_\varepsilon(\tilde{\mathbb{P}}'_m)$ to be $c'((\boldsymbol{X}_1, \xi_1), (\boldsymbol{X}_2, \xi_2)) := \|\mathcal{E}(\boldsymbol{X}_1) - \mathcal{E}(\boldsymbol{X}_2)\|_1$ with zero cost for $\xi$. Let $g(\boldsymbol{X}, \xi) := \xi_1 \|\mathcal{E}(\boldsymbol{X}_{\bar{r}})_{\mathcal{S}^c_r} \mathcal{E}(\boldsymbol{X}_{\bar{r}})^{\mathsf{T}}_{\mathcal{S}_r}\|_{B,1,\infty}$. Consider the Lipschitz constants of $g$:

$$\begin{aligned}
\text{lip}_{c'}(g) &\leqslant \sup_{\boldsymbol{X}, \xi, \boldsymbol{X}', \xi'} \frac{|g(\boldsymbol{X}, \xi) - g(\boldsymbol{X}', \xi')|}{c'((\boldsymbol{X}, \xi), (\boldsymbol{X}', \xi'))} \\
&\leqslant \sup_{\boldsymbol{X}, \boldsymbol{X}'} \frac{\|\mathcal{E}(\boldsymbol{X}_{\bar{r}})_{\mathcal{S}^c_r} \mathcal{E}(\boldsymbol{X}_{\bar{r}})^{\mathsf{T}}_{\mathcal{S}_r}\|_{B,1,\infty} + \|\mathcal{E}(\boldsymbol{X}'_{\bar{r}})_{\mathcal{S}^c_r} \mathcal{E}(\boldsymbol{X}'_{\bar{r}})^{\mathsf{T}}_{\mathcal{S}_r}\|_{B,1,\infty}}{\|\mathcal{E}(\boldsymbol{X}) - \mathcal{E}(\boldsymbol{X}')\|_1} \\
&\leqslant 2\rho_{\max}|\mathcal{S}_r|.
\end{aligned} \tag{9}$$

Therefore, by the Kantorovich-Rubinstein theorem [Kantorovich and Rubinshtein, 1958],

$$\begin{aligned}
\|\boldsymbol{H}^{\mathbb{Q}}_{\mathcal{S}^c_r \mathcal{S}_r} - \tilde{\boldsymbol{H}}_{\mathcal{S}^c_r \mathcal{S}_r}\|_{B,1,\infty} &\leqslant \sup_{\tilde{\mathbb{P}}'_m, \mathbb{Q}' \in \mathcal{A}^{W_p}_\varepsilon(\tilde{\mathbb{P}}'_m)} |\mathbb{E}_{\mathbb{Q}'} g(\boldsymbol{X}, \xi) - \mathbb{E}_{\tilde{\mathbb{P}}'_m} g(\boldsymbol{X}, \xi)| \\
&\leqslant \sup_{\tilde{\mathbb{P}}'_m, \mathbb{Q}' \in \mathcal{A}^{W_p}_\varepsilon(\tilde{\mathbb{P}}'_m)} \text{lip}_{c'}(g) |\mathbb{E}_{\mathbb{Q}'} g(\boldsymbol{X}, \xi)/\text{lip}_{c'}(g) - \mathbb{E}_{\tilde{\mathbb{P}}'_m} g(\boldsymbol{X}, \xi)/\text{lip}_{c'}(g)| \\
&\leqslant \sup_{\tilde{\mathbb{P}}'_m, \mathbb{Q}' \in \mathcal{A}^{W_p}_\varepsilon(\tilde{\mathbb{P}}'_m)} \text{lip}_{c'}(g) W_1(\mathbb{Q}', \tilde{\mathbb{P}}'_m) \\
&\leqslant \text{lip}_{c'}(g) \varepsilon \\
&\leqslant 2\varepsilon \rho_{\max}|\mathcal{S}_r|.
\end{aligned}$$

Similarly,

$$\|\boldsymbol{H}^{\mathbb{Q}}_{\mathcal{S}_r \mathcal{S}_r} - \tilde{\boldsymbol{H}}_{\mathcal{S}_r \mathcal{S}_r}\|_{\infty,\infty} \leqslant 2\varepsilon|\mathcal{S}_r|.$$

Based on the above two inequalities, we find that

$$\begin{aligned}
\|\boldsymbol{H}^{\mathbb{Q}}_{\mathcal{S}^c_r \mathcal{S}_r} - \boldsymbol{H}_{\mathcal{S}^c_r \mathcal{S}_r}\|_{B,1,\infty} &\leqslant \|\boldsymbol{H}^{\mathbb{Q}}_{\mathcal{S}^c_r \mathcal{S}_r} - \tilde{\boldsymbol{H}}_{\mathcal{S}^c_r \mathcal{S}_r}\|_{B,1,\infty} + \|\tilde{\boldsymbol{H}}_{\mathcal{S}^c_r \mathcal{S}_r} - \boldsymbol{H}_{\mathcal{S}^c_r \mathcal{S}_r}\|_{B,1,\infty} \\
&\leqslant 2\varepsilon \rho_{\max}|\mathcal{S}_r| + t,
\end{aligned} \tag{10}$$

with probability at least $1 - 2|\mathcal{S}^c_r||\mathcal{S}_r| \exp\left(-\frac{mt^2}{2\rho^2_{\max}|\mathcal{S}_r|^2}\right)$, and

$$\|\boldsymbol{H}^{\mathbb{Q}}_{\mathcal{S}_r \mathcal{S}_r} - \boldsymbol{H}_{\mathcal{S}_r \mathcal{S}_r}\|_{\infty,\infty} \leqslant 2\varepsilon|\mathcal{S}_r| + t, \tag{11}$$

with probability at least $1 - 2|\mathcal{S}_r|^2 \exp\left(-\frac{mt^2}{2|\mathcal{S}_r|^2}\right)$.

Based on Equation (8), we also have

$$\|[\boldsymbol{H}_{\mathcal{S}_r \mathcal{S}_r} - \boldsymbol{H}^{\mathbb{Q}}_{\mathcal{S}_r \mathcal{S}_r}]\|_F \leqslant 2\varepsilon|\mathcal{S}_r| + t, \tag{12}$$

with probability at least $1 - 2|\mathcal{S}_r|^2 \exp\left(-\frac{mt^2}{2|\mathcal{S}_r|^2}\right)$.

Next we look at the upper bound on the difference between the inverses of $\boldsymbol{H}^{\mathbb{Q}}_{\mathcal{S}_r \mathcal{S}_r}$ and $\boldsymbol{H}_{\mathcal{S}_r \mathcal{S}_r}$. Observe that

$$\begin{aligned}
\|(\boldsymbol{H}^{\mathbb{Q}}_{\mathcal{S}_r \mathcal{S}_r})^{-1} - (\boldsymbol{H}_{\mathcal{S}_r \mathcal{S}_r})^{-1}\|_{\infty,\infty} &= \|(\boldsymbol{H}_{\mathcal{S}_r \mathcal{S}_r})^{-1}[\boldsymbol{H}_{\mathcal{S}_r \mathcal{S}_r} - \boldsymbol{H}^{\mathbb{Q}}_{\mathcal{S}_r \mathcal{S}_r}](\boldsymbol{H}^{\mathbb{Q}}_{\mathcal{S}_r \mathcal{S}_r})^{-1}\|_{\infty,\infty} \\
&\leqslant \sqrt{|\mathcal{S}_r|} \|(\boldsymbol{H}_{\mathcal{S}_r \mathcal{S}_r})^{-1}[\boldsymbol{H}_{\mathcal{S}_r \mathcal{S}_r} - \boldsymbol{H}^{\mathbb{Q}}_{\mathcal{S}_r \mathcal{S}_r}](\boldsymbol{H}^{\mathbb{Q}}_{\mathcal{S}_r \mathcal{S}_r})^{-1}\|_{2,2} \\
&\leqslant \sqrt{|\mathcal{S}_r|} \|(\boldsymbol{H}_{\mathcal{S}_r \mathcal{S}_r})^{-1}\|_{2,2} \|[\boldsymbol{H}_{\mathcal{S}_r \mathcal{S}_r} - \boldsymbol{H}^{\mathbb{Q}}_{\mathcal{S}_r \mathcal{S}_r}]\|_{2,2} \|(\boldsymbol{H}^{\mathbb{Q}}_{\mathcal{S}_r \mathcal{S}_r})^{-1}\|_{2,2} \\
&\leqslant \sqrt{\frac{|\mathcal{S}_r|}{\Lambda_{\min}(\boldsymbol{H}_{\mathcal{S}_r \mathcal{S}_r})}} \|[\boldsymbol{H}_{\mathcal{S}_r \mathcal{S}_r} - \boldsymbol{H}^{\mathbb{Q}}_{\mathcal{S}_r \mathcal{S}_r}]\|_{2,2} \|(\boldsymbol{H}^{\mathbb{Q}}_{\mathcal{S}_r \mathcal{S}_r})^{-1}\|_{2,2}.
\end{aligned}$$

According to Lemma 7, with probability at least $1 - 2|\mathcal{S}_r|^2 \exp\left(-\frac{mt^2}{2|\mathcal{S}_r|^2}\right)$, we have

$$\Lambda_{\min}(\boldsymbol{H}_{\mathcal{S}_r\mathcal{S}_r}^{\mathbb{Q}}) \geqslant \Lambda_{\min}(\boldsymbol{H}_{\mathcal{S}_r\mathcal{S}_r}) - 4\varepsilon|\mathcal{S}_r|^{\frac{1}{2}} - t.$$

Let $t = \frac{1}{2}\Lambda_{\min}(\boldsymbol{H}_{\mathcal{S}_r\mathcal{S}_r})$ and $\varepsilon \leqslant \frac{\Lambda_{\min}(\boldsymbol{H}_{\mathcal{S}_r\mathcal{S}_r})}{16|\mathcal{S}_r|^{\frac{1}{2}}}$. We get that, with probability at least $1 - 2|\mathcal{S}_r|^2 \exp\left(-\frac{m(\Lambda_{\min}(\boldsymbol{H}_{\mathcal{S}_r\mathcal{S}_r}))^2}{8|\mathcal{S}_r|^2}\right)$,

$$\Lambda_{\min}(\boldsymbol{H}_{\mathcal{S}_r\mathcal{S}_r}^{\mathbb{Q}}) \geqslant \frac{1}{4}\Lambda_{\min}(\boldsymbol{H}_{\mathcal{S}_r\mathcal{S}_r})$$

$$\Longrightarrow \||(\boldsymbol{H}_{\mathcal{S}_r\mathcal{S}_r}^{\mathbb{Q}})^{-1}\||_{2,2} \leqslant \sqrt{\frac{4}{\Lambda_{\min}(\boldsymbol{H}_{\mathcal{S}_r\mathcal{S}_r})}}. \tag{13}$$

Set $t = \frac{t\Lambda_{\min}(\boldsymbol{H}_{\mathcal{S}_r\mathcal{S}_r})}{4\sqrt{|\mathcal{S}_r|}}$ and $\varepsilon \leqslant \frac{t\Lambda_{\min}(\boldsymbol{H}_{\mathcal{S}_r\mathcal{S}_r})}{8|\mathcal{S}_r|\sqrt{|\mathcal{S}_r|}}$ in Equation (12), we get that, with probability at least $1 - 2|\mathcal{S}_r|^2 \exp\left(-\frac{mt^2(\Lambda_{\min}(\boldsymbol{H}_{\mathcal{S}_r\mathcal{S}_r}))^2}{32|\mathcal{S}_r|^3}\right)$,

$$\||[\boldsymbol{H}_{\mathcal{S}_r\mathcal{S}_r} - \boldsymbol{H}_{\mathcal{S}_r\mathcal{S}_r}^{\mathbb{Q}}]\||_{2,2} \leqslant \||[\boldsymbol{H}_{\mathcal{S}_r\mathcal{S}_r} - \boldsymbol{H}_{\mathcal{S}_r\mathcal{S}_r}^{\mathbb{Q}}]\||_F \leqslant \frac{t\Lambda_{\min}(\boldsymbol{H}_{\mathcal{S}_r\mathcal{S}_r})}{2\sqrt{|\mathcal{S}_r|}}.$$

Therefore, with probability at least $1 - 2|\mathcal{S}_r|^2 \exp\left(-\frac{mt^2(\Lambda_{\min}(\boldsymbol{H}_{\mathcal{S}_r\mathcal{S}_r}))^2}{32|\mathcal{S}_r|^3}\right) - 2|\mathcal{S}_r|^2 \exp\left(-\frac{m(\Lambda_{\min}(\boldsymbol{H}_{\mathcal{S}_r\mathcal{S}_r}))^2}{8|\mathcal{S}_r|^2}\right)$ and $\varepsilon \leqslant \min\left(\frac{t\Lambda_{\min}(\boldsymbol{H}_{\mathcal{S}_r\mathcal{S}_r})}{8|\mathcal{S}_r|\sqrt{|\mathcal{S}_r|}}, \frac{\Lambda_{\min}(\boldsymbol{H}_{\mathcal{S}_r\mathcal{S}_r})}{16|\mathcal{S}_r|^{\frac{1}{2}}}\right)$,

$$\||(\boldsymbol{H}_{\mathcal{S}_r\mathcal{S}_r}^{\mathbb{Q}})^{-1} - (\boldsymbol{H}_{\mathcal{S}_r\mathcal{S}_r})^{-1}\||_{\infty,\infty} \leqslant t. \tag{14}$$

Now we are ready to obtain upper bounds for the four terms recapitulated here:

$$\|\boldsymbol{H}_{\mathcal{S}_r^c\mathcal{S}_r}^{\mathbb{Q}}(\boldsymbol{H}_{\mathcal{S}_r\mathcal{S}_r}^{\mathbb{Q}})^{-1}\|_{B,1,\infty} \leqslant \|\boldsymbol{H}_{\mathcal{S}_r^c\mathcal{S}_r}^{\mathbb{Q}}[(\boldsymbol{H}_{\mathcal{S}_r\mathcal{S}_r}^{\mathbb{Q}})^{-1} - (\boldsymbol{H}_{\mathcal{S}_r\mathcal{S}_r})^{-1}]\|_{B,1,\infty}$$

$$+ \|[\boldsymbol{H}_{\mathcal{S}_r^c\mathcal{S}_r}^{\mathbb{Q}} - \boldsymbol{H}_{\mathcal{S}_r^c\mathcal{S}_r}](\boldsymbol{H}_{\mathcal{S}_r\mathcal{S}_r})^{-1}\|_{B,1,\infty}$$

$$+ \|[\boldsymbol{H}_{\mathcal{S}_r^c\mathcal{S}_r}^{\mathbb{Q}} - \boldsymbol{H}_{\mathcal{S}_r^c\mathcal{S}_r}][(\boldsymbol{H}_{\mathcal{S}_r\mathcal{S}_r}^{\mathbb{Q}})^{-1} - (\boldsymbol{H}_{\mathcal{S}_r\mathcal{S}_r})^{-1}]\|_{B,1,\infty}$$

$$+ \|\boldsymbol{H}_{\mathcal{S}_r^c\mathcal{S}_r}(\boldsymbol{H}_{\mathcal{S}_r\mathcal{S}_r})^{-1}\|_{B,1,\infty}.$$

We derive the bounds separately.

For the first term, based on Assumption 4, consider

$$\|\boldsymbol{H}_{\mathcal{S}_r^c\mathcal{S}_r}[(\boldsymbol{H}_{\mathcal{S}_r\mathcal{S}_r}^{\mathbb{Q}})^{-1} - (\boldsymbol{H}_{\mathcal{S}_r\mathcal{S}_r})^{-1}]\|_{B,1,\infty}$$

$$= \|\boldsymbol{H}_{\mathcal{S}_r^c\mathcal{S}_r}(\boldsymbol{H}_{\mathcal{S}_r\mathcal{S}_r})^{-1}[\boldsymbol{H}_{\mathcal{S}_r\mathcal{S}_r} - \boldsymbol{H}_{\mathcal{S}_r\mathcal{S}_r}^{\mathbb{Q}}](\boldsymbol{H}_{\mathcal{S}_r\mathcal{S}_r}^{\mathbb{Q}})^{-1}\|_{B,1,\infty}$$

$$\leqslant \|\boldsymbol{H}_{\mathcal{S}_r^c\mathcal{S}_r}(\boldsymbol{H}_{\mathcal{S}_r\mathcal{S}_r})^{-1}\|_{B,1,\infty}\||\boldsymbol{H}_{\mathcal{S}_r\mathcal{S}_r} - \boldsymbol{H}_{\mathcal{S}_r\mathcal{S}_r}^{\mathbb{Q}}\||_{\infty,\infty}\||(\boldsymbol{H}_{\mathcal{S}_r\mathcal{S}_r}^{\mathbb{Q}})^{-1}\||_{\infty,\infty}$$

$$\leqslant (1-\alpha)\||\boldsymbol{H}_{\mathcal{S}_r\mathcal{S}_r} - \boldsymbol{H}_{\mathcal{S}_r\mathcal{S}_r}^{\mathbb{Q}}\||_{\infty,\infty}\sqrt{|\mathcal{S}_r|}\||(\boldsymbol{H}_{\mathcal{S}_r\mathcal{S}_r}^{\mathbb{Q}})^{-1}\||_{2,2}.$$

Taking $t = \frac{\alpha}{24(1-\alpha)}\sqrt{\frac{\Lambda_{\min}(\boldsymbol{H}_{\mathcal{S}_r\mathcal{S}_r})}{|\mathcal{S}_r|}}$ and $\varepsilon \leqslant \frac{\alpha}{48(1-\alpha)|\mathcal{S}_r|}\sqrt{\frac{\Lambda_{\min}(\boldsymbol{H}_{\mathcal{S}_r\mathcal{S}_r})}{|\mathcal{S}_r|}}$ in Equation (11) and adopting Equation (13), we conclude that, with probability at least $1 - 2|\mathcal{S}_r|^2 \exp\left(-\frac{m\alpha^2\Lambda_{\min}(\boldsymbol{H}_{\mathcal{S}_r\mathcal{S}_r})}{1152(1-\alpha)^2|\mathcal{S}_r|^3}\right) - 2|\mathcal{S}_r|^2 \exp\left(-\frac{m(\Lambda_{\min}(\boldsymbol{H}_{\mathcal{S}_r\mathcal{S}_r}))^2}{8|\mathcal{S}_r|^2}\right)$ and $\varepsilon \leqslant \min\left(\frac{\alpha}{48(1-\alpha)|\mathcal{S}_r|}\sqrt{\frac{\Lambda_{\min}(\boldsymbol{H}_{\mathcal{S}_r\mathcal{S}_r})}{|\mathcal{S}_r|}}, \frac{\Lambda_{\min}(\boldsymbol{H}_{\mathcal{S}_r\mathcal{S}_r})}{16|\mathcal{S}_r|^{\frac{1}{2}}}\right)$,

$$\|\boldsymbol{H}_{\mathcal{S}_r^c\mathcal{S}_r}[(\boldsymbol{H}_{\mathcal{S}_r\mathcal{S}_r}^{\mathbb{Q}})^{-1} - (\boldsymbol{H}_{\mathcal{S}_r\mathcal{S}_r})^{-1}]\|_{B,1,\infty} \leqslant \frac{\alpha}{6}.$$

For the second term, rewrite it as

$$\|[\boldsymbol{H}_{\mathcal{S}_r^c\mathcal{S}_r}^{\mathbb{Q}} - \boldsymbol{H}_{\mathcal{S}_r^c\mathcal{S}_r}](\boldsymbol{H}_{\mathcal{S}_r\mathcal{S}_r})^{-1}\|_{B,1,\infty}$$

$$\leqslant \|[\boldsymbol{H}_{\mathcal{S}_r^c\mathcal{S}_r}^{\mathbb{Q}} - \boldsymbol{H}_{\mathcal{S}_r^c\mathcal{S}_r}]\|_{B,1,\infty}\||(\boldsymbol{H}_{\mathcal{S}_r\mathcal{S}_r})^{-1}\||_{\infty,\infty}$$

$$\leqslant \|[\boldsymbol{H}_{\mathcal{S}_r^c\mathcal{S}_r}^{\mathbb{Q}} - \boldsymbol{H}_{\mathcal{S}_r^c\mathcal{S}_r}]\|_{B,1,\infty}\sqrt{|\mathcal{S}_r|}\||(\boldsymbol{H}_{\mathcal{S}_r\mathcal{S}_r})^{-1}\||_{2,2}$$

$$\leqslant \|[\boldsymbol{H}_{\mathcal{S}_r^c\mathcal{S}_r}^{\mathbb{Q}} - \boldsymbol{H}_{\mathcal{S}_r^c\mathcal{S}_r}]\|_{B,1,\infty}\sqrt{\frac{|\mathcal{S}_r|}{\Lambda_{\min}(\boldsymbol{H}_{\mathcal{S}_r\mathcal{S}_r})}}.$$

Using Equation (10) by setting $t = \frac{\alpha}{12}\sqrt{\frac{\Lambda_{\min}(\boldsymbol{H}_{\mathcal{S}_r\mathcal{S}_r})}{|\mathcal{S}_r|}}$ and $\varepsilon \leqslant \frac{\alpha}{24\rho_{\max}|\mathcal{S}_r|}\sqrt{\frac{\Lambda_{\min}(\boldsymbol{H}_{\mathcal{S}_r\mathcal{S}_r})}{|\mathcal{S}_r|}}$, we have, with probability at least $1 - 2|\mathcal{S}_r^c||\mathcal{S}_r|\exp\left(-\frac{m\alpha^2\Lambda_{\min}(\boldsymbol{H}_{\mathcal{S}_r\mathcal{S}_r})}{288\rho_{\max}^2|\mathcal{S}_r|^3}\right)$ and $\varepsilon \leqslant \frac{\alpha}{24\rho_{\max}|\mathcal{S}_r|}\sqrt{\frac{\Lambda_{\min}(\boldsymbol{H}_{\mathcal{S}_r\mathcal{S}_r})}{|\mathcal{S}_r|}}$,

$$\|[\boldsymbol{H}_{\mathcal{S}_r^c\mathcal{S}_r}^{\mathbb{Q}} - \boldsymbol{H}_{\mathcal{S}_r^c\mathcal{S}_r}](\boldsymbol{H}_{\mathcal{S}_r\mathcal{S}_r})^{-1}\|_{B,1,\infty} \leqslant \frac{\alpha}{6}.$$

For the third term, we obtain the upper bound

$$\|[\boldsymbol{H}_{\mathcal{S}_r^c\mathcal{S}_r}^{\mathbb{Q}} - \boldsymbol{H}_{\mathcal{S}_r^c\mathcal{S}_r}][(\boldsymbol{H}_{\mathcal{S}_r\mathcal{S}_r}^{\mathbb{Q}})^{-1} - (\boldsymbol{H}_{\mathcal{S}_r\mathcal{S}_r})^{-1}]\|_{B,1,\infty}$$
$$\leqslant \|[\boldsymbol{H}_{\mathcal{S}_r^c\mathcal{S}_r}^{\mathbb{Q}} - \boldsymbol{H}_{\mathcal{S}_r^c\mathcal{S}_r}]\|_{B,1,\infty}\|[(\boldsymbol{H}_{\mathcal{S}_r\mathcal{S}_r}^{\mathbb{Q}})^{-1} - (\boldsymbol{H}_{\mathcal{S}_r\mathcal{S}_r})^{-1}]\|_{\infty,\infty}.$$

Taking $t = \sqrt{\frac{\alpha}{6}}$ in Equation (14). Taking $t = \frac{1}{2}\sqrt{\frac{\alpha}{6}}$ and $2\varepsilon\rho_{\max}|\mathcal{S}_r| \leqslant \frac{1}{2}\sqrt{\frac{\alpha}{6}}$ in Equation (10). We establish the upper bound that, with probability at least $1 - 2|\mathcal{S}_r^c||\mathcal{S}_r|\exp\left(-\frac{m\alpha}{48\rho_{\max}^2|\mathcal{S}_r|^2}\right) - 2|\mathcal{S}_r|^2\exp\left(-\frac{m\alpha(\Lambda_{\min}(\boldsymbol{H}_{\mathcal{S}_r\mathcal{S}_r}))^2}{192|\mathcal{S}_r|^3}\right) - 2|\mathcal{S}_r|^2\exp\left(-\frac{m(\Lambda_{\min}(\boldsymbol{H}_{\mathcal{S}_r\mathcal{S}_r}))^2}{8|\mathcal{S}_r|^2}\right)$ and $\varepsilon \leqslant \min\left(\frac{1}{4\rho_{\max}|\mathcal{S}_r|}\sqrt{\frac{\alpha}{6}}, \frac{\Lambda_{\min}(\boldsymbol{H}_{\mathcal{S}_r\mathcal{S}_r})}{8|\mathcal{S}_r|}\sqrt{\frac{\alpha}{6|\mathcal{S}_r|}}, \frac{\Lambda_{\min}(\boldsymbol{H}_{\mathcal{S}_r\mathcal{S}_r})}{16|\mathcal{S}_r|^{\frac{1}{2}}}\right)$,

$$\|[\boldsymbol{H}_{\mathcal{S}_r^c\mathcal{S}_r}^{\mathbb{Q}} - \boldsymbol{H}_{\mathcal{S}_r^c\mathcal{S}_r}][(\boldsymbol{H}_{\mathcal{S}_r\mathcal{S}_r}^{\mathbb{Q}})^{-1} - (\boldsymbol{H}_{\mathcal{S}_r\mathcal{S}_r})^{-1}]\|_{B,1,\infty} \leqslant \frac{\alpha}{6}.$$

For the fourth term, in accordance with Assumption 4,

$$\|\boldsymbol{H}_{\mathcal{S}_r^c\mathcal{S}_r}(\boldsymbol{H}_{\mathcal{S}_r\mathcal{S}_r})^{-1}\|_{B,1,\infty} \leqslant 1 - \alpha.$$

In conclusion, we have shown that, with probability at least $1 - 2|\mathcal{S}_r|^2\exp\left(-\frac{m\alpha^2\Lambda_{\min}(\boldsymbol{H}_{\mathcal{S}_r\mathcal{S}_r})}{1152(1-\alpha)^2|\mathcal{S}_r|^3}\right) - 2|\mathcal{S}_r|^2\exp\left(-\frac{m(\Lambda_{\min}(\boldsymbol{H}_{\mathcal{S}_r\mathcal{S}_r}))^2}{8|\mathcal{S}_r|^2}\right) - 2|\mathcal{S}_r^c||\mathcal{S}_r|\exp\left(-\frac{m\alpha^2\Lambda_{\min}(\boldsymbol{H}_{\mathcal{S}_r\mathcal{S}_r})}{288\rho_{\max}^2|\mathcal{S}_r|^3}\right) - 2|\mathcal{S}_r^c||\mathcal{S}_r|\exp\left(-\frac{m\alpha}{48\rho_{\max}^2|\mathcal{S}_r|^2}\right) - 2|\mathcal{S}_r|^2\exp\left(-\frac{m\alpha(\Lambda_{\min}(\boldsymbol{H}_{\mathcal{S}_r\mathcal{S}_r}))^2}{192|\mathcal{S}_r|^3}\right) - 2|\mathcal{S}_r|^2\exp\left(-\frac{m(\Lambda_{\min}(\boldsymbol{H}_{\mathcal{S}_r\mathcal{S}_r}))^2}{8|\mathcal{S}_r|^2}\right)$ and

$$\varepsilon \leqslant \min\Big(\frac{\alpha}{48(1-\alpha)|\mathcal{S}_r|}\sqrt{\frac{\Lambda_{\min}(\boldsymbol{H}_{\mathcal{S}_r\mathcal{S}_r})}{|\mathcal{S}_r|}}, \frac{\Lambda_{\min}(\boldsymbol{H}_{\mathcal{S}_r\mathcal{S}_r})}{16|\mathcal{S}_r|^{\frac{1}{2}}}, \frac{\alpha}{24\rho_{\max}|\mathcal{S}_r|}\sqrt{\frac{\Lambda_{\min}(\boldsymbol{H}_{\mathcal{S}_r\mathcal{S}_r})}{|\mathcal{S}_r|}},$$
$$\frac{1}{4\rho_{\max}|\mathcal{S}_r|}\sqrt{\frac{\alpha}{6}}, \frac{\Lambda_{\min}(\boldsymbol{H}_{\mathcal{S}_r\mathcal{S}_r})}{8|\mathcal{S}_r|}\sqrt{\frac{\alpha}{6|\mathcal{S}_r|}}, \frac{\Lambda_{\min}(\boldsymbol{H}_{\mathcal{S}_r\mathcal{S}_r})}{16|\mathcal{S}_r|^{\frac{1}{2}}}\Big),$$

the mutual incoherence condition holds for any worst-case distributions:

$$\|\boldsymbol{H}_{\mathcal{S}_r^c\mathcal{S}_r}^{\mathbb{Q}}(\boldsymbol{H}_{\mathcal{S}_r\mathcal{S}_r}^{\mathbb{Q}})^{-1}\|_{B,1,\infty} \leqslant 1 - \frac{\alpha}{2}.$$

Simplifying the above expressions, with probability at least $1 - \mathcal{O}\left(\exp\left(-\frac{Cm}{\rho_{\max}^2|\mathcal{S}_r|^3} + \log|\mathcal{S}_r^c| + \log|\mathcal{S}_r|\right)\right)$ and $\varepsilon \leqslant \frac{C}{\rho_{\max}|\mathcal{S}_r|^{3/2}}$,

$$\|\boldsymbol{H}_{\mathcal{S}_r^c\mathcal{S}_r}^{\mathbb{Q}}(\boldsymbol{H}_{\mathcal{S}_r\mathcal{S}_r}^{\mathbb{Q}})^{-1}\|_{B,1,\infty} \leqslant 1 - \frac{\alpha}{2},$$

where $C$ only depends on $\alpha, \Lambda_{\min}(\boldsymbol{H}_{\mathcal{S}_r\mathcal{S}_r})$. $\qquad\qquad\square$

**Lemma 12.** *If Assumption 1 holds, then for any $\mathbb{Q} \in \mathcal{A}_\varepsilon^{W_p}(\tilde{\mathbb{P}}_m)$ and $\alpha \in (0, 1]$, with probability at least $1 - |\mathcal{S}_r|\rho_r\exp\left(-\frac{m\mu^2}{2\sigma^2}\right)$, $\varepsilon \leqslant \frac{\mu}{\sigma}$ and $\lambda_B^* > \frac{32\mu\sqrt{\rho_r}(1-\alpha/2)}{\alpha}$, we have*

$$\|\mathbb{E}_{\mathbb{Q}}\mathcal{E}(\boldsymbol{X}_{\bar{r}})_{\mathcal{S}_r}\boldsymbol{e}^{\mathsf{T}}\|_{2,\infty} \leqslant \frac{\lambda_B^*\alpha}{8(1-\alpha/2)}.$$

*With probability at least $1 - |\boldsymbol{Co}_r|\rho_r\exp\left(-\frac{m\mu^2}{2\sigma^2}\right)$, $\varepsilon \leqslant \frac{\mu}{\sigma}$ and $\lambda_B^* > \frac{32\mu\sqrt{\rho_{max}\rho_r}}{\alpha}$, we have*

$$\|\mathbb{E}_{\mathbb{Q}}\mathcal{E}(\boldsymbol{X}_{\bar{r}})_{\mathcal{S}_r^c}\boldsymbol{e}^{\mathsf{T}}\|_{B,2,\infty} \leqslant \frac{\lambda_B^*\alpha}{8}.$$

*Proof.* We start with $\|\mathbb{E}_\mathbb{Q}\mathcal{E}(\boldsymbol{X}_{\bar{r}})_{\mathcal{S}_r}\boldsymbol{e}^\intercal\|_{2,\infty}$. After some algeraic manipulation, we find that

$$\begin{aligned}
\|\mathbb{E}_\mathbb{Q}\mathcal{E}(\boldsymbol{X}_{\bar{r}})_{\mathcal{S}_r}\boldsymbol{e}^\intercal\|_{2,\infty} &\leqslant \max_{i\in\mathcal{S}_r}\|\mathbb{E}_\mathbb{Q}\mathcal{E}(\boldsymbol{X}_{\bar{r}})_i\boldsymbol{e}\|_2 \\
&\leqslant \max_{i\in\mathcal{S}_r}\sqrt{\rho_r}\max_{j\in\rho_r}|\mathbb{E}_\mathbb{Q}\mathcal{E}(\boldsymbol{X}_{\bar{r}})_ie_j| \\
&\leqslant \max_{i\in\mathcal{S}_r}\sqrt{\rho_r}\max_{j\in\rho_r}\mathbb{E}_\mathbb{Q}|\mathcal{E}(\boldsymbol{X}_{\bar{r}})_ie_j| \\
&\leqslant \max_{i\in\mathcal{S}_r}\sqrt{\rho_r}\max_{j\in\rho_r}\mathbb{E}_\mathbb{Q}|e_j|.
\end{aligned}$$

Since $|e_j|$ is a bounded random variable according to Assumption 1, we apply Hoeffding's inequality to get

$$\mathrm{Prob}(\mathbb{E}_{\tilde{\mathbb{P}}_m}|e_j| \geqslant \mu + t) \leqslant \exp\left(-\frac{mt^2}{2\sigma^2}\right).$$

Base on a similar argument as Equation (9), we can derive

$$\mathbb{E}_\mathbb{Q}|e_j| - \mathbb{E}_{\tilde{\mathbb{P}}_m}|e_j| \leqslant 2\varepsilon\sigma,$$

which leads to

$$\mathrm{Prob}(\mathbb{E}_\mathbb{Q}|e_j| \geqslant 2\varepsilon\sigma + \mu + t) \leqslant \exp\left(-\frac{mt^2}{2\sigma^2}\right).$$

Taking the union bound over all $i\in\mathcal{S}_r$ and $j\in\rho_r$, we find that

$$\mathrm{Prob}(\|\mathbb{E}_\mathbb{Q}\mathcal{E}(\boldsymbol{X}_{\bar{r}})_{\mathcal{S}_r}\boldsymbol{e}^\intercal\|_{2,\infty} \geqslant \sqrt{\rho_r}(2\varepsilon\sigma+\mu+t)) \leqslant |\mathcal{S}_r|\rho_r\exp\left(-\frac{mt^2}{2\sigma^2}\right).$$

Setting $t = \mu$ and $\varepsilon \leqslant \frac{\mu}{\sigma}$ while requiring $\lambda_B^* > \frac{32\mu\sqrt{\rho_r}(1-\alpha/2)}{\alpha}$. With probability at least $1 - |\mathcal{S}_r|\rho_r\exp\left(-\frac{m\mu^2}{2\sigma^2}\right)$, we have

$$\|\mathbb{E}_\mathbb{Q}\mathcal{E}(\boldsymbol{X}_{\bar{r}})_{\mathcal{S}_r}\boldsymbol{e}^\intercal\|_{2,\infty} \leqslant \frac{\lambda_B^*\alpha}{8(1-\alpha/2)}. \tag{15}$$

Then we consider $\|\mathbb{E}_\mathbb{Q}\mathcal{E}(\boldsymbol{X}_{\bar{r}})_{\mathcal{S}_r^c}\boldsymbol{e}^\intercal\|_{B,2,\infty}$:

$$\begin{aligned}
\|\mathbb{E}_\mathbb{Q}\mathcal{E}(\boldsymbol{X}_{\bar{r}})_{\mathcal{S}_r^c}\boldsymbol{e}^\intercal\|_{B,2,\infty} &\leqslant \max_{i\in\mathbf{Co}_r}\|\mathbb{E}_\mathbb{Q}\mathcal{E}(X_i)\boldsymbol{e}^\intercal\|_{2,2} \\
&\leqslant \max_{i\in\mathbf{Co}_r}\sqrt{\rho_i\rho_r}\max_{j\in\rho_i,k\in\rho_r}|\mathbb{E}_\mathbb{Q}\mathcal{E}(X_i)_je_k| \\
&\leqslant \max_{i\in\mathbf{Co}_r}\sqrt{\rho_i\rho_r}\max_{k\in\rho_r}\mathbb{E}_\mathbb{Q}|e_k|.
\end{aligned}$$

Similarly, applying Hoeffding's inequality and the Kantorovich-Rubinstein theorem gives us

$$\mathrm{Prob}(\|\mathbb{E}_\mathbb{Q}\mathcal{E}(\boldsymbol{X}_{\bar{r}})_{\mathcal{S}_r^c}\boldsymbol{e}^\intercal\|_{B,2,\infty} \geqslant \sqrt{\rho_{\max}\rho_r}(2\varepsilon\sigma+\mu+t)) \leqslant |\mathbf{Co}_r|\rho_r\exp\left(-\frac{mt^2}{2\sigma^2}\right).$$

Let $t = \mu$, $\varepsilon \leqslant \frac{\mu}{\sigma}$ and $\lambda_B^* > \frac{32\mu\sqrt{\rho_{\max}\rho_r}}{\alpha}$ hold, we have, with probability at least $1 - |\mathbf{Co}_r|\rho_r\exp\left(-\frac{m\mu^2}{2\sigma^2}\right)$,

$$\|\mathbb{E}_\mathbb{Q}\mathcal{E}(\boldsymbol{X}_{\bar{r}})_{\mathcal{S}_r^c}\boldsymbol{e}^\intercal\|_{B,2,\infty} \leqslant \frac{\lambda_B^*\alpha}{8}.$$

$\square$

**Theorem 9.** *Given a Bayesian network $(\mathcal{G}, \mathbb{P})$ of $n$ categorical random variables and its skeleton $\mathcal{G}_{skel} := (\mathcal{V}, \mathcal{E}_{skel})$. Assume that the condition $\|\boldsymbol{W}^*\|_{B,2,1} \leqslant \bar{B}$ holds for some $\bar{B} > 0$ associated with an optimal Lagrange multiplier $\lambda_B^* > 0$ for $\boldsymbol{W}^*$ defined in Equation (1). Suppose that $\hat{\boldsymbol{W}}$ is a DRO risk minimizer of Equation (4) with a Wasserstein distance of order 1 and an ambiguity radius*

$\varepsilon = \varepsilon_0/m$ where $m$ is the number of samples drawn i.i.d. from $\mathbb{P}$. *Under Assumptions 1, 2, 3, 4, if the number of samples satisfies*

$$m = \mathcal{O}\Big(\frac{C(\varepsilon_0 + \log{(n/\delta)} + \log{\rho_{[n]}})\sigma^2 \rho_{max}^4 \rho_{[n]}^3}{\min(\mu^2, 1)}\Big),$$

*where $C$ only depends on $\alpha$, $\Lambda$, and if the Lagrange multiplier satisfies*

$$\frac{32\mu\rho_{max}}{\alpha} < \lambda_B^* < \frac{\beta}{(\alpha/(4-2\alpha)+2)\rho_{max}\sqrt{\rho_{[n]}}}\sqrt{\frac{\Lambda}{4}},$$

*then for any $\delta \in (0, 1]$, $r \in [n]$, with probability at least $1 - \delta$, the following properties hold:*

(a) *The optimal estimator $\hat{\boldsymbol{W}}$ is unique.*

(b) *All the non-neighbor nodes are excluded: $\boldsymbol{Co}_r \subseteq \hat{\boldsymbol{Co}}_r$.*

(c) *All the neighbor nodes are identified: $\boldsymbol{Ne}_r \subseteq \hat{\boldsymbol{Ne}}_r$.*

(d) *The true skeleton is successfully reconstructed: $\mathcal{G}_{skel} = \hat{\mathcal{G}}_{skel}$.*

*Proof.* We prove the statements in this theorem in several steps. In order to prove (a) and (b), we will show that the DRO problem is strictly convex if true non-neighbors are known so that there is an optimal solution. Next we would like to demonstrate that this solution with a non-neighbor constraint is indeed unique for all the solutions without constraints. The proof for uniqueness comes with a conclusion that we do not accidentally include any edge between the current node and its non-neighbors. Next, to prove (c), we present a generalization bound for the DRO estimator in terms of its true risk, which leads to a $\ell_\infty$ bound of the difference between the estimator $\hat{\boldsymbol{W}}$ and the true weight matrix $\boldsymbol{W}^*$. Combined with the assumption on the minimum weight, it implies that we include all the neighbor nodes successfully. Finally, by taking a union bound for all the nodes, we could conclude that the correct skeleton is recovered with high probability, which proves (d).

**(i) Given the true non-neighbors, there is a unique solution.**

We start with the Wasserstein DRO problem, which we recapitulate here for convenience:

$$\hat{\boldsymbol{W}} \in \arg\inf_{\boldsymbol{W}} \sup_{\mathbb{Q} \in \mathcal{A}_\varepsilon^{W_p}(\tilde{\mathbb{P}}_m)} \frac{1}{2}\mathbb{E}_{\mathbb{Q}}\|\mathcal{E}(X_r) - \boldsymbol{W}^\intercal \mathcal{E}(\boldsymbol{X}_{\bar{r}})\|_2^2.$$

The objective is convex because it is a supremum of convex functions.

For now, we assume that the non-neighbor nodes $\boldsymbol{Co}_r$ are given. We can then explicitly restrict $\boldsymbol{W}_i = \boldsymbol{0}$ for all $i \in \boldsymbol{Co}_r$. The Hessian of $\boldsymbol{W}_{\mathcal{S}_r\cdot}$ is a block diagonal matrix reads

$$\nabla^2 R^{\mathbb{Q}}(\boldsymbol{W}_{\mathcal{S}_r\cdot}) = \begin{bmatrix} \boldsymbol{H}_{\mathcal{S}_r\mathcal{S}_r}^{\mathbb{Q}} & \boldsymbol{0} & \cdots & \boldsymbol{0} \\ \boldsymbol{0} & \boldsymbol{H}_{\mathcal{S}_r\mathcal{S}_r}^{\mathbb{Q}} & \cdots & \boldsymbol{0} \\ \vdots & \vdots & \ddots & \vdots \\ \boldsymbol{0} & \boldsymbol{0} & \cdots & \boldsymbol{H}_{\mathcal{S}_r\mathcal{S}_r}^{\mathbb{Q}} \end{bmatrix} \in \mathbb{R}^{\rho_r \rho_{\boldsymbol{Ne}_r} \times \rho_r \rho_{\boldsymbol{Ne}_r}},$$

where

$$\boldsymbol{H}^{\mathbb{Q}} := \mathbb{E}_{\mathbb{Q}}[\mathcal{E}(\boldsymbol{X}_{\bar{r}})\mathcal{E}(\boldsymbol{X}_{\bar{r}})^\intercal] \in \mathbb{R}^{\rho_{\bar{r}} \times \rho_{\bar{r}}}$$

is the covariance matrix of encodings of $\boldsymbol{X}_{\bar{r}}$ under some distribution $\mathbb{Q} \in \mathcal{A}_\varepsilon^{W_p}(\tilde{\mathbb{P}}_m)$.

Since $\boldsymbol{W}_{\mathcal{S}_r^c\cdot}$ is fixed to be zero and $\nabla^2 R^{\mathbb{Q}}(\boldsymbol{W}_{\mathcal{S}_r\cdot})$ is a block diagonal matrix, we focus on showing that $\boldsymbol{H}_{\mathcal{S}_r\mathcal{S}_r}^{\mathbb{Q}} > \boldsymbol{0}$.

We apply Lemma 7 to get the bound

$$\Lambda_{\min}(\boldsymbol{H}_{\mathcal{S}_r\mathcal{S}_r}^{\mathbb{Q}}) \geqslant \Lambda_{\min}(\boldsymbol{H}_{\mathcal{S}_r\mathcal{S}_r}) - 4\varepsilon|\mathcal{S}_r|^{\frac{1}{2}} - t,$$

with probability at least $1 - 2|\mathcal{S}_r|^2 \exp\left(-\frac{mt^2}{2|\mathcal{S}_r|^2}\right)$. $\Lambda_{\min}(\boldsymbol{H}_{\mathcal{S}_r,\mathcal{S}_r}) - 4\varepsilon|\mathcal{S}_r|^{\frac{1}{2}} - t > 0$ will guarantee that the DRO problem in Equation (4) has a unique solution when the $\boldsymbol{W}_i = \boldsymbol{0}$ is satisfied for non-neighbor nodes.

**(ii) Given the true non-neighbors, the solution is optimal.**

We would like to show that the solution to Equation (4) with true non-neighbor constraints is optimal. In this way, we do not recover any non-neighbor nodes in the skeleton. We adopt the primal-dual witness (PDW) [Wainwright, 2009] method to show optimality for the constrained unique solution.

Recall that we assume $\|\boldsymbol{W}\|_{B,2,1} \leqslant \bar{B}$. To begin with, we write the dual problem as

$$\hat{\boldsymbol{W}} \in \arg\inf_{\boldsymbol{W}} \sup_{\mathbb{Q} \in \mathcal{A}_\varepsilon^{W_p}(\tilde{\mathbb{P}}_m), \|\boldsymbol{Z}\|_{B,2,\infty} \leqslant 1, \lambda_B \geqslant 0} \frac{1}{2}\mathbb{E}_{\mathbb{Q}}\|\mathcal{E}(X_r) - \boldsymbol{W}^{\mathsf{T}}\mathcal{E}(\boldsymbol{X}_{\bar{r}})\|_2^2 + \lambda_B(\langle \boldsymbol{Z}, \boldsymbol{W} \rangle - \bar{B})$$
(16)

s.t. $\quad \forall i \in \mathbf{Co}_r \quad \boldsymbol{W}_i = \boldsymbol{0},$

where $\lambda_B$ is the Lagrange multiplier for the norm constraint on $\boldsymbol{W}$.

$\hat{\boldsymbol{W}}$ is optimal if and only if there exists $(\mathbb{Q}^*, \boldsymbol{Z}^*, \lambda_B^*)$ that satisfies the KKT condition:

$$\mathbb{E}_{\mathbb{Q}^*}\mathcal{E}(\boldsymbol{X}_{\bar{r}})\mathcal{E}(\boldsymbol{X}_{\bar{r}})^{\mathsf{T}}\hat{\boldsymbol{W}} - \mathbb{E}_{\mathbb{Q}^*}\mathcal{E}(\boldsymbol{X}_{\bar{r}})\mathcal{E}(X_r)^{\mathsf{T}} + \lambda_B^*\boldsymbol{Z}^* = \boldsymbol{0}$$
$$\mathbb{Q}^* \in \mathcal{A}_\varepsilon^{W_p}(\tilde{\mathbb{P}}_m), \|\boldsymbol{Z}^*\|_{B,2,\infty} \leqslant 1, \lambda_B^* \geqslant 0, \|\hat{\boldsymbol{W}}\|_{B,2,1} \leqslant \bar{B}$$
$$\langle \boldsymbol{Z}^*, \hat{\boldsymbol{W}} \rangle = \|\hat{\boldsymbol{W}}\|_{B,2,1}, \lambda_B^*(\|\hat{\boldsymbol{W}}\|_{B,2,1} - \bar{B}) = 0.$$

Note that we assume that the constraint $\|\boldsymbol{W}\|_{B,2,1} \leqslant \bar{B}$ is active such that $\lambda_B^* > 0$. This assumption is only for convenience of theoretical analysis and not restrictive. If it is not active, we have $\|\hat{\boldsymbol{W}}\|_{B,2,1} = \check{B} < \bar{B}$ for some $\check{B}$ and $\lambda_B^* = 0$, which leads to an unconstrained problem similar to the ordinary least square problem, which is known to suffer from overfitting. Instead, we are usually interested in solutions that have finite norms so we can always find $\bar{B} = \check{B} - \epsilon < \check{B}$ for some small positive constant $\epsilon > 0$ to make the constraint active and thus $\lambda_B^* > 0$.

Substituting $\mathcal{E}(X_r) = \boldsymbol{W}^{*\mathsf{T}}\mathcal{E}(\boldsymbol{X}_{\bar{r}}) + \boldsymbol{e}$ into the first-order optimality condition yields

$$\mathbb{E}_{\mathbb{Q}^*}\mathcal{E}(\boldsymbol{X}_{\bar{r}})\mathcal{E}(\boldsymbol{X}_{\bar{r}})^{\mathsf{T}}(\hat{\boldsymbol{W}} - \boldsymbol{W}^*) - \mathbb{E}_{\mathbb{Q}^*}\mathcal{E}(\boldsymbol{X}_{\bar{r}})\boldsymbol{e}^{\mathsf{T}} + \lambda_B^*\boldsymbol{Z}^* = \boldsymbol{0}$$
$$\Longleftrightarrow \begin{bmatrix} \boldsymbol{H}_{\mathcal{S}_r\mathcal{S}_r}^{\mathbb{Q}^*} & \boldsymbol{H}_{\mathcal{S}_r\mathcal{S}_r^c}^{\mathbb{Q}^*} \\ \boldsymbol{H}_{\mathcal{S}_r^c\mathcal{S}_r}^{\mathbb{Q}^*} & \boldsymbol{H}_{\mathcal{S}_r^c\mathcal{S}_r^c}^{\mathbb{Q}^*} \end{bmatrix} \begin{bmatrix} \hat{\boldsymbol{W}}_{\mathcal{S}_r\cdot} - \boldsymbol{W}_{\mathcal{S}_r\cdot}^* \\ \boldsymbol{0} \end{bmatrix} - \begin{bmatrix} \mathbb{E}_{\mathbb{Q}^*}\mathcal{E}(\boldsymbol{X}_{\bar{r}})_{\mathcal{S}_r}\boldsymbol{e}^{\mathsf{T}} \\ \mathbb{E}_{\mathbb{Q}^*}\mathcal{E}(\boldsymbol{X}_{\bar{r}})_{\mathcal{S}_r^c}\boldsymbol{e}^{\mathsf{T}} \end{bmatrix} + \lambda_B^* \begin{bmatrix} \boldsymbol{Z}_{\mathcal{S}_r\cdot}^* \\ \boldsymbol{Z}_{\mathcal{S}_r^c\cdot}^* \end{bmatrix} = \begin{bmatrix} \boldsymbol{0} \\ \boldsymbol{0} \end{bmatrix}. \quad (17)$$

Solving for $\boldsymbol{Z}_{\mathcal{S}_r^c\cdot}^*$, we find that

$$\lambda_B^*\boldsymbol{Z}_{\mathcal{S}_r^c\cdot}^* = \lambda_B^*\boldsymbol{H}_{\mathcal{S}_r^c\mathcal{S}_r}^{\mathbb{Q}^*}(\boldsymbol{H}_{\mathcal{S}_r\mathcal{S}_r}^{\mathbb{Q}^*})^{-1}\boldsymbol{Z}_{\mathcal{S}_r\cdot}^* - \boldsymbol{H}_{\mathcal{S}_r^c\mathcal{S}_r}^{\mathbb{Q}^*}(\boldsymbol{H}_{\mathcal{S}_r\mathcal{S}_r}^{\mathbb{Q}^*})^{-1}\mathbb{E}_{\mathbb{Q}^*}\mathcal{E}(\boldsymbol{X}_{\bar{r}})_{\mathcal{S}_r}\boldsymbol{e}^{\mathsf{T}} + \mathbb{E}_{\mathbb{Q}^*}\mathcal{E}(\boldsymbol{X}_{\bar{r}})_{\mathcal{S}_r^c}\boldsymbol{e}^{\mathsf{T}},$$

which can be bounded such that

$$\lambda_B^*\|\boldsymbol{Z}_{\mathcal{S}_r^c\cdot}^*\|_{B,2,\infty}$$
$$= \|\lambda_B^*\boldsymbol{H}_{\mathcal{S}_r^c\mathcal{S}_r}^{\mathbb{Q}^*}(\boldsymbol{H}_{\mathcal{S}_r\mathcal{S}_r}^{\mathbb{Q}^*})^{-1}\boldsymbol{Z}_{\mathcal{S}_r\cdot}^* - \boldsymbol{H}_{\mathcal{S}_r^c\mathcal{S}_r}^{\mathbb{Q}^*}(\boldsymbol{H}_{\mathcal{S}_r\mathcal{S}_r}^{\mathbb{Q}^*})^{-1}\mathbb{E}_{\mathbb{Q}^*}\mathcal{E}(\boldsymbol{X}_{\bar{r}})_{\mathcal{S}_r}\boldsymbol{e}^{\mathsf{T}} + \mathbb{E}_{\mathbb{Q}^*}\mathcal{E}(\boldsymbol{X}_{\bar{r}})_{\mathcal{S}_r^c}\boldsymbol{e}^{\mathsf{T}}\|_{B,2,\infty}$$
$$\leqslant \lambda_B^*\|\boldsymbol{H}_{\mathcal{S}_r^c\mathcal{S}_r}^{\mathbb{Q}^*}(\boldsymbol{H}_{\mathcal{S}_r\mathcal{S}_r}^{\mathbb{Q}^*})^{-1}\boldsymbol{Z}_{\mathcal{S}_r\cdot}^*\|_{B,2,\infty} + \|\boldsymbol{H}_{\mathcal{S}_r^c\mathcal{S}_r}^{\mathbb{Q}^*}(\boldsymbol{H}_{\mathcal{S}_r\mathcal{S}_r}^{\mathbb{Q}^*})^{-1}\mathbb{E}_{\mathbb{Q}^*}\mathcal{E}(\boldsymbol{X}_{\bar{r}})_{\mathcal{S}_r}\boldsymbol{e}^{\mathsf{T}}\|_{B,2,\infty} + \|\mathbb{E}_{\mathbb{Q}^*}\mathcal{E}(\boldsymbol{X}_{\bar{r}})_{\mathcal{S}_r^c}\boldsymbol{e}^{\mathsf{T}}\|_{B,2,\infty}$$
$$\leqslant \lambda_B^*\|\boldsymbol{H}_{\mathcal{S}_r^c\mathcal{S}_r}^{\mathbb{Q}^*}(\boldsymbol{H}_{\mathcal{S}_r\mathcal{S}_r}^{\mathbb{Q}^*})^{-1}\|_{B,1,\infty}\|\boldsymbol{Z}_{\mathcal{S}_r\cdot}^*\|_{2,\infty} + \|\boldsymbol{H}_{\mathcal{S}_r^c\mathcal{S}_r}^{\mathbb{Q}^*}(\boldsymbol{H}_{\mathcal{S}_r\mathcal{S}_r}^{\mathbb{Q}^*})^{-1}\|_{B,1,\infty}\|\mathbb{E}_{\mathbb{Q}^*}\mathcal{E}(\boldsymbol{X}_{\bar{r}})_{\mathcal{S}_r}\boldsymbol{e}^{\mathsf{T}}\|_{2,\infty}$$
$$+ \|\mathbb{E}_{\mathbb{Q}^*}\mathcal{E}(\boldsymbol{X}_{\bar{r}})_{\mathcal{S}_r^c}\boldsymbol{e}^{\mathsf{T}}\|_{B,2,\infty}.$$

Note that

$$\|\boldsymbol{Z}_{\mathcal{S}_r\cdot}^*\|_{2,\infty} \leqslant \|\boldsymbol{Z}^*\|_{B,2,\infty} \leqslant 1.$$

Recall that $0 < \alpha \leqslant 1$ in Assumption 4. Based on Lemma 8 and Lemma 12, we may write

$$\lambda_B^* \|\boldsymbol{Z}_{\mathcal{S}_r^c}^*\|_{B,2,\infty}$$

$$\leqslant \lambda_B^* \|\boldsymbol{H}_{\mathcal{S}_r^c \mathcal{S}_r}^{\mathbb{Q}^*} (\boldsymbol{H}_{\mathcal{S}_r \mathcal{S}_r}^{\mathbb{Q}^*})^{-1}\|_{B,1,\infty} \|\boldsymbol{Z}_{\mathcal{S}_r \cdot}^*\|_{2,\infty} + \|\boldsymbol{H}_{\mathcal{S}_r^c \mathcal{S}_r}^{\mathbb{Q}^*} (\boldsymbol{H}_{\mathcal{S}_r \mathcal{S}_r}^{\mathbb{Q}^*})^{-1}\|_{B,1,\infty} \|\mathbb{E}_{\mathbb{Q}*} \mathcal{E}(\boldsymbol{X}_{\bar{r}})_{\mathcal{S}_r} \boldsymbol{e}^{\mathsf{T}}\|_{2,\infty}$$

$$\quad + \|\mathbb{E}_{\mathbb{Q}*} \mathcal{E}(\boldsymbol{X}_{\bar{r}})_{\mathcal{S}_r^c} \boldsymbol{e}^{\mathsf{T}}\|_{B,2,\infty}$$

$$\leqslant \lambda_B^* (1 - \frac{\alpha}{2}) + (1 - \frac{\alpha}{2})(\frac{\lambda_B^* \alpha}{8(1 - \alpha/2)}) + \frac{\lambda_B^* \alpha}{8}$$

$$\leqslant \lambda_B^* (1 - \frac{\alpha}{4})$$

$$< \lambda_B^*,$$

with high probability and certain conditions on $\lambda_B^*$ and $\varepsilon$.

Henceforth, $\|\boldsymbol{Z}_{\mathcal{S}_r^c}^*\|_{B,2,\infty} < 1$ satisfies strict dual feasibility and we must have $\|\hat{\boldsymbol{W}}_{\mathcal{S}_r^c}\|_{B,2,1} = 0$ according to complementary slackness: $\langle \boldsymbol{Z}^*, \hat{\boldsymbol{W}} \rangle = \|\hat{\boldsymbol{W}}\|_{B,2,1}$. In other words, we have

$$\forall i \in \mathbf{Co}_r \quad \hat{\boldsymbol{W}}_i = \boldsymbol{0},$$

with high probability. This guarantees that we do not recover any node that is not a neighbor of $r$ with high probability.

**(iii) Without information about the true skeleton, we have a unique and optimal solution.**

We follow the proof of Lemma 11.2 in Hastie et al. [2015].

We have shown that $\hat{\boldsymbol{W}}$ satisfying $\hat{\boldsymbol{W}}_i = \boldsymbol{0}$ $\quad \forall i \in \mathbf{Co}_r$ is an optimal solution with optimal dual variables $\|\boldsymbol{Z}_{\mathcal{S}_r^c}^*\|_{B,2,\infty} < 1$.

To avoid clutter of notations, we define

$$L^{\mathrm{DRO}}(\boldsymbol{W}) := \sup_{\mathbb{Q} \in \mathcal{A}_\varepsilon^{W_p}(\tilde{\mathbb{P}}_m)} \frac{1}{2} \mathbb{E}_{\mathbb{Q}} \|\mathcal{E}(X_r) - \boldsymbol{W}^{\mathsf{T}} \mathcal{E}(\boldsymbol{X}_{\bar{r}})\|_2^2.$$

Let $(\check{\boldsymbol{W}}, \check{\lambda})$ be any other optimal solution to $\inf_{\boldsymbol{W}} \sup_\lambda L^{\mathrm{DRO}}(\boldsymbol{W}) + \lambda(\|\boldsymbol{W}\|_{B,2,1} - \bar{B})$. By definition,

$$L^{\mathrm{DRO}}(\check{\boldsymbol{W}}) + \check{\lambda}(\|\check{\boldsymbol{W}}\|_{B,2,1} - \bar{B}) = L^{\mathrm{DRO}}(\hat{\boldsymbol{W}}) + \lambda_B^*(\langle \boldsymbol{Z}^*, \hat{\boldsymbol{W}} \rangle - \bar{B})$$

$$\iff L^{\mathrm{DRO}}(\check{\boldsymbol{W}}) + \check{\lambda}(\|\check{\boldsymbol{W}}\|_{B,2,1} - \bar{B}) - \lambda_B^* \langle \boldsymbol{Z}^*, \check{\boldsymbol{W}} \rangle = L^{\mathrm{DRO}}(\hat{\boldsymbol{W}}) + \lambda_B^*(\langle \boldsymbol{Z}^*, \hat{\boldsymbol{W}} - \check{\boldsymbol{W}} \rangle - \bar{B}).$$

The first-order optimality condition for $\hat{\boldsymbol{W}}$ says

$$\nabla L^{\mathrm{DRO}}(\hat{\boldsymbol{W}}) + \lambda_B^* \boldsymbol{Z}^* = \boldsymbol{0},$$

which implies

$$\check{\lambda}(\|\check{\boldsymbol{W}}\|_{B,2,1} - \bar{B}) + \lambda_B^*(\bar{B} - \langle \boldsymbol{Z}^*, \check{\boldsymbol{W}} \rangle) = L^{\mathrm{DRO}}(\hat{\boldsymbol{W}}) + \langle \nabla L^{\mathrm{DRO}}(\hat{\boldsymbol{W}}), \check{\boldsymbol{W}} - \hat{\boldsymbol{W}} \rangle - L^{\mathrm{DRO}}(\check{\boldsymbol{W}}).$$

By definition, $\|\check{\boldsymbol{W}}\|_{B,2,1} - \bar{B} = 0$ and $\lambda_B^* > 0$. Since $L^{\mathrm{DRO}}(\cdot)$ is convex, the RHS of the above equation should be non-positive, or equivalently,

$$\|\check{\boldsymbol{W}}\|_{B,2,1} \leqslant \langle \boldsymbol{Z}^*, \check{\boldsymbol{W}} \rangle.$$

On the other hand,

$$\langle \boldsymbol{Z}^*, \check{\boldsymbol{W}} \rangle \leqslant \|\boldsymbol{Z}^*\|_{B,2,\infty} \|\check{\boldsymbol{W}}\|_{B,2,1} \leqslant \|\check{\boldsymbol{W}}\|_{B,2,1}.$$

Therefore, the equality holds for the above inequalities, which leads to

$$\|\check{\boldsymbol{W}}\|_{B,2,1} = \langle \boldsymbol{Z}^*, \check{\boldsymbol{W}} \rangle.$$

Recall that $\|\boldsymbol{Z}_{\mathcal{S}_r^c}^*\|_{B,2,\infty} < 1$. In order for $\|\check{\boldsymbol{W}}\|_{B,2,1} = \langle \boldsymbol{Z}^*, \check{\boldsymbol{W}} \rangle$ to hold, we must have

$$\check{\boldsymbol{W}}_{\mathcal{S}_r^c} = \boldsymbol{0}.$$

In that wise, all the optimal solutions $\check{W}$ have

$$\check{W}_i = 0 \quad \forall i \in \mathbf{Co}_r.$$

This implies that we have a unique solution that excludes all the non-neighbor nodes without information about the true skeleton. Until now, we have proven properties (a) and (b).

**(iv) The set of correct neighbors is recovered.**

Consider again the first-order optimality condition in Equation (17),

$$\hat{W}_{\mathcal{S}_r \cdot} - W_{\mathcal{S}_r \cdot}^* = (H_{\mathcal{S}_r \mathcal{S}_r}^{\mathbb{Q}*})^{-1}(\mathbb{E}_{\mathbb{Q}*}\mathcal{E}(X_{\bar{r}})_{\mathcal{S}_r}e^{\mathsf{T}} - \lambda_B^* Z_{\mathcal{S}_r \cdot}^*)$$

$$\implies \|\hat{W}_{\mathcal{S}_r \cdot} - W_{\mathcal{S}_r \cdot}^*\|_{B,2,\infty} = \|(H_{\mathcal{S}_r \mathcal{S}_r}^{\mathbb{Q}*})^{-1}(\mathbb{E}_{\mathbb{Q}*}\mathcal{E}(X_{\bar{r}})_{\mathcal{S}_r}e^{\mathsf{T}} - \lambda_B^* Z_{\mathcal{S}_r \cdot}^*)\|_{B,2,\infty}$$

$$\leqslant \|(H_{\mathcal{S}_r \mathcal{S}_r}^{\mathbb{Q}*})^{-1}\|_{B,1,\infty}\|\mathbb{E}_{\mathbb{Q}*}\mathcal{E}(X_{\bar{r}})_{\mathcal{S}_r}e^{\mathsf{T}} - \lambda_B^* Z_{\mathcal{S}_r \cdot}^*\|_{2,\infty}$$

$$\leqslant \|(H_{\mathcal{S}_r \mathcal{S}_r}^{\mathbb{Q}*})^{-1}\|_{B,1,\infty}(\|\mathbb{E}_{\mathbb{Q}*}\mathcal{E}(X_{\bar{r}})_{\mathcal{S}_r}e^{\mathsf{T}}\|_{2,\infty} + \|\lambda_B^* Z_{\mathcal{S}_r \cdot}^*\|_{2,\infty})$$

$$\leqslant \rho_{\max}\|(H_{\mathcal{S}_r \mathcal{S}_r}^{\mathbb{Q}*})^{-1}\|_{\infty,\infty}(\|\mathbb{E}_{\mathbb{Q}*}\mathcal{E}(X_{\bar{r}})_{\mathcal{S}_r}e^{\mathsf{T}}\|_{2,\infty} + \lambda_B^*)$$

$$\leqslant \rho_{\max}\sqrt{|\mathcal{S}_r|}\|(H_{\mathcal{S}_r \mathcal{S}_r}^{\mathbb{Q}*})^{-1}\|_{2,2}(\|\mathbb{E}_{\mathbb{Q}*}\mathcal{E}(X_{\bar{r}})_{\mathcal{S}_r}e^{\mathsf{T}}\|_{2,\infty} + \lambda_B^*).$$

According to Equation (13), with probability at least $1 - 2|\mathcal{S}_r|^2 \exp\left(-\frac{m(\Lambda_{\min}(H_{\mathcal{S}_r \mathcal{S}_r}))^2}{8|\mathcal{S}_r|^2}\right)$ and $\varepsilon \leqslant \frac{\Lambda_{\min}(H_{\mathcal{S}_r \mathcal{S}_r})}{16|\mathcal{S}_r|^{\frac{1}{2}}}$,

$$\|(H_{\mathcal{S}_r \mathcal{S}_r}^{\mathbb{Q}})^{-1}\|_{2,2} \leqslant \sqrt{\frac{4}{\Lambda_{\min}(H_{\mathcal{S}_r \mathcal{S}_r})}}.$$

According to Equation (15), with probability at least $1 - |\mathcal{S}_r|\rho_r \exp\left(-\frac{m\mu^2}{2\sigma^2}\right)$, $\varepsilon \leqslant \frac{\mu}{\sigma}$ and $\lambda_B^* > \frac{32\mu\sqrt{\rho_r}(1-\alpha/2)}{\alpha}$, we have

$$\|\mathbb{E}_{\mathbb{Q}}\mathcal{E}(X_{\bar{r}})_{\mathcal{S}_r}e^{\mathsf{T}}\|_{2,\infty} \leqslant \frac{\lambda_B^*\alpha}{8(1-\alpha/2)}.$$

On that account, with probability at least $1 - 2|\mathcal{S}_r|^2 \exp\left(-\frac{m(\Lambda_{\min}(H_{\mathcal{S}_r \mathcal{S}_r}))^2}{8|\mathcal{S}_r|^2}\right) - |\mathcal{S}_r|\rho_r \exp\left(-\frac{m\mu^2}{2\sigma^2}\right)$ and $\varepsilon \leqslant \min\left(\frac{\Lambda_{\min}(H_{\mathcal{S}_r \mathcal{S}_r})}{16|\mathcal{S}_r|^{\frac{1}{2}}}, \frac{\mu}{\sigma}\right)$ while requiring $\lambda_B^* > \frac{32\mu\sqrt{\rho_r}(1-\alpha/2)}{\alpha}$,

$$\|\hat{W}_{\mathcal{S}_r \cdot} - W_{\mathcal{S}_r \cdot}^*\|_{B,2,\infty} \leqslant \rho_{\max}\sqrt{|\mathcal{S}_r|}\sqrt{\frac{4}{\Lambda_{\min}(H_{\mathcal{S}_r \mathcal{S}_r})}}\lambda_B^*\left(\frac{\alpha}{8(1-\alpha/2)} + 1\right).$$

By Assumption 2, if the condition $\lambda_B^* < \frac{\beta}{2\left(\frac{\alpha}{8(1-\alpha/2)}+1\right)\rho_{\max}\sqrt{|\mathcal{S}_r|}}\sqrt{\frac{\Lambda_{\min}(H_{\mathcal{S}_r \mathcal{S}_r})}{4}}$ is satisfied, the following inequality holds:

$$\|\hat{W}_{\mathcal{S}_r \cdot} - W_{\mathcal{S}_r \cdot}^*\|_{B,2,\infty} < \beta/2.$$

In this way, we are able to recover all the neighbor nodes with a threshold $\beta/2$. This proves (c).

**(v) The true skeleton is recovered with high probability.**

The above arguments tell us that with high probability and certain conditions for $\varepsilon$ and $\lambda_B^*$ satisfied, for each node $r$, we do not recover any non-neighbor and we do recover all the neighbor nodes. The correct $\mathbf{Ne}_r$ and $\mathbf{Co}_r$ are thus identified. Now we are ready to prove (d).

Putting everything together and taking the the union bound for all nodes $r \in [n]$, with probability at least $1 - \mathcal{O}(n\exp\left(-\frac{Cm\mu^2}{\sigma^2\rho_{\max}^4\rho_{[n]}^3} + 2\log\rho_{[n]}\right))$, $\varepsilon \leqslant \frac{C\mu}{\sigma\rho_{\max}\rho_{[n]}^{3/2}}$ and $\frac{32\mu\rho_{\max}}{\alpha} < \lambda_B^* < \frac{\beta}{2\left(\frac{\alpha}{8(1-\alpha/2)}+1\right)\rho_{\max}\sqrt{\rho_{[n]}}}\sqrt{\frac{\Lambda}{4}}$, where $C$ only depends on $\alpha, \Lambda$, we have

$$\hat{\mathcal{G}}_{\text{skel}} = \mathcal{G}_{\text{skel}}.$$

Setting $\varepsilon = \frac{\varepsilon_0}{m}$ and making the dependence on the sample size more explicit. We draw the conclusion that, if the number of samples satisfies

$$m = \mathcal{O}\big(\frac{C(\varepsilon_0 + \log{(n/\delta)} + \log \rho_{[n]})\sigma^2 \rho_{\max}^4 \rho_{[n]}^3}{\min(\mu^2, 1)}\big),$$

where $C$ only depends on $\alpha, \Lambda$, and if $\lambda_B^*$ satisfies

$$\frac{32\mu\rho_{\max}}{\alpha} < \lambda_B^* < \frac{\beta}{(\alpha/(4-2\alpha)+2)\rho_{\max}\sqrt{\rho_{[n]}}}\sqrt{\frac{\Lambda}{4}},$$

then with probability at least $1 - \delta$ for $\delta \in (0, 1]$:

$$\hat{\mathcal{G}}_{\text{skel}} = \mathcal{G}_{\text{skel}}.$$

Moreover, if we assume that the target graph has a bounded degree of $d$, the sample complexity becomes logarithmic in $n$:

$$m = \mathcal{O}\big(\frac{C(\varepsilon_0 + \log{(n/\delta)} + \log n + \log \rho_{\max})\sigma^2 \rho_{\max}^7 d^3}{\min(\mu^2, 1)}\big).$$

$\square$

**Theorem 10.** *Suppose that $\hat{W}$ is a DRO risk minimizer of Equation (4) with the KL divergence and an ambiguity radius $\varepsilon = \varepsilon_0/m$. Given the same definitions of $(\mathcal{G}, \mathbb{P})$, $\mathcal{G}_{skel}$, $\bar{B}$, $\lambda_B^*$, $m$ in Theorem 9. Under Assumptions 1, 2, 3, 4, if the number of samples satisfies*

$$m = \mathcal{O}\big(\frac{C(\varepsilon_0 + \log{(n/\delta)} + \log \rho_{[n]})\sigma^2 \rho_{max}^4 \rho_{[n]}^3}{\min(\mu^2, 1)}\big).$$

*where $C$ depends on $\alpha, \Lambda$ while independent of $n$, and if the Lagrange multiplier satisfies the same condition as in Theorem 9, then for any $\delta \in (0, 1]$, $r \in [n]$, with probability at least $1 - \delta$, the properties (a)-(d) in Theorem 9 hold.*

*Proof.* Define

$$\ell_{\boldsymbol{W}}(\boldsymbol{X}) := \frac{1}{2}\|\mathcal{E}(X_r) - \boldsymbol{W}^\mathsf{T}\mathcal{E}(\boldsymbol{X}_{\bar{r}})\|_2^2.$$

According to Theorem 7 in Lam [2019], the worst-case risk with a KL divergence ambiguity set can be bounded as follows:

$$\sup_{\mathbb{Q}\in\mathcal{A}_\varepsilon^D(\tilde{\mathbb{P}}_m)} \mathbb{E}_{\mathbb{Q}}\ell_{\boldsymbol{W}}(\boldsymbol{X}) \leq \mathbb{E}_{\tilde{\mathbb{P}}_m}\ell_{\boldsymbol{W}}(\boldsymbol{X}) + \sqrt{\varepsilon}\sqrt{\frac{1}{m}\sum_{i\in[m]}(\ell_{\boldsymbol{W}}(\boldsymbol{x}^{(i)}) - \bar{\ell_{\boldsymbol{W}}})^2} + C\varepsilon\frac{\sum_{i\in[m]}|\ell_{\boldsymbol{W}}(\boldsymbol{x}^{(i)}) - \bar{\ell_{\boldsymbol{W}}}|^3}{\sum_{i\in[m]}(\ell_{\boldsymbol{W}}(\boldsymbol{x}^{(i)}) - \bar{\ell_{\boldsymbol{W}}})^2}$$

$$\leq \mathbb{E}_{\tilde{\mathbb{P}}_m}\ell_{\boldsymbol{W}}(\boldsymbol{X}) + \sqrt{\varepsilon}\max_{i\in[m]}|\ell_{\boldsymbol{W}}(\boldsymbol{x}^{(i)}) - \bar{\ell_{\boldsymbol{W}}}| + C\varepsilon\max_{i\in[m]}|\ell_{\boldsymbol{W}}(\boldsymbol{x}^{(i)}) - \bar{\ell_{\boldsymbol{W}}}|,$$

where $\bar{\ell_{\boldsymbol{W}}} = \frac{1}{m}\sum_{i\in[m]}\ell_{\boldsymbol{W}}(\boldsymbol{x}^{(i)})$ and $C > 0$ is constant independent of $n$.

Consider

$$\max_{i \in [m]} |\ell_{\boldsymbol{W}}(\boldsymbol{x}^{(i)}) - \bar{\ell}_{\boldsymbol{W}}| \leqslant \max_{\boldsymbol{W}, \boldsymbol{W}', \boldsymbol{x}, \boldsymbol{x}'} |\ell_{\boldsymbol{W}}(\boldsymbol{x}) - \ell_{\boldsymbol{W}'}(\boldsymbol{x}')|$$

$$\leqslant \max_{\boldsymbol{W}, \boldsymbol{x}} |\ell_{\boldsymbol{W}}(\boldsymbol{x})|$$

$$\leqslant \frac{1}{2} \max_{\boldsymbol{W}, \boldsymbol{x}} (\|\mathcal{E}(X_r)\|_2 + \|\boldsymbol{W}^{\mathsf{T}} \mathcal{E}(\boldsymbol{X}_{\bar{r}})\|_2)^2$$

$$\leqslant \frac{1}{2} \max_{\boldsymbol{W}, \boldsymbol{x}} (\sqrt{\rho_{\max}} + \|\|\boldsymbol{W}^{\mathsf{T}}\|\|_{\infty, 2})^2$$

$$\leqslant \frac{1}{2} \max_{\boldsymbol{W}, \boldsymbol{x}} (\sqrt{\rho_{\max}} + \|\boldsymbol{W}\|_{1,2})^2$$

$$\leqslant \frac{1}{2} \max_{\boldsymbol{W}, \boldsymbol{x}} (\sqrt{\rho_{\max}} + \sqrt{\rho_{[n]}} \|\boldsymbol{W}\|_F)^2$$

$$\leqslant \frac{1}{2} \max_{\boldsymbol{W}, \boldsymbol{x}} (\sqrt{\rho_{\max}} + \sqrt{\rho_{[n]}} \|\boldsymbol{W}\|_{B,2,1})^2$$

$$\leqslant \frac{1}{2} (\sqrt{\rho_{\max}} + \sqrt{\rho_{[n]}} \bar{B})^2$$

$$:= B_\rho.$$

Define $\varepsilon_{\max} := \max(\sqrt{\varepsilon}, \varepsilon)$. Therefore, we find that

$$\sup_{\mathbb{Q} \in \mathcal{A}_\varepsilon^D(\tilde{\mathbb{P}}_m)} \mathbb{E}_{\mathbb{Q}} \ell_{\boldsymbol{W}}(\boldsymbol{X}) \leqslant \mathbb{E}_{\tilde{\mathbb{P}}_m} \ell_{\boldsymbol{W}}(\boldsymbol{X}) + C \varepsilon_{\max} B_\rho.$$

Similar to the Wasserstein robust risk, we observe that the following results hold for any $\mathbb{Q} \in \mathcal{A}_\varepsilon^D(\tilde{\mathbb{P}}_m)$.

With probability at least $1 - 2|\mathcal{S}_r|^2 \exp\left(-\frac{mt^2}{2|\mathcal{S}_r|^2}\right)$, we have

$$\Lambda_{\min}(\boldsymbol{H}_{\mathcal{S}_r \mathcal{S}_r}^{\mathbb{Q}}) \geqslant \Lambda_{\min}(\boldsymbol{H}_{\mathcal{S}_r \mathcal{S}_r}) - C \varepsilon_{\max} |\mathcal{S}_r|^{\frac{1}{2}} - t.$$

With probability at least $1 - 2|\mathcal{S}_r^c||\mathcal{S}_r| \exp\left(-\frac{mt^2}{2\rho_{\max}^2 |\mathcal{S}_r|^2}\right)$,

$$\|\boldsymbol{H}_{\mathcal{S}_r^c \mathcal{S}_r}^{\mathbb{Q}} - \boldsymbol{H}_{\mathcal{S}_r^c \mathcal{S}_r}\|_{B,1,\infty} \leqslant C \varepsilon_{\max} \rho_{\max} |\mathcal{S}_r| + t.$$

With probability at least $1 - 2|\mathcal{S}_r|^2 \exp\left(-\frac{mt^2}{2|\mathcal{S}_r|^2}\right)$,

$$\|\boldsymbol{H}_{\mathcal{S}_r \mathcal{S}_r}^{\mathbb{Q}} - \boldsymbol{H}_{\mathcal{S}_r \mathcal{S}_r}\|_{\infty,\infty} \leqslant C \varepsilon_{\max} |\mathcal{S}_r| + t.$$

With probability at least $1 - 2|\mathcal{S}_r|^2 \exp\left(-\frac{mt^2(\Lambda_{\min}(\boldsymbol{H}_{\mathcal{S}_r \mathcal{S}_r}))^2}{32|\mathcal{S}_r|^3}\right) - 2|\mathcal{S}_r|^2 \exp\left(-\frac{m(\Lambda_{\min}(\boldsymbol{H}_{\mathcal{S}_r \mathcal{S}_r}))^2}{8|\mathcal{S}_r|^2}\right)$ and $\varepsilon_{\max} \leqslant C \min\left(\frac{t\Lambda_{\min}(\boldsymbol{H}_{\mathcal{S}_r \mathcal{S}_r})}{8|\mathcal{S}_r|\sqrt{|\mathcal{S}_r|}}, \frac{\Lambda_{\min}(\boldsymbol{H}_{\mathcal{S}_r \mathcal{S}_r})}{16|\mathcal{S}_r|^{\frac{1}{2}}}\right)$,

$$\|(\boldsymbol{H}_{\mathcal{S}_r \mathcal{S}_r}^{\mathbb{Q}})^{-1} - (\boldsymbol{H}_{\mathcal{S}_r \mathcal{S}_r})^{-1}\|_{\infty,\infty} \leqslant t.$$

With probability at least $1 - \mathcal{O}\left(\exp\left(-\frac{Cm}{\rho_{\max}^2 |\mathcal{S}_r|^3} + \log|\mathcal{S}_r^c| + \log|\mathcal{S}_r|\right)\right)$ and $\varepsilon_{\max} \leqslant \frac{C}{\rho_{\max} |\mathcal{S}_r|^{3/2}}$,

$$\|\boldsymbol{H}_{\mathcal{S}_r^c \mathcal{S}_r}^{\mathbb{Q}}(\boldsymbol{H}_{\mathcal{S}_r \mathcal{S}_r}^{\mathbb{Q}})^{-1}\|_{B,1,\infty} \leqslant 1 - \frac{\alpha}{2},$$

where $C$ only depends on $\alpha$, $\Lambda_{\min}(\boldsymbol{H}_{\mathcal{S}_r \mathcal{S}_r})$.

Thanks to the boundedness of the error term $\boldsymbol{e}$, we have similar conclusions to Lemma 12 if $\varepsilon_{\max} \leqslant \frac{\mu}{\sigma}$ holds.

In such wise, the properties in Theorem 9 hold with the same condition on $\lambda_B^*$ and the condition on $\varepsilon_{\max}$ that $\varepsilon_{\max} \leqslant \frac{C\mu}{\sigma \rho_{\max} \rho_{[n]}^{3/2}}$. Since we set $\varepsilon = \frac{\varepsilon_0}{m}$ and define $\varepsilon_{\max} := \max(\sqrt{\varepsilon}, \varepsilon)$, the condition on $\varepsilon_{\max}$ implies that

$$m \geqslant \max\left(\frac{\varepsilon_0 C^2 \sigma^2 \rho_{\max}^2 \rho_{[n]}^3}{\mu^2}, \frac{\varepsilon_0 C \sigma \rho_{\max} \rho_{[n]}^{3/2}}{\mu}\right).$$

Table 2: Comparisons of F1 scores for benchmark datasets and BIC for real-world datasets (backache, voting). BIC is not applicable to skeletons. The best and runner-up results are marked in bold. Significant differences are marked by † (paired t-test, $p < 0.05$).

| Dataset | n | m | Noise | ζ | Wass | KL | Reg | MMPC | GRASP | Wass+HC | KL+HC | Reg+HC | MMPC+HC | GRASP+HC | HC |
|---|---|---|---|---|---|---|---|---|---|---|---|---|---|---|---|
| asia | 8 | 1000 | Noisefree | 0 | 0.7800† | 0.7285† | 0.7897† | **0.9067** | **0.8167** | 0.5123 | 0.6367 | 0.5743 | **0.6667** | **0.6583** | 0.6550 |
| asia | 8 | 1000 | Huber | 0.2 | **0.7333†** | 0.7124† | **0.7297†** | 0.5468 | 0.6570 | **0.3943** | **0.3724** | 0.3487 | 0.2907 | 0.3664 | 0.2183 |
| asia | 8 | 1000 | Independent | 0.2 | **0.6933** | 0.6797 | **0.6868** | 0.6359 | 0.3632† | **0.2676** | **0.2632** | 0.2581 | 0.2469 | 0.1794 | 0.2443 |
| cancer | 5 | 1000 | Noisefree | 0 | **1.0000†** | **1.0000†** | **1.0000†** | 0.6133 | 0.6133 | **0.2800** | **0.2800** | **0.2800** | **0.2800** | **0.2800** | **0.2800** |
| cancer | 5 | 1000 | Huber | 0.5 | **0.9156†** | 0.8933† | **0.9092†** | 0.6133 | 0.5357 | **0.4333** | 0.3833 | **0.4143** | 0.2589 | 0.2714 | 0.2589 |
| cancer | 5 | 1000 | Independent | 0.2 | **0.9048†** | **0.9029†** | 0.8992† | 0.0000 | 0.0000 | 0.0000 | 0.0000 | 0.0000 | 0.0000 | 0.0000 | 0.0000 |
| earthquake | 5 | 1000 | Noisefree | 0 | 0.8447† | 0.9333† | **0.9778** | **1.0000** | **0.9778** | 0.2000 | **0.2500** | **0.2500** | **0.2500** | **0.2500** | 0.2278† |
| earthquake | 5 | 1000 | Huber | 0.2 | **0.7509†** | **0.7509†** | **0.7509†** | 0.5978 | 0.6583† | **0.4618** | **0.4618** | **0.4618** | 0.3860 | 0.4547 | 0.3860 |
| earthquake | 5 | 1000 | Independent | 0.2 | **0.6786†** | **0.6350†** | **0.6350†** | 0.0000 | 0.0000 | 0.0000 | 0.0000 | 0.0000 | 0.0000 | 0.0000 | 0.0000 |
| sachs | 11 | 1000 | Noisefree | 0 | 0.8357† | **0.8402†** | 0.8374† | **0.9697** | 0.7678† | 0.4310† | 0.4535† | 0.4641† | **0.5935** | 0.4112† | **0.5873** |
| sachs | 11 | 1000 | Huber | 0.2 | 0.7765 | **0.8064** | **0.7893** | 0.7498 | 0.5663† | **0.5194** | 0.4815 | 0.4520 | 0.4736 | 0.2380 | **0.5028** |
| sachs | 11 | 1000 | Independent | 0.5 | **0.5268†** | **0.5208†** | 0.5172† | 0.0000 | 0.0000 | 0.0000 | 0.0000 | 0.0000 | 0.0000 | 0.0000 | 0.0000 |
| survey | 6 | 1000 | Noisefree | 0 | **0.6596** | **0.6545** | 0.6506 | 0.6533 | 0.1714† | **0.1789** | **0.1789** | **0.1789** | **0.1789** | 0.0571 | **0.1789** |
| survey | 6 | 1000 | Huber | 0.2 | **0.7303†** | 0.6778† | **0.7095†** | 0.5396 | 0.3810 | **0.1444** | **0.1444** | **0.1444** | **0.1444** | **0.1516** | **0.1444** |
| survey | 6 | 1000 | Independent | 0.2 | **0.6311†** | **0.6705†** | 0.6220† | 0.2032 | 0.0000† | **0.1071** | **0.1071** | **0.1143** | **0.1071** | 0.0000 | **0.1071** |
| alarm | 37 | 1000 | Noisefree | 0 | 0.4750† | 0.7863† | **0.8042†** | **0.8530** | 0.6824† | 0.3483† | 0.4949† | 0.4470† | **0.5635** | **0.4976** | 0.4494† |
| alarm | 37 | 1000 | Huber | 0.2 | 0.1432† | 0.1619† | **0.6571†** | **0.5486** | 0.1945† | 0.2192 | 0.1680† | **0.3148** | **0.2774** | 0.2092† | 0.2582 |
| alarm | 37 | 1000 | Independent | 0.2 | 0.1419† | 0.1448† | **0.5458†** | **0.4309** | 0.2830† | 0.0000 | 0.0000 | 0.0000 | 0.0000 | 0.0000 | 0.0000 |
| barley | 48 | 1000 | Noisefree | 0 | 0.1521† | 0.2632† | 0.4913† | **0.5847** | **0.5636** | 0.1995 | 0.1970† | **0.2503** | **0.2510** | 0.2245 | **0.2526** |
| barley | 48 | 1000 | Huber | 0.2 | 0.1452† | 0.1592† | **0.4027†** | **0.4522** | 0.4000† | 0.1396 | 0.1151 | **0.1658** | 0.1463 | 0.1530 | **0.1685** |
| barley | 48 | 1000 | Independent | 0.2 | 0.1463† | 0.1501† | 0.2767† | **0.4273** | **0.4923†** | 0.0598 | 0.0769 | **0.0838** | 0.0727 | **0.0840** | **0.0838** |
| voting | 17 | 216 | Noisefree | 0 | N/A | N/A | N/A | N/A | N/A | **−2451.8631** | **−2453.2737** | −2453.4091 | −2475.5799 | −2482.3835 | −2456.1489 |
| voting | 17 | 216 | Huber | 0.2 | N/A | N/A | N/A | N/A | N/A | **−4418.9731** | **−4418.9731** | −4487.4544 | −4450.3941 | −4445.0175 | **−4418.9731** |
| voting | 17 | 216 | Independent | 0.2 | N/A | N/A | N/A | N/A | N/A | **−4453.8298** | **−4453.8298** | −4522.5521 | −4465.1076 | −4473.8612 | **−4453.8298** |
| backache | 32 | 90 | Noisefree | 0 | N/A | N/A | N/A | N/A | N/A | −1729.8364 | −1726.8465 | **−1710.7248** | −1719.5002 | **−1713.7583** | −1729.7991 |
| backache | 32 | 90 | Huber | 0.2 | N/A | N/A | N/A | N/A | N/A | **−3186.5001** | **−3186.5001** | **−3186.5001** | **−3186.5001** | **−3186.5001** | **−3186.5001** |
| backache | 32 | 90 | Independent | 0.2 | N/A | N/A | N/A | N/A | N/A | −2800.9386 | −2800.9386 | −2800.9386 | −2800.9386 | −2800.9386 | −2800.9386 |
| connect-4_6000 | 43 | 6000 | Noisefree | 0 | N/A | N/A | N/A | N/A | N/A | **−38956.4300** | **−38956.4300** | −38954.9501 | −39004.8512 | −39933.6041† | **−38956.4300** |
| connect-4_6000 | 43 | 6000 | Huber | 0.2 | N/A | N/A | N/A | N/A | N/A | **−99616.2848** | **−99616.2848** | −102878.2766 | −99673.5320 | −100212.9773 | **−99616.2848** |
| connect-4_6000 | 43 | 6000 | Independent | 0.2 | N/A | N/A | N/A | N/A | N/A | −107403.2543 | −107403.2543 | −107403.2543 | −107403.2543 | −107403.2543 | −107403.2543 |

The final sample complexity becomes

$$m = \mathcal{O}\Big(\frac{C(\varepsilon_0 + \log(n/\delta) + \log \rho_{[n]})\sigma^2 \rho_{\max}^4 \rho_{[n]}^3}{\min(\mu^2, 1)}\Big).$$

□

# D   More Empirical Results

Table 2 lists the complete experimental results.

