# OpenReview forum: "Distributionally Robust Skeleton Learning of Discrete Bayesian Networks"
_NeurIPS.cc/2023/Conference — NeurIPS 2023 spotlight_

### Official Review · Reviewer_Np2S · 2023-06-25

**Soundness:** 3 good
**Presentation:** 4 excellent
**Contribution:** 3 good
**Rating:** 8
**Confidence:** 3

**Summary:**

The authors give novel algorithms to learn the structure of a Bayes net from noisy data.

**Strengths:**

Originality:
Work seems original.

Quality:
Quality is high, writing-wise and content-wise; see questions.

Clarity:
Writing is clear; see questions.

Significance:
Topic is significant.

**Weaknesses:**

No significant issues detected; see questions.

**Questions:**

Line 21:
What do you mean by "ordinal?"

Line 69 to Line 70:
Unclear.

Line 127:
Can you please elaborate on faithfulness?

Line 135:
Please define "unweighted effects."

Can you please explain Equation (2)?

Can you please justify Assumption 2?

Can you please explain Equation (5)?

Proof of Theorem 9:
I do not understand the union bound argument.

Line 304:
Why is it always possible to find such a solution?

Line 313 to Line 314:
Unclear.

Figure 1:
Can you please add some short *further* explanation in the caption?

Experiments look OK!

**Limitations:**

Yes.

---

> ### Author Rebuttal · Authors · 2023-08-09
>
> Thanks for your positive comments. Our rebuttals can be found below.
>
> **Q1: Line 21: What do you mean by "ordinal?"**
>
> If there is an ordinal relationship among variables, the relation is important to learning and inference. For example, if we are to rate the performance of an employee out of very poor, poor, neutral, good, very good, and the employee is actually doing a very good job, assigning a rating of very poor to them is less desirable than a rating of neutral. In contrast, if discrete random variables are not ordinal, different values have no additional interpretation but difference.
>
> **Q2: Line 69 to Line 70: Unclear.**
>
> An ambiguity set is a set distributions that reflects our uncertainty about the underly true distribution. It can be defined in various ways, for example, we can simply use all the distributions, distributions with finite moments or distributions whose support set is included in a box constraint. A typical and intuitive way is to make use of the available samples to form an empirical distribution and considers distributions close to it. The closeness can be measured by a divergence between two distributions.
>
> **Q3: Line 127: Can you please elaborate on faithfulness?**
>
> The faithfulness assumption is a commonly adopted assupmtion for constraint-based causal structure learning methods. A distribution is faithful to a DAG if all (conditional) independences that hold true in the distribution are entailed by the DAG, which is commonly violated in practice.
>
> **Q4: Line 135: Please define "unweighted effects."**
>
> In the unweighted effect coding, the column of the coding matrix sums to zero, whereas the weighted effect coding considers weights defined by samples. For example, given four categories, the set of encoding vectors could be $\\{(1, 0, 0), (0, 1, 0), (0, 0, 1), (-1, -1, -1)\\}$.
>
> **Q5: Can you please explain Equation (2)?**
>
> Equation 2 optimizes the empirical version of Equation 1 with available data samples. The constraint of Equation 1 is abandoned since the neighborhood and graph information are not available in practice. Regularization is a widely adopted technique to combat overfitting in machine learning, which is also the one of the limitations in existing methods that we attempt to address in the paper.
>
> **Q6: Can you please justify Assumption 2?**
>
> Thanks for pointing it out. Assumption 2 is made to guarantee that the neighbor nodes have a noteworthy impact on the node of interest such that the impact can be quantified as a non-zero value bounded away from zero. In this way, neighbor nodes can be identified and distinguished from nodes with an impact close to "zero". We have added this justification in the revised manuscript.
>
> **Q7: Can you please explain Equation (5)?**
>
> Equation 5 is a corollary of Theorem 1 in [1]. For the special case in our work, briefly speaking, the proof writes out the Largrangian dual problem and notes that the supremum over distributions can be simplified into a finite-dimensional problem based on each point of the empirical distribution. We have added the citation to [1] here.
>
> **Q8: Proof of Theorem 9: I do not understand the union bound argument.**
>
> Before taking the union bound, we have attempted to show that for every node, there is a probabilistic bound to recover all of its true neighbor nodes. However, the events of neighborhood recovery for different nodes may or may not be independent. A union bound gives a strict condition for the event that all the sub-events for each node's neighborhood recovery hold true simultaneously, though the upper bound may be far from a tight bound for special cases. Taking the union bound thus allows us to combine the bounds for individual nodes and reach the conclusion that the whole skeleton is consistently recovered.
>
> **Q9: Line 304: Why is it always possible to find such a solution?**
>
> Thanks for pointing it out. First, our objective is non-negative and thus always greater than negative infinity ($>-\infty$). Second, by assuming that the loss function is finite for at least one $\mathbf{W}$, we have a non-trivial optimization problem, otherwise the loss is always equal to $+\infty$ thus trivial. For example, $\mathbf{W} = 0$ yields a finite loss for any finite encodings. Third, there is a typo such that $\bar{B}$ does not have to be finite in the unconstrained case with $\lambda_B^* = 0$. Therefore, as long as it is not a trivial optimization problem, we can always find a solution incurring a finite loss value.
>
> We have revised this paragraph in our revision to make the explanation more clear.
>
> **Q10: Line 313 to Line 314: Unclear.**
>
> Precisely speaking, we refer to a continuous distribution as an absolutely continuous distribution. It means that a continuous distribution will **never** be accounted for by a KL ambiguity set defined by a discrete distribution.
>
> For example, given that the nominal distribution is a univariate discrete distribution with $\mathbb{P}(X = 1) = 0.2$, $\mathbb{P}(X = 2) = 0.3$, $\mathbb{P}(X = 3) = 0.5$, then a uniform distribution over $X \in [1, 3]$ will never fall within the KL ambiguity set because any distribution in the set will have, for example, $\text{Prob}(1.1 < X < 1.9) = 0$ (since only $X = 1, 2, 3$ can have non-zero mass) whereas the absolutely continuous uniform distribution induces $\text{Prob}(1.1 < X < 1.9) = 0.4 > 0$.
>
> We have rephrased the related sentences in our revision.
>
> **Q11: Figure 1: Can you please add some short further explanation in the caption?**
>
> Thanks for your suggestion. The caption has been revised.
>
>
> [1] Jose Blanchet and Karthyek Murthy. Quantifying distributional model risk via optimal transport. Mathematics of Operations Research, 44(2):565–600, 2019.

---

### Official Review · Reviewer_onGN · 2023-06-30

**Soundness:** 3 good
**Presentation:** 2 fair
**Contribution:** 3 good
**Rating:** 5
**Confidence:** 1

**Summary:**

The authors propose methods for learning Bayesian network skeletons based on discrete data when the data is corrupted by arbitrary noise.

The authors show guarantees on discovering the correct skeleton in terms of the required sample size. Some empirical experiments are also presented.

**Strengths:**

+ rigorous mathematical derivations

**Weaknesses:**

- very dense mathematical derivations
- limited and inconclusive empirical results

**Questions:**

To be honest, the level of mathematical detail in the paper is too much for me. It's not that I wouldn't be able to refresh my measure theory and calculus but I just can't afford to spend a week on a review. This is of course primarily my problem, but I have reviewed and published at NeurIPS before, and based on my experience, this paper is mathematically too dense for at least 90% of the NeurIPS community. But perhaps this is fine in case the 10% find it relevant and useful -- I just couldn't possibly tell.

In Sec. 4, explain what exactly you mean my the F1 score: the F1 score is used for binary classification, so I'm guessing you're perhaps looking at presence/absence of each of the edges in the DAG. Please clarify further which methods are proposed in this paper (Wass + KL?) and which are benchmarks (Reg, MMPC, GRASP?) so that it's easier to quickly grasp.

Am I right in that the punchline is Figs 1.b and 1.c? If so, please make it easier for the reader to see this or otherwise they'll miss it.

I take it the results indicate that the proposed methods are otherwise about as good as prior work except that they're better when the failure probability is close to 0.5 (Figs. 1.b and 1.c)? If so, doesn't this require an extremely high rate of failures, to the extent that the data is mostly random noise?

detailed comments:
- p. 1 "DAG structure of a Bayesian network is usually unknown": citation needed (or maybe just say "often")
- p. 2 "contains as subgraphs the skeleton": I'm confused by the plural/singular tenses here: if there's only one skeleton, how is it contained as subgraphs
- p. 2: "(fixed-parameter) tractable": what does this mean?
- p. 2: "observational data is commonly contaminated [...]": again, citation needed
- p. 2: "may produce false edges": may it also miss edges?
- p. 2: "do not assume any specific form of conditional distributions" vs "set of distributions characterized by certain a priori properties": aren't these statements contradictory? please elaborate
- p. 2: "small amount of samples (high-dimensional) and potential perturbations": the word 'high-dimensional' is grammatically speaking placed in an odd spot; also, does "small amount " refer only to samples or also to perturbations -- please clarify the sentence

**Limitations:**

ok

---

> ### Author Rebuttal · Authors · 2023-08-09
>
> Thanks for your very nice and helpful comments.
>
> **Q1: In Sec. 4, explain what exactly you mean my the F1 score: the F1 score is used for binary classification, so I'm guessing you're perhaps looking at presence/absence of each of the edges in the DAG. Please clarify further which methods are proposed in this paper (Wass + KL?) and which are benchmarks (Reg, MMPC, GRASP?) so that it's easier to quickly grasp.**
>
> The F1 score can be leveraged as an evaluation criterion in structured prediction. Specifically, the label of each edge (zero for absence, one for presence) can be considered a binary random variable and the total number of predictions to be evaluated is the total number of edges. The methods proposed and evaluated in the experiments are Wass and KL corresponding to the two DRO methods with Wasserstein and KL ambiguity sets respectively. The baselines are Reg (Bank and Honorio [2020]), MMPC and GRASP. While the HC algorithm is adopted to orient the undirected edges in the predicted skeleton to obtain a valid DAG. We have elaborated on these points in our revision.
>
> **Q2: Am I right in that the punchline is Figs 1.b and 1.c? If so, please make it easier for the reader to see this or otherwise they'll miss it.**
>
> Yes, you are right about this point. Figure 1.b and 1.c demonstrate our superior performance in challenging settings in addition to the comparable performance in noisefree settings in Table 1. We have clarified this punchline in our revision.
>
> **Q3: I take it the results indicate that the proposed methods are otherwise about as good as prior work except that they're better when the failure probability is close to 0.5 (Figs. 1.b and 1.c)? If so, doesn't this require an extremely high rate of failures, to the extent that the data is mostly random noise?**
>
> A failure probability close to 0.5 is a challenging setting that showcases the advantage of DRO methods. Due to space limitation, we did not illustrate results on other datasets with varying failure rate. Usually, a noise probability of about 0.2 is sufficient to distinguish our methods from non-robust methods. It is noteworthy that for constraint-based methods like MMPC, the selected contamination model and distribution is likely to introduce dependency if a variable is binary.
>
> **Q4: detailed comments: p. 1 "DAG structure of a Bayesian network is usually unknown": citation needed (or maybe just say "often")**
>
> Thanks for pointing it out. We have added citations for this statement. In fact, there are often confounders in real world. In fact, we can only use data likelihood for evaluation on real-world datasets.
>
> **Q5: p. 2 "contains as subgraphs the skeleton": I'm confused by the plural/singular tenses here: if there's only one skeleton, how is it contained as subgraphs**
>
> We agree that the usage here may be confusing. What we really mean is that the super-structure is a superset of the skeleton, or the skeleton is a subgraph of the super-structure. We have revised the statement accordingly in our revision.
>
> **Q6: p. 2: "(fixed-parameter) tractable": what does this mean?**
>
> Fixed-parameter tractability is the parametrized analogue of polynomial time tractability, e.g., $f(k) n^{\mathcal{O}(1)}$ for some computation function $f$. We have added a citation in the complexity literature for completeness.
>
> **Q7: p. 2: "observational data is commonly contaminated [...]": again, citation needed**
>
> Citations added.
>
> **Q8: p. 2: "may produce false edges": may it also miss edges?**
>
> Yes, false edges, missing edges, or anything as long as the prediction label (present, absent) does not match the ground truth label. We have revised the sentence accordingly.
>
> **Q9: p. 2: "do not assume any specific form of conditional distributions" vs "set of distributions characterized by certain a priori properties": aren't these statements contradictory? please elaborate**
>
> The former refers to the Bayesian network distribution of interest, e.g., a log-linear distribution, whereas the latter means that the ambiguity set is constructed based on certain a priori properties about our belief on how the uncertainty looks like.
>
> **Q10: p. 2: "small amount of samples (high-dimensional) and potential perturbations": the word 'high-dimensional' is grammatically speaking placed in an odd spot; also, does "small amount " refer only to samples or also to perturbations -- please clarify the sentence**
>
> Thank you for pointing out the grammatical issue. Small amount only refers to samples and high-dimensional means $m < n$ (number of samples is less than number of features). We have revised this part in our revision.

---

> > ### Comment · Reviewer_onGN · 2023-08-21
> > **Changing my score borderline reject => borderline accept**
> >
> > I thank the authors for detailed rebuttals to my own review and those of the other reviewers. I believe that the numerous clarification that the authors promise to make in the revised version will improve the paper enough to bump my score up to borderline accept.
> >
> > The reason why I'm not increasing my score higher is that I'm still not entirely happy with a) the modest empirical results compared to prior methods, and b) the dense mathematical presentation that is making the paper unnecessarily(?) hard to follow, but perhaps this paper will inspire follow up work that can further improve empirical performance and simplify the assumptions and the theoretical results.

---

### Official Review · Reviewer_YEGV · 2023-07-06

**Soundness:** 2 fair
**Presentation:** 3 good
**Contribution:** 2 fair
**Rating:** 4
**Confidence:** 3

**Summary:**

In this paper, the authors revisit the problem of structure learning of Bayesian networks. Here, we are interested to learn the underlying DAG structure that encodes the conditional independence properties of a Bayes net from random samples. A number of algorithms have been proposed to solve this problem over the years which can be broadly classified into two groups: score-based and constraint-based. Despite a lot of research activity over the last several decades the general problem is not yet known to be solvable in polynomial time and several approaches are shown to be NP-hard.

The authors of this paper start with an approach first proposed by Bank and Honorio in 2020: to reduce this discrete structure learning problem to a continuous regression problem based on encoding schemes and surrogate parameters. The idea is that once cast as a regression problem, techniques from structure learning in continuous models such as regression can be employed. The main contribution of this paper is a distributionally robust optimization (DRO) method that builds upon the aforesaid framework to handle noisy data. In this proposed method, the best structure is searched among a ball of certain radius around the empirical distribution that minimizes the regression cost. Specifically, balls according to the Wasserstein distance and KL divergence are considered. The authors give two theorems for each of the above two distance functions under certain assumptions. They further verify the effectiveness of their methods by experimenting with benchmark real and synthetic datasets.



**Strengths:**

The problem considered in extremely important from a theoretical and practical standpoint. The approach followed by the authors to reduce it to a regression problem as in the previous work could be promising for certain datasets.

**Weaknesses:**

A major weakness of the paper is that a number of assumptions are made in order to derive the results, which looks ad-hoc and too restrictive in my opinion. See page 4 of the paper for these assumptions. See also line number 271.

I could not help but notice several vague statements throughout the paper, a few of which are listed below.

Line number 224-233 the NP-hardness argument is very vague. Looks like you are using a brute-force approach, if so, the algorithms wont scale beyond 15 nodes.

Line number 240-241: the citation of Chen and Paschalidis [2018] here is for the first time and sudden. What question are they solving? How is it related to your problem?

Unfortunately, I am not convinced from the experimental results that the proposed algorithms are overall performing better than the existing methods. Many of the entries in table 1 show that other methods are performing superior to the proposed algorithms.

Finally, I found the paper very hard to follow, maybe it reads more complicated than it actually is. The author may try to see whether writing it in a simpler manner with small examples makes the paper more readable or not.


**Questions:**

None

---

> ### Author Rebuttal · Authors · 2023-08-09
>
> Thanks for your insightful comments.
>
> **Q1: A major weakness of the paper is that a number of assumptions are made in order to derive the results, which looks ad-hoc and too restrictive in my opinion. See page 4 of the paper for these assumptions. See also line number 271.**
>
> Thanks for the good question. Assumption 1 simply finds the upper bounds of random errors but does not require them to be small. Assumption 2 makes sense such that if a neighbor node has weight close to zero, it will become indistinguishable from those having little impact on the current node. Our intuitive assumption is that neighbor nodes should have a larger impact than non-neighbor nodes do. Assumption 3 is made to make sure that the solution is unique. However, the original problem is convex without this assumption. Therefore, this assumption only serves as a tool for theoretical analysis but does not lessen its practicability. Assumption 4 has been shown to hold in practice by Bank and Honorio [2020]. We guess you mean the inequality right after line 271. This is not an assumption but the derived condition to be satisfied.
>
> **Q2: I could not help but notice several vague statements throughout the paper, a few of which are listed below. Line number 224-233 the NP-hardness argument is very vague. Looks like you are using a brute-force approach, if so, the algorithms wont scale beyond 15 nodes.**
>
> Line 224-233 argues that the supremum problem is a NP-hard problem where we omit the proof of reducing MAXQP to our problem. We are **not** using a brute-force approach but a random greedy algorithm whose execution time is $\mathcal{O}(n^2m\rho_{max}T)$ where $T$ is the specified number of iterations. The global optimal solution is obtained as $T \rightarrow \infty$. However, data in practice often elicits fewer than $10$ iterations to the find the solution. Given you question about it, we have revised this paragraph and added NP-hardness proof.
>
> **Q3: Line number 240-241: the citation of Chen and Paschalidis [2018] here is for the first time and sudden. What question are they solving? How is it related to your problem?**
>
> Thanks for the good point. Chen and Paschalidis [2018] solved a distributionally robust regression problem with continuous random variables and showed that the distributionally robust optimization problem is equivalent to a regularized ERM problem under some conditions. However, this does not hold in our case. We have added clarifications in the revision.
>
>
> **Q4: Unfortunately, I am not convinced from the experimental results that the proposed algorithms are overall performing better than the existing methods. Many of the entries in table 1 show that other methods are performing superior to the proposed algorithms.**
>
> The results in Table 1 show that our methods (Wass, KL) or the resulting DAG with the aid of HC achieve either the best or the runner-up performance. We will add experimental results on more datasets and make the advantages more clear in the revision.
>
>
> **Q5: Finally, I found the paper very hard to follow, maybe it reads more complicated than it actually is. The author may try to see whether writing it in a simpler manner with small examples makes the paper more readable or not.**
>
> Thanks for the suggestion. We have added small examples, made more concrete explanations and reduced mathematical details in our revision.

---

> > ### Comment · Reviewer_YEGV · 2023-08-18
> > **Response**
> >
> > I have read the rebuttal by the authors. I am still not convinced about the contribution of the paper and am leaning towards rejection.

---

> > > ### Author Response · Authors · 2023-08-20
> > > **More specific**
> > >
> > > Dear Reviewer YEGV,
> > >
> > > Thanks for responding to our rebuttal.
> > >
> > > Could you please be more specific about which answer to your questions you are not convinced of? If you are not convinced about any of them, just let us know and we will reply with a more detailed rebuttal that responds to all your concerns.
> > >
> > > Wish you all the best in NeurIPS 2023 both as a reviewer and author.
> > >
> > > Authors

---

> > > > ### Comment · Reviewer_YEGV · 2023-08-20
> > > > **response**
> > > >
> > > > My main concern includes the assumptions of the paper which I believe is an inherent component of the paper.

---

### Official Review · Reviewer_AT8n · 2023-07-06

**Soundness:** 4 excellent
**Presentation:** 3 good
**Contribution:** 3 good
**Rating:** 7
**Confidence:** 2

**Summary:**

The paper presents a method for learning the structure of discrete Bayesian networks from potentially corrupted data. It utilizes distributionally robust optimization and regression, optimizing the worst-case risk over a set of distributions. The approach works with general categorical variables without assuming specific conditions and provides efficient algorithms. Non-asymptotic guarantees for successful structure learning are derived, and numerical experiments validate its effectiveness.

**Strengths:**

Originality:
The paper addresses the challenge of missing data in distributionally robust skeleton learning. This aspect adds a unique and valuable contribution to the field.

Quality:
The quality of writing in the paper is commendable. The ideas and concepts are presented effectively, and the paper demonstrates a strong grasp of the subject matter.

Clarity:
The paper's clarity is notable, as it is organized in a logical manner where each section builds upon the previous ones. The flow of information is well-structured, making it easy for readers to follow the paper's content.

Significance:
The task of building discrete Bayesian networks from potentially corrupted data holds significant importance, as it has wide-ranging applications in various fields. By addressing the challenge of distributionally robust skeleton learning, the paper contributes to the advancement of techniques that can handle data uncertainties and enable more accurate modeling and inference in real-world scenarios. The significance of this research lies in its potential to enhance decision-making processes, predictive modeling, and knowledge discovery across diverse domains.

**Weaknesses:**

- The author's assertion that they had limited space to relate their work to the existing literature is understandable. However, it is crucial to provide a clear positioning of their own work within the broader research landscape. Emphasizing the novelty and distinguishing features of their approach will strengthen the paper's contribution and demonstrate its unique value.

- The paper lacks clarity on whether cross-validation or hyperparameter tuning was performed during the experiments. It would be beneficial to address this issue explicitly to ensure the robustness of the results. Additionally, to further improve the reliability of the findings, it is recommended to conduct a rigorous statistical test, such as the corrected Student's t-test proposed by Nadeau and Bengio [1], using 15 hold-out folds. Incorporating such statistical analyses will enhance the credibility and validity of the experimental results.

[1] https://link.springer.com/article/10.1023/A:1024068626366

**Questions:**

l 105: How is T defined in A_{ST}?

**Limitations:**

- The proposed method accounts for the effect of outliers by optimizing the worst-case risk over a family of distributions within bounded Wasserstein distance or KL divergence. However, the extent to which the method can effectively handle outliers and their impact on the learned network structure should be further explored and evaluated.
- The proposed approach is designed for general categorical random variables without assuming faithfulness, an ordinal relationship, or a specific form of conditional distribution. While this makes the approach flexible, it may also limit its applicability to certain domains or types of data. Investigating the generalizability of the approach to a broader range of variable types or distributions could be a potential limitation.

---

> ### Author Rebuttal · Authors · 2023-08-09
>
> Thanks for your positive comments. Please see our point-by-point responses as follows.
>
> **Q1: The author's assertion that they had limited space to relate their work to the existing literature is understandable. However, it is crucial to provide a clear positioning of their own work within the broader research landscape.**
>
> Thanks for the suggestion. We have added more related work to put our work in a more clear posistioning within the research landscape by assuming that extra space is available.
>
> **Q2: The paper lacks clarity on whether cross-validation or hyperparameter tuning was performed during the experiments. ... Additionally, to further improve the reliability of the findings, it is recommended to conduct a rigorous statistical test...**
>
> We tune the hyper-parameters based on randomly generated Bayesian network with the same number of nodes as indicated by the input data. We will include this information as well as statistical tests in our revision.
>
> **Q3: l 105: How is T defined in A_{ST}?**
>
> $\mathcal{T}$ is a subset of $\\{1, \dots, m\\}$. Revised.
>
> **Q4: The proposed method accounts for the effect of outliers by optimizing the worst-case risk over a family of distributions within bounded Wasserstein distance or KL divergence. However, the extent to which the method can effectively handle outliers and their impact on the learned network structure should be further explored and evaluated.**
>
> The class of distributionally robust optimization methods account for outliers in terms of uniform performance over a set of distributions and the outlier is assumed to be identifiable through divergence. The way of robustly modeling is general but lacking case-by-case analysis for a specific or given type of corruption. We aim to demonstrate a robust learning method that subsumes existing methods as special cases while enjoying certain robustness. There is also an important issue on robust identifiability of Bayesian network or causal structures. These are all interesting future directions to study further. Discussion on this point has been added in the revision.
>
> **Q5: The proposed approach is designed for general categorical random variables without assuming faithfulness, an ordinal relationship, or a specific form of conditional distribution. While this makes the approach flexible, it may also limit its applicability to certain domains or types of data. Investigating the generalizability of the approach to a broader range of variable types or distributions could be a potential limitation.**
>
> Good point. We agree that we sacrifice some efficiency and applicability to special domains for general formulations and bounds. For example, if the possible values of the random variables are ordinal or continuous, or the variables follow a Gaussian or Poisson distribution, a more efficient algorithm and a tighter bound can be established. We have added discussion on this point at the end of the manuscript in revision.

---

> > ### Comment · Reviewer_AT8n · 2023-08-14
> >
> > Thank you for taking the time to read my comments.
> >
> > The authors have taken all my suggestions into account and provided valuable feedback on my questions. I think that by clarifying the limitations of the method and improving the related work section, the paper will be of more value to the community.
> >
> > I increase my vote from 6 to 7.

---

### Official Review · Reviewer_Hyho · 2023-07-31

**Soundness:** 3 good
**Presentation:** 4 excellent
**Contribution:** 3 good
**Rating:** 8
**Confidence:** 4

**Summary:**

The paper considers the problem of learning the exact skeleton of general discrete Bayesian networks from potentially corrupted data. The estimator optimizes the worst case risk over a family of distributions within bounded Wasserstein distance and KL divergence to the empirical distribution. Under mild conditions, the paper presents non-asymptotic guarantees for successful structure learning with logarithmic sample complexities for bounded degree graphs. Numerical studies on synthetic and real datasets validates the performance of the estimator.

**Strengths:**



**Weaknesses:**



**Questions:**

1) Is it possible to extend the DRO framework to learn the structure of continuous Bayesian networks from corrupted samples. A high level discussion regarding this would shed more light on the limitations of the present contributions.

2) Since no assumptions on the underlying true distribution is made, it is only possible to recover the structure upto a Markov equivalence class. However the proposed approach overcomes this limitation and recovers the true skeleton with high probability. A discussion as to why DRO framework achieves this will be highly appreciated.

**Limitations:**

---

> ### Author Rebuttal · Authors · 2023-08-09
>
> Thanks for the positive feedback. Please find our rebuttals below.
>
> **Q1: Is it possible to extend the DRO framework to learn the structure of continuous Bayesian networks from corrupted samples?**
>
> The proposed DRO framework is based on a regression approach, which is naturally tailored to tackling continuous variables. For example, by assuming a linear Gaussian structural equation model, we can simply let the encoding function be the identity mapping. We have added the discussion on this point in our revised manuscript.
>
> **Q2: A discussion as to why DRO framework achieves this (recovers the true skeleton with high probability without distributional assumptions) will be highly appreciated.**
>
> Our methods are based on a series of work on DAG structure learning in polynomial time. Since the non-robust method our work is built upon is a special case of our methods, the desirable properties are naturally inherited. Moreover, the proposed DRO framework enjoys robustness that is modeled explicitly and converges to the true global optima as $m \rightarrow \infty$ given clean data samples as data distribution uncertainty vanishes. The Discussion section has been revised to address this issue.

---

> > ### Comment · Reviewer_Hyho · 2023-08-14
> >
> > The authors have addressed my concerns and I am quite happy with the response.

---

### Author Rebuttal · Authors · 2023-08-09

Dear reviewers,

Thanks for your helpful reviews that would certainly improve our manuscript. Please see our rebuttals after individual reviews. We are more than glad to have further discussions or make clarifications in the discussion period.

We would like to restate that our main contribution is the first robust Bayesian network skeleton learning method with polynomial time and sample complexities without distributional assumptions.

Best,
Authors

---

### Decision · Program_Chairs · 2023-09-21

**Decision:**

Accept (spotlight)

**Comment:**

This paper considers the problem of learning the structure of Discrete Bayesian Networks from noisy data. The review process has revealed that the paper is theoretically sound, well written, and makes an important contribution of interest to the NeurIPS audience.